# NAc-VTA circuit underlies emotional stress-induced anxiety-like behavior in the three-chamber vicarious social defeat stress mouse model

Guangjian Qi[1,2,3,8], Pei Zhang [1,4,5,8], Tongxia Li[1,8], Ming Li[1], Qian Zhang[1], Feng He[1], Lijun Zhang[1], Hongwei Cai[1], Xinyuan Lv[1], Haifa Qiao [2,3], Xiaoqian Chen[4,5,6], Jie Ming[7✉] & Bo Tian [1,4,5✉]

Emotional stress is considered a severe pathogenetic factor of psychiatric disorders. However, the circuit mechanisms remain largely unclear. Using a three-chamber vicarious social defeat stress (3C-VSDS) model in mice, we here show that chronic emotional stress (CES) induces anxiety-like behavior and transient social interaction changes. Dopaminergic neurons of ventral tegmental area (VTA) are required to control this behavioral deficit. VTA dopaminergic neuron hyperactivity induced by CES is involved in the anxiety-like behavior in the innate anxiogenic environment. Chemogenetic activation of VTA dopaminergic neurons directly triggers anxiety-like behavior, while chemogenetic inhibition of these neurons promotes resilience to the CES-induced anxiety-like behavior. Moreover, VTA dopaminergic neurons receiving nucleus accumbens (NAc) projections are activated in CES mice. Bidirectional modulation of the NAc-VTA circuit mimics or reverses the CES-induced anxiety-like behavior. In conclusion, we propose that a NAc-VTA circuit critically establishes and regulates the CES-induced anxiety-like behavior. This study not only characterizes a preclinical model that is representative of the nuanced aspect of CES, but also provides insight to the circuit-level neuronal processes that underlie empathy-like behavior.

[1] Department of Neurobiology, School of Basic Medicine, Tongji Medical College, Huazhong University of Science and Technology, Wuhan, Hubei Province 430030, P. R. China. [2] College of Acupuncture & Massage, Shaanxi University of Chinese Medicine, Xixian New Area, Shaanxi Province 712046, P. R. China. [3] Key Laboratory of Acupuncture & Medicine of Shaanxi Province, Shaanxi University of Chinese Medicine, Xixian New Area, Shaanxi Province 712046, P. R. China. [4] Institute for Brain Research, Huazhong University of Science and Technology, Wuhan, Hubei Province 430030, P. R. China. [5] Key Laboratory of Neurological Diseases, Ministry of Education, Wuhan, Hubei Province 430030, P. R. China. [6] Department of Pathophysiology, School of Basic Medicine, Tongji Medical College, Huazhong University of Science and Technology, Wuhan, Hubei Province 430030, P. R. China. [7] Department of Breast and Thyroid Surgery, Union Hospital, Huazhong University of Science and Technology, Wuhan, Hubei Province 430022, P. R. China. [8] These authors contributed equally: Guangjian Qi, Pei Zhang, Tongxia Li. ✉email: mingjiewh@126.com; tianbo@mails.tjmu.edu.cn

Emotional stress, different from the physical stress, is a form of tension in which the stressor is a psychological response to a situation subjectively perceived as traumatic[1]. Emotional challenges are ubiquitous in everyday lives, and emotional stressors can upset body homeostasis[2]. In addition to impairing the function of peripheral organs, maladaptive and uncontrollable responses to chronic emotional stress have been linked to various disorders in the central nervous system, such as anxiety, post-traumatic stress disorder (PTSD), seizures, and ischemia stroke[3–5].

In the mammalian brain, several brain areas have been implicated in coping with emotional stress. These encompass the medial prefrontal cortex, amygdala, anterior insula, hippocampus, and striatum[6–8]. For example, the potent inhibitory projections from the ventromedial prefrontal cortex (vmPFC) to the amygdala, have been related to various processes that may facilitate stress coping[9]. Notably, human neuroimaging evidence suggests that vmPFC is one of the critical loci of the adaptive behavioral coping circuit that plays a role in the regulation of anxious emotion[10,11]. Many previous studies have proved that the stress system, immune system, and oxidative system can form a coactivation state under emotional stress[12]. Clinical and preclinical evidence indicates that the vicious circle of coactivation of pathways of those systems may lead to some of the behavioral and biochemical changes that are also typical in mood disorders[13]. Despite accumulated information about the biological factors and brain regions involved in emotional stress, the underlying, circuit-level, neural mechanisms remain unknown.

Its subjective nature makes emotional stress more challenging to study, and animal models of purely emotional stress are scarce. Therefore, the development of an ideal animal model could improve progress in emotional stress research. In the following set of experiments, we improved the previous vicarious social defeat stress (VSDS) model[14,15] and redesigned the three-chamber VSDS (3C-VSDS) model with high face, construct, and predictive validity, to study the neurobiological consequences of emotional stress. In this animal model, the observer mouse is conditioned vicariously for scene-dependent emotion (chronic emotional stress, CES) by observing a demonstrator mouse that receives repetitive social defeats (chronic social defeat stress, CSDS). This direct observation and sharing of aversive experience can enhance CES responses in mice via emotional contagion. The 3C-VSDS model may be a valuable tool for directly evaluating CES processes in mice at the circuit- and neuron-specific levels.

One region of interest is the ventral tegmental area (VTA), which signals reward and aversion[16]. Multiple psychiatric disorders are associated with altered brain circuitry, which affects reward and aversion processing[17]. The VTA is a heterogeneous structure, with DA neurons comprising 65% of the population. However, whether VTA$^{DA}$ neurons mediate or modulate CES-induced behaviors is currently unknown.

The current study combined viral tracing, in vivo electrophysiological and Ca$^{2+}$ recordings, and chemogenetic and optogenetic methods in the 3C-VSDS model to demonstrate that VTA$^{DA}$ neurons are essential for the CES-induced anxiety-like behavior. To observe how these neurons' activity is modulated, we examined the functional organization of the nucleus accumbens (NAc) to the VTA circuit and its role in this behavioral phenotype. Based on these findings, we propose that a NAc–VTA circuit critically establishes and regulates the anxiety-like behavior of CES.

## Results

### CES produces anxiety-like behavior and transient social dysfunction but not depression-like behavior.
To investigate the effect of CES on mice's behaviors, we employed a customized cage divided into three chambers with perforated plexiglass partitions (Fig. 1a). The mouse in the right chamber (intruder) was introduced into the middle chamber and subjected to physical contact and defeat by the aggressive CD1 mouse (resident). Without receiving direct physical stimuli, the CES mouse in the left chamber is vicariously conditioned for scene-dependent emotion by observing the CSDS mouse that received repetitive social defeats in the middle chamber (Fig. 1b). Control mice were housed with CD1 mice in the three-chamber cage, but received no physical or emotional stimuli (Fig. 1b).

Mice were assessed for behavioral deficits following the 10-day exposure period (Fig. 1c). First, to address whether CES increased vulnerability to social defeat-induced social avoidance, the social interaction test (SIT) was conducted at different time points. The CSDS mice exhibited, and sustained, dramatically low social interaction levels, whereas the CES mice showed transient social disorder in the early stage of the experiment (Day 3–7), but with the extension of time, the social interaction time returned to normal (Fig. 1d). On day 11, CSDS mice spent significantly less time in the interaction zone, whereas the CES and control mice showed similar interaction levels (Fig. 1e, f). We also analyzed the total distance traveled in the SIT arena. Statistical analyses suggested that the variance tendency of total distance traveled between the three groups of mice was very similar to that of interaction time (Supplementary Fig. 1a, b). These results indicate that CES produces transient social dysfunction.

Next, the open-field test (OFT) and the elevated plus maze (EPM) test were conducted to assess anxiety-like behavior. CES and CSDS mice displayed decreased central exploration time and central distance relative to the control group (Fig. 1g–i). Locomotor activity, measured by total distance traveled, did not differ between the CES and control groups, but decreased in the CSDS group (Fig. 1j). Consistent with the OFT results, CES and CSDS mice spent significantly less time in the EPM open arms (Fig. 1k, l). No significant differences were observed in the number of open-arm entries (Fig. 1m). These data demonstrate that CES and CSDS mice develop an anxiety-like phenotype after repeated emotional or physical stress exposure.

The tail-suspension test (TST) was used to measure behavioral despair/helplessness. The CSDS mice showed increased immobility time compared with control mice, which was not found in the CES mice (Fig. 1n). Finally, the sucrose-preference test (SPT) was used to measure anhedonia, a core depression symptom. CSDS, but not CES, significantly reduced sucrose preference compared with control mice. However, total fluid consumption did not differ (Fig. 1o, p).

Together, these results demonstrate that CSDS resulted in anxiety-like and depression-like phenotypes, and social dysfunction, whereas CES produced a lesser affective phenotype, characterized by anxiety-like behavior and transient social dysfunction.

### VTA$^{DA}$ neurons mediate the CES-induced anxiety-like behavior.
The VTA is involved in emotion-related behaviors, particularly in processing stressful events[18]. Therefore, we examined VTA neuron activity following single emotional stress (SES). There was a significant increase in the number of c-fos-positive cells in the VTA area compared with control mice (Fig. 2a–c). In addition, we also measured elevated c-fos protein expression in the red nucleus, which was dorsal to the VTA (Supplementary Fig. 2a).

To further explore which aspects of VTA function underlie CES, VTA$^{DA}$ neuron activity was recorded using in vivo electrophysiology following CES, and the firing of putative DA neurons was sorted and analyzed. The in vivo recordings showed that the basal firing rate of VTA$^{DA}$ neuron was almost the same in CES and control mice under freely moving condition (Fig. 2d–f).

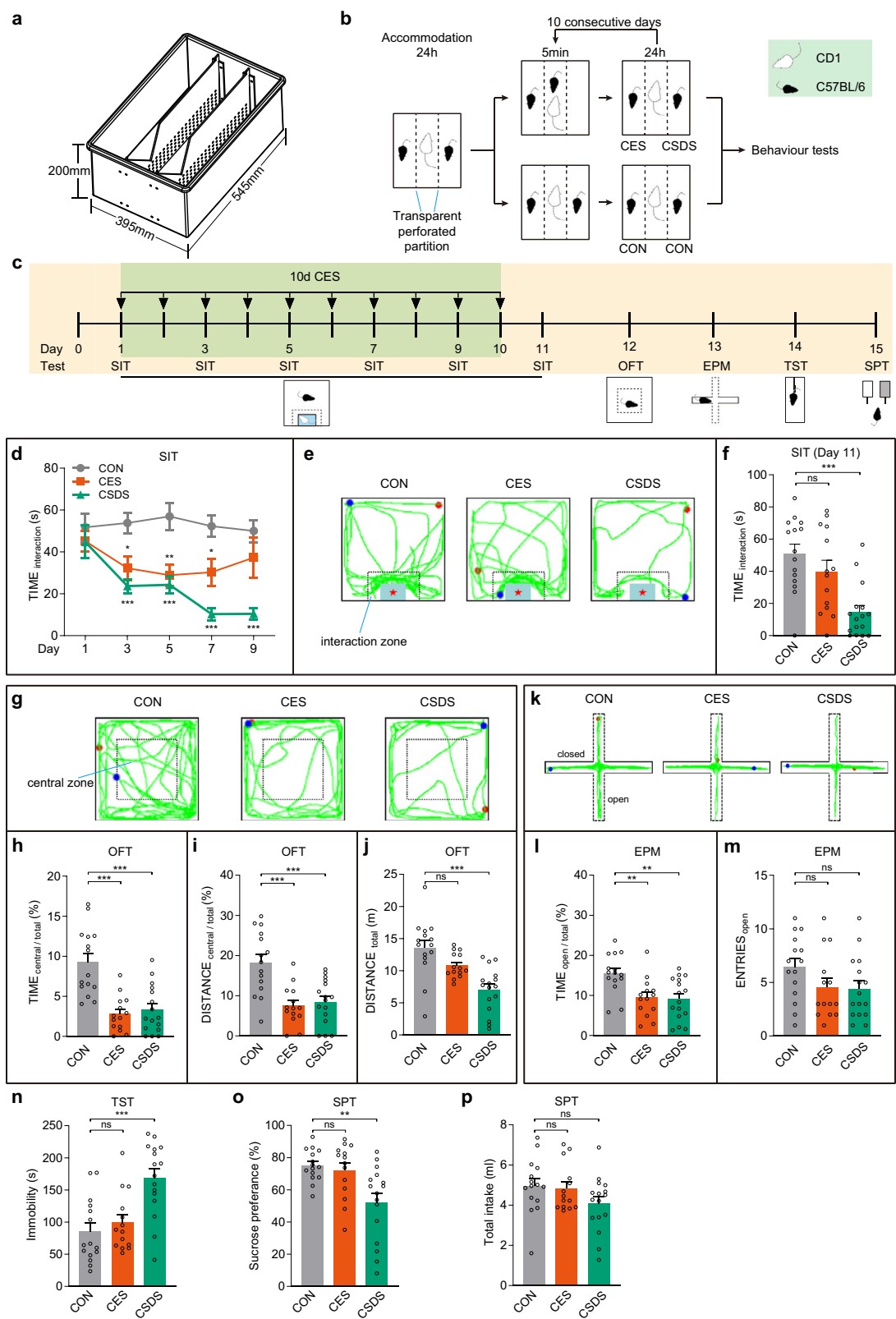

Because of the difficulties of in vivo electrophysiology in analyzing the neural activity of normal-moving mice in behavioral testing, we investigated the dynamics of VTA$^{DA}$ neuron activation and their role in anxious behaviors by in vivo fiber photometry, which could record intracellular calcium (Ca$^{2+}$) transients in behaving mice. The genetically encoded Ca$^{2+}$ indicator GCaMP6s was expressed in VTA$^{DA}$ neurons by stereotaxic infusion of AAV2/9-DIO-GCaMP6s and AAV2/9-tyrosine hydroxylase (TH)-Cre into the VTA area of wild-type C57BL/6 mice (Fig. 2g, h). After two weeks' rest, optic-fiber cannula was implanted over the same site (Fig. 2g, h). GCaMP6s expression in the VTA$^{DA}$ neurons was confirmed post hoc by colocalization with TH (Fig. 2i; Supplementary Fig. 3a, b). We found that the onset of anxiety-inducing events, in this case, the target mice received emotional stress by observing their conspecific partners

**Fig. 1 CES produces anxiety-like behavior and transient social dysfunction, but not depression-like behavior. a** Dimension and diagram of the customized three-chamber emotional stress cage. The length, width, and height of the customized three-chamber cage are 545 mm, 395 mm, and 200 mm, respectively. **b** Behavioral modeling paradigm of CES and CSDS. CES chronic emotional stress, CSDS chronic social defeat stress, CON control. **c** Experimental scheme of CES modeling and behavioral tests. SIT social interaction test, OFT open-field test, EPM elevated plus maze, TST tail-suspension test, SPT sucrose-preference test. **d–f** Results of SIT. **d** Time course of social interaction time in SIT (two-way RM ANOVA, group: $P = 5.71 \times 10^{-10}$, time: $P = 0.0053$, interaction: $P = 0.0385$; post hoc Tukey's test, day1: CON vs. CES: $P = 0.6937$, CON vs. CSDS: $P = 0.6472$, day3: CON vs. CES: $P = 0.0221$, CON vs. CSDS: $P = 0.0004$, day5: CON vs. CES: $P = 0.0018$, CON vs. CSDS: $P = 0.0001$, day7: CON vs. CES: $P = 0.0192$, CON vs. CSDS: $P = 6.10 \times 10^{-7}$, day9: CON vs. CES: $P = 0.2583$, CON vs. CSDS: $P = 2.70 \times 10^{-6}$). **e** Representative exploration traces in SIT on day 11. **f** Statistical results of social interaction time on day 11 (Kruskal–Wallis test, $P = 0.0005$; CON vs. CES: $P = 0.9752$, CON vs. CSDS: $P = 0.0005$). **g–j** Results of OFT. **g** Representative locomotion traces in OFT. **h** Percentage of time spent in the central zone of OFT (one-way ANOVA, $P = 1.95 \times 10^{-6}$; post hoc Tukey's test, CON vs. CES: $P = 9.61 \times 10^{-6}$, CON vs. CSDS: $P = 2.30 \times 10^{-5}$). **i** Percentage of distance traveled in the central area of OFT (one-way ANOVA, $P = 3.16 \times 10^{-5}$; post hoc Tukey's test, CON vs. CES: $P = 0.0001$, CON vs. CSDS: $P = 0.0003$). **j** The total distance traveled in OFT (one-way ANOVA, $P = 2.88 \times 10^{-5}$; post hoc Tukey's test, CON vs. CES: $P = 0.0998$, CON vs. CSDS: $P = 1.76 \times 10^{-5}$). **k–m** Results of EPM. **k** Representative exploration traces in EPM. **l** Percentage of time spent in the open arms of EPM (one-way ANOVA, $P = 0.0016$; post hoc Tukey's test, CON vs. CES: $P = 0.0070$, CON vs. CSDS: $P = 0.0032$). **m** Number of open-arm entries in EPM (one-way ANOVA, $P = 0.1305$; post hoc Tukey's test, CON vs. CES: $P = 0.2197$, CON vs. CSDS: $P = 0.1587$). **n** Immobility time in the TST (one-way ANOVA, $P = 9.91 \times 10^{-5}$; post hoc Tukey's test, CON vs. CES: $P = 0.7461$, CON vs. CSDS: $P = 0.0002$). **o** Percentage of sucrose intake in SPT (one-way ANOVA, $P = 0.0014$; post hoc Tukey's test, CON vs. CES: $P = 0.8943$, CON vs. CSDS: $P = 0.0024$). **p** Total fluid consumption in SPT (one-way ANOVA, $P = 0.1401$; post hoc Tukey's test, CON vs. CES: $P = 0.9671$, CON vs. CSDS: $P = 0.1603$). $n = 15$ for CON mice, $n = 14$ for CES mice, and $n = 16$ for CSDS mice. The red and blue dots in the OFT, EPM, and SIT locomotion traces reflect the start and end points of the mouse, respectively. All data are shown as mean ± SEM. The $P$ values were calculated by CES or CSDS versus the corresponding control group. ns not significant, $^{*}P \leq 0.05$, $^{**}P \leq 0.01$, $^{***}P \leq 0.001$.

attacked by CD1 mice, significantly activated VTA$^{DA}$ neurons, showing remarkable calcium transients (Fig. 2j, k). Similarly, the AAV2/9-DIO-GCaMP6s was injected into the VTA of dopamine transporter (DAT)-Cre transgenic mice and VTA$^{DA}$ neurons showed significant activation of GCaMP6s activity following emotional stress exposure (Supplementary Fig. 4a–f). These findings reveal that VTA$^{DA}$ neurons were indeed strongly activated by emotional stress, indicating that VTA$^{DA}$ neurons participate in emotional stress processing.

Next, we investigated how VTA$^{DA}$ neurons are engaged during the exploration of innately anxiogenic environments in freely moving mice. C57BL/6 mice were infused with AAV2/9-DIO-GCaMP6s and AAV2/9-TH-Cre into the VTA, and optic-fiber cannula was implanted over the same site. After a week's rest, 10-day of CES modeling was conducted, followed by EPM test and simultaneous VTA$^{DA}$ neuronal activity detection using position-synchronized in vivo calcium imaging (Fig. 2l, m). CES mice showed decreased exploring time in the open arms and increased time in the closed arms of EPM (Fig. 2n, o). Meanwhile, CES mice exhibited a significantly higher average Ca$^{2+}$ activities and higher variation rate of Ca$^{2+}$ fluorescence in the VTA$^{DA}$ neurons during open-arm exploration, but not closed-arm exploration (Fig. 2p–s). We also tested whether VTA$^{DA}$ neurons are engaged during the SIT. Compared with control mice, CES mice showed no significant differences in time in the interaction zone, average Ca$^{2+}$ activities, and variation rate of Ca$^{2+}$ fluorescence (Fig. 2t–x). Similarly, employing the same strategy in DAT-Cre mice, we also found that the VTA$^{DA}$ neurons of DAT-Cre mice showed significant activation of GCaMP6s activity in anxiety-inducing contexts (Supplementary Fig. 5a–m), which is in agreement with our above studies. Together, these results indicate that the activation of VTA$^{DA}$ neurons by CES is involved in the anxiety-like behavior in the innate anxiogenic environment, but does not directly affect mice's social interaction behavior.

**Chemogenetic activation of VTA$^{DA}$ neurons directly triggers anxiety-like behavior.** The above results linked the VTA$^{DA}$ neuron hyperactivity to the anxiety-like behavior of CES. Therefore, we tested whether the activation of VTA$^{DA}$ neurons is sufficient and/or necessary to induce anxiety-like behavior.

We bilaterally infected AAV2/9-DIO-hM3Dq-mCherry in VTA of DAT-Cre mice and measured the anxiety-like and social interaction behavior induced by VTA$^{DA}$ neuron activation by clozapine-N-oxide (CNO) administration (Fig. 3a, b). Post hoc immunofluorescent examination verified that hM3Dq-mCherry expression was primarily restricted to DA neurons in the VTA region (Fig. 3c, d). We found that compared with mCherry and saline controls, mice with CNO injections significantly reduced the exploration time and travel distance in the central area of OFT, but showed increment on locomotor activities (Fig. 3e–h). Similarly, CNO-mediated VTA$^{DA}$ neuron activation decreased the time spent in the open arms (Fig. 3i, j). No significant differences were observed in the number of open-arm entries (Fig. 3k). Moreover, we assessed the effects of VTA$^{DA}$ neuron activation in the SIT. CNO injection did not alter the time spent in the social interaction zone compared with the mCherry and saline controls (Fig. 3l, m).

Besides, employing the same strategy in wild-type mice, the effects of chemogenetic activation of VTA$^{DA}$ neurons on behavioral changes were also tested. To prove the experimental validity of chemogenetic activation, we unilaterally infected with AAV2/9-DIO-hM3Dq-mCherry and AAV2/9-TH-Cre, and implanted an electrode in the VTA of C57BL/6 mice (Supplementary Fig. 6a, b). We measured the effects of CNO on the firing rate of putative DA neurons in mice, using multichannel in vivo recording techniques. CNO administration significantly increased spike rates in hM3Dq-expressing VTA$^{DA}$ neurons (Supplementary Fig. 6c, d). Post hoc immunofluorescent examination verified that hM3Dq-mCherry expression was primarily restricted to DA neurons in the VTA region (Supplementary Fig. 6e, f; Supplementary Fig. 7a). Next, we bilaterally infected AAV2/9-DIO-hM3Dq-mCherry and AAV2/9-TH-Cre in VTA and measured the anxiety-like and social interaction behavior induced by VTA$^{DA}$ neuron activation by CNO administration (Supplementary Fig. 6g, h). We also found that activating DA neurons in the VTA contributed to the expression of anxiety, but displayed little response to social interaction behavior (Supplementary Fig. 6i–q). These results demonstrate that VTA$^{DA}$ neuron activation produces anxiety-like behavior, but does not affect social interaction behavior.

**Chemogenetic inhibition of VTA$^{DA}$ neurons promotes resilience to the CES-induced anxiety-like behavior.** The above activation results suggest that VTA$^{DA}$ neuron hyperactivity is a trait of anxious states. Therefore, we hypothesized that artificially

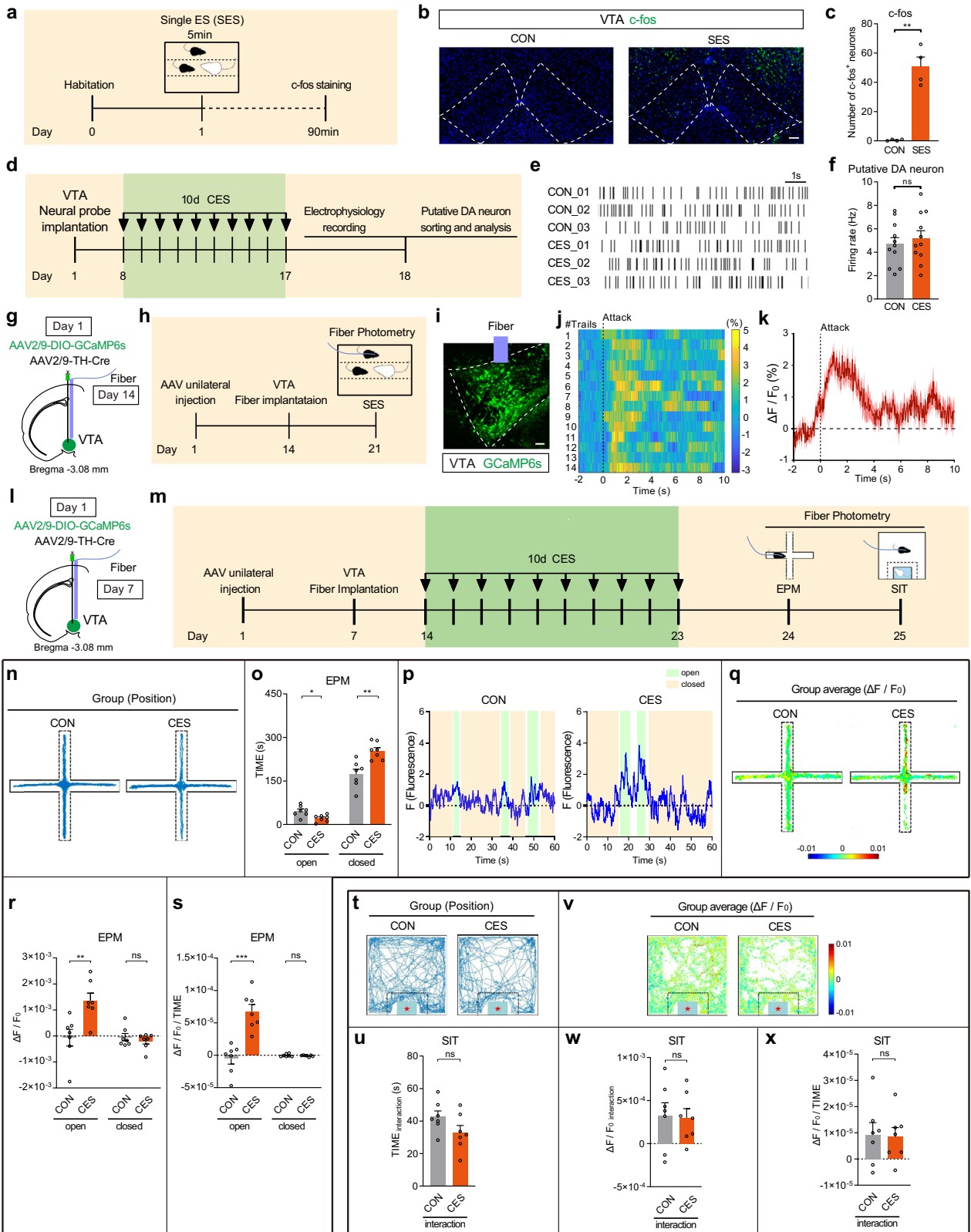

inhibiting $VTA^{DA}$ neuron activity would promote resilience to the anxiety-like behavior of CES.

To prove the experimental validity of chemogenetic inactivation, we unilaterally infected with AAV2/9-DIO-hM4Di-mCherry and AAV2/9-TH-Cre, and implanted an electrode in the VTA (Fig. 4a, b). CNO injection significantly suppressed activity in hM4Di-expressing putative $VTA^{DA}$ neurons (Fig. 4c, d). Post hoc

immunofluorescent examination verified that hM4Di-mCherry expression was primarily restricted to DA neurons in the VTA region (Fig. 4e, f; Supplementary Fig. 8a).

Next, the effects of chemogenetic inhibition of $VTA^{DA}$ neurons in vivo on behavioral changes were tested. C57BL/6 mice were bilaterally infused with AAV2/9-DIO-hM4Di-mCherry and AAV2/9-TH-Cre in the VTA (Fig. 4g) and exposed to 10-day

**Fig. 2 VTA$^{DA}$ neurons mediate the CES-induced anxiety-like behavior. a–c** Results of c-fos staining in single emotional stress (SES) model. **a** Experimental diagram for SES modeling. **b** The c-fos expression in VTA area in SES and control (CON) mice. Scale bar, 100 μm. **c** Statistical results of c-fos-positive cells in CON and SES mice (two-tailed unpaired $t$-test with Welch's correction, $P = 0.0045$). $n = 4$ mice for each group. **d–f** Detection of the basic firing rate of putative VTA$^{DA}$ neurons by in vivo electrophysiological technique in free-moving CES mice. **d** Experimental scheme. **e** Raster plot of spontaneous spikes of putative VTA$^{DA}$ neurons. **f** Firing rate of putative VTA$^{DA}$ neurons in CES and CON mice (two-tailed unpaired $t$-test, $P = 0.5557$). $n = 11$ neurons from 3 mice/group. **g–k** Detection of neuronal activity of VTA$^{DA}$ neurons of wild-type C57BL/6 mice in anxiety-inducing events by in vivo calcium recording. **g** Schematic of viral injection and optical fiber implantation. **h** Experimental scheme of anxiety-inducing events. **i** Representative image of VTA-injection sites. Scale bar, 50 μm. **j** Trial-by-trial heatmap of anxiety-inducing event-evoked Ca$^{2+}$ transients. The black dotted line represents the onset of anxiety-inducing event exposure. Each trail represents the target mice that observed their conspecific received attacks by CD1 aggressor. **k** Perievent plot of average Ca$^{2+}$ transients ($n = 4$ mice). Black dotted line, the onset of emotional stress exposure (target mice observe a conspecific attacked by CD1 mice). Surrounding shaded areas indicate error bars (SEM). **l–x** Detection of neuronal activity of VTA$^{DA}$ neurons of CES mice in anxiety-inducing contexts by in vivo calcium recording, $n = 7$ mice for each group. **l** Schematic of viral injection and optical fiber implantation. **m** Experimental scheme of CES modeling and behavioral tests. **n–s** Results of EPM. **n** Group of exploration traces in EPM. **o** Statistics of time spent in the open arm and closed arm of EPM (open arms: two-tailed unpaired $t$-test, $P = 0.012$; closed arms: two-tailed unpaired $t$-test, $P = 0.0020$). **p** The representative traces of Ca$^{2+}$ fluorescence in CON and CES mouse. The transparent cyan background and papaya whip color display the mice located in the open arms and closed arms, respectively. **q** Group average of Ca$^{2+}$ activities of VTA$^{DA}$ neurons in EPM. **r** Statistics of average Ca$^{2+}$ activities of VTA$^{DA}$ neurons in the open arm and closed arm of EPM (open arms: two-tailed unpaired $t$-test, $P = 0.0062$; closed arms: two-tailed unpaired $t$-test, $P = 0.4331$). **s** Statistics of variation rate of Ca$^{2+}$ fluorescence in the open arm and closed arm of EPM (open arms: two-tailed unpaired $t$-test, $P = 0.0002$; closed arms: two-tailed unpaired $t$-test, $P = 0.5442$). **t–x** Results of SIT. **t** Group of exploration traces in SIT. **u** Statistics of time spent in the social interaction zone of SIT (two-tailed unpaired $t$-test, $P = 0.1043$). **v** Group average of Ca$^{2+}$ activities of VTA$^{DA}$ neurons in SIT. **w** Statistics of average Ca$^{2+}$ activities of VTA$^{DA}$ neurons in the social interaction zone of SIT (two-tailed unpaired $t$-test, $P = 0.8763$). **x** Statistics of variation rate of Ca$^{2+}$ fluorescence in the social interaction zone of SIT (two-tailed unpaired $t$-test, $P = 0.8843$). All data are shown as mean ± SEM. ns not significant, $^*P \leq 0.05$, $^{**}P \leq 0.01$, $^{***}P \leq 0.001$.

CES. CNO was injected 30 min before each session (Fig. 4h). At the behavioral level, VTA$^{DA}$ neuron inhibition during CES suppressed anxiety-like behavior in the OFT. This result was reflected by a significant increase in central exploration time and central distance compared with the mCherry and saline controls, with no locomotor effects (Fig. 4i–l). Similarly, CNO-mediated VTA$^{DA}$ neuron inhibition increased the time spent in the open arms and the number of open-arm entries in the EPM (Fig. 4m–o). While CNO injection did not alter the time spent in the social interaction zone relative to the mCherry and saline controls in SIT tests (Fig. 4p, q). These findings support the role of VTA$^{DA}$ neurons in the anxiety-like behavior of CES.

Furthermore, we investigated whether one-time acute VTA dopamine inhibition prior to behavioral tests can also block CES-induced anxiogenic effects. AAV2/9-DIO-hM4Di-mCherry and AAV2/9-TH-Cre were bilaterally injected into the VTA of C57BL/6 mice (Fig. 5a, b). All mice received 10-day CES treatment and a battery of behavior tests following the one-time administration of CNO or saline on day 11. At the behavioral level, we found that acute VTA dopamine inhibition prior to OFT and EPM can also significantly block the anxiogenic effects of CES (Fig. 5c–i). Similarly to the results of Fig. 4p, q, we also found that one-time acute VTA dopamine inhibition prior to SIT did not alter the time spent in the social interaction zone after 10-day CES (Fig. 5j, k). Collectively, these data suggest that inhibiting the activity of VTA$^{DA}$ neurons arrests CES-induced anxiety-like behavior.

**VTA$^{DA}$ neurons receive direct, monosynaptic GABA inputs from the NAc.** After confirming the sufficient and necessary role of VTA$^{DA}$ neuron activation in CES-induced anxiety-like behavior, we further clarified the upstream circuit of VTA$^{DA}$ neurons[19]. Cre-dependent helper viruses (AAV2/9-DIO-TVA-EGFP and AAV2/9-DIO-RVG) and AAV2/9-TH-Cre were injected into the VTA. After 10 days of CES exposure, rabies virus (RV-ENVA-ΔG-DsRed) was injected into the same site (Fig. 6a–d; Supplementary Fig. 9a). The helper viruses enabled retrograde transport of the rabies virus across the monosynapse. Then, whole-brain quantification of DsRed-labeled upstream-projecting neurons of VTA$^{DA}$ and percent of total cells in a given projecting brain area was calculated. The results revealed that

many brain areas have projections to VTA$^{DA}$ neurons, but among them, VTA$^{DA}$ neurons receive fewer inputs from the NAc area in CES mice compared with controls (Fig. 6e), suggesting that the structural plasticity of axons of direct inputs from NAc to VTA$^{DA}$ neurons was changed after CES. The principal projection neurons in NAc are all GABAergic[20]. Using retrograde tracing techniques and immunofluorescence staining, we also showed that NAc GABAergic neurons innervate VTA$^{DA}$ neurons (Fig. 6f–h). Similarly, employing the same strategy in DAT-Cre mice, we also found that the VTA–projecting NAc neurons are GABAergic (Supplementary Fig. 10a–c). We reconfirm the NAc–VTA circuit by anterograde tracing from NAc$^{GABA}$ neurons. NAc$^{GABA}$ axons and presynaptic terminals were labeled by infection of AAV2/9-DIO-mCherry and AAV2/9-GAD67-EGFP-2A-Cre into the NAc area of C57BL/6 mice (Fig. 6i, j). The result indicated that NAc$^{GABA}$ neurons have long-range projections that densely innervate the VTA (Fig. 6k, l).

We next assessed the function of the NAc–VTA circuit in vivo. C57BL/6 mice were infused with AAV2/9-DIO-GCaMP6s into the VTA, and AAV2/1-TH-Cre into the NAc, which could anterograde transport from the neuronal cell body to the axon terminal and then across the monosynapse into its projecting neurons[21,22]. Optic-fiber cannula was implanted over the VTA (Fig. 6m). After a week's rest, 10-day CES modeling was conducted, followed by the EPM test and simultaneous VTA$^{DA}$ neuronal activity detection using position-synchronized in vivo calcium imaging (Fig. 6n). Through fluorescence verification, AAV2/1-TH-Cre virus could cross the NAc–VTA synapse and help the expression of GCaMP6s in VTA$^{DA}$ neurons (Fig. 6o; Supplementary Fig. 11a, b). Similar to the previous results, CES mice again showed decreased exploring time in the open arms and increased time in the closed arms of EPM (Fig. 6p, q). Meanwhile, the position-synchronized in vivo calcium recording showed that, compared with control mice, CES mice exhibited a significantly higher average Ca$^{2+}$ activities and higher variation rate of Ca$^{2+}$ fluorescence in the VTA$^{DA}$ neurons receiving NAc projections during open-arm exploration, but not closed-arm exploration (Fig. 6r–u). These results indicate that NAc–VTA circuit exhibits robust activation during exposure to an innate anxiogenic environment and functionally mediates anxiety-like behavior of CES.

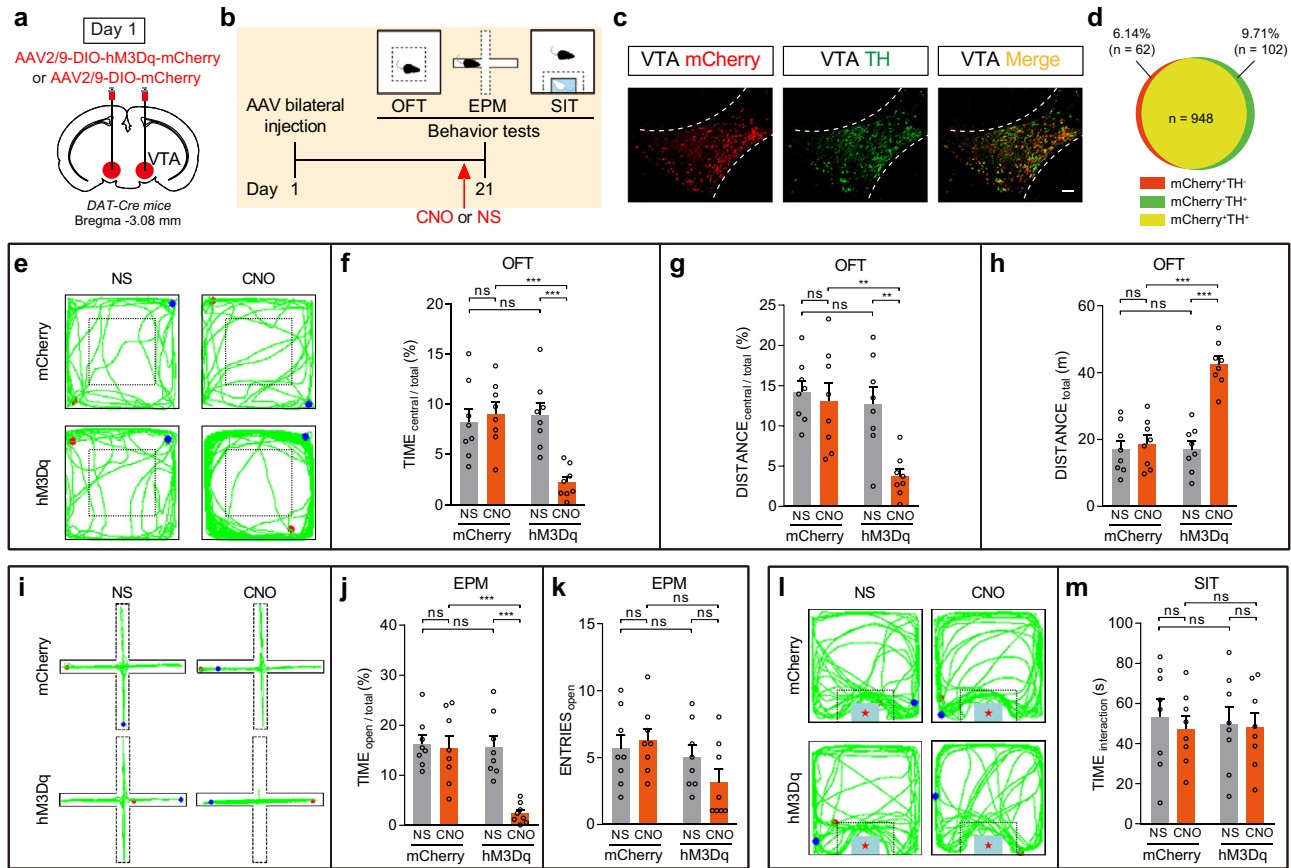

**Fig. 3 Chemogenetic activation of VTA^DA neurons in DAT-Cre mice directly triggers anxiety-like behavior. a** Schematic of bilateral virus infection in DAT-Cre mice. **b** Experimental scheme of chemogenetic activation of VTA^DA neurons and behavioral tests. **c** Representative image of VTA-injection sites. Scale bar, 100 μm. **d** The quantitative analysis of Venn diagram shows the co-expression level of mCherry with TH in VTA (3 sections per mouse from 3 mice). **e–h** Results of OFT. **e** Representative exploration traces of OFT. **f** Percentage of time spent in the central area of OFT (two-way ANOVA, group: $P = 0.0100$, treatment: $P = 0.0125$, interaction: $P = 0.0019$; post hoc Tukey's test, hM3Dq(NS vs. CNO): $P = 0.0010$, CNO(mCherry vs. hM3Dq): $P = 0.0008$). **g** Percentage of distance traveled in the central area of OFT (two-way ANOVA, group: $P = 0.0038$, treatment: $P = 0.0068$, interaction: $P = 0.0303$; post hoc Tukey's test, hM3Dq(NS vs. CNO): $P = 0.0052$, CNO(mCherry vs. hM3Dq): $P = 0.0034$). **h** Total distance traveled in the OFT (two-way ANOVA, group: $P = 4.97 \times 10^{-5}$, treatment: $P = 9.13 \times 10^{-6}$, interaction: $P = 5.60 \times 10^{-5}$; post hoc Tukey's test, hM3Dq(NS vs. CNO): $P = 4.78 \times 10^{-7}$, CNO(mCherry vs. hM3Dq): $P = 1.49 \times 10^{-6}$). **i–k** Results of EPM. **i** Representative exploration traces in EPM. **j** Percentage of time spent in the open arms of EPM (two-way ANOVA, group: $P = 0.0014$, treatment: $P = 0.0009$, interaction: $P = 0.0032$; post hoc Tukey's test, hM3Dq(NS vs. CNO): $P = 0.0002$, CNO(mCherry vs. hM3Dq): $P = 0.0003$). **k** Number of open-arm entries in the EPM (two-way ANOVA, group: $P = 0.0561$, treatment: $P = 0.5119$, interaction: $P = 0.1947$; post hoc Tukey's test, hM3Dq(NS vs. CNO): $P = 0.5042$, CNO(mCherry vs. hM3Dq): $P = 0.1111$). **l, m** Results of SIT. **l** Representative traces in SIT. **m** Statistics of time spent in the social interaction zone of SIT (two-way ANOVA, group: $P = 0.8647$, treatment: $P = 0.6315$, interaction: $P = 0.7857$; post hoc Tukey's test, hM3Dq(NS vs. CNO): $P = 0.9988$, CNO(mCherry vs. hM3Dq): $P = 0.9999$). The red and blue dots in the OFT, EPM, and SIT locomotion traces reflect the start and end points of the mouse, respectively. $n = 8$ mice for each group. All data are shown as mean ± SEM. ns not significant, **$P \leq 0.01$, ***$P \leq 0.001$.

**Bidirectional modulation of NAc–VTA circuit could mimic or reverse the CES-induced anxiety-like behavior.** Given that VTA^DA neuronal activity is increased during CES-induced anxiety-like behavior, and that the NAc primarily comprises GABAergic neurons, we hypothesized that inhibiting NAc GABAergic axons in the VTA would be sufficient to increase anxiety-like behaviors. C57BL/6 mice were bilaterally infused with AAV2/9-GAD67-EGFP-2A-Cre and AAV2/9-DIO-NpHR3.0-mCherry (NpHR group), using AAV2/9-DIO-mCherry as control (mCherry group), into the NAc. Optical fibers were implanted above the VTA, and the circuit was optically inhibited with yellow light (Fig. 7a, b). Post hoc imaging showed that mCherry and EGFP expression was restricted to NAc neurons (Fig. 7c, d). In the OFT, the NpHR mice exhibited decreased central exploration time and central distance during the laser On epoch, indicating the increment of an anxiety-like

phenotype (Fig. 7e–g), but not disrupt locomotor activity in the NpHR group, as measured by total distance (Fig. 7h). In the EPM, NpHR mice showed significantly reduced open-arm exploration during the On epoch (Fig. 7i, j), but no effect on open arm entries during the On epoch compared with the Off epoch (Fig. 7k).

To test the effects of activation of NAc–VTA circuit, mice were bilaterally injected with AAV2/9-GAD67-EGFP-2A-Cre and AAV2/9-DIO-ChR2-mCherry (ChR2) using AAV2/9-DIO-mCherry as control, into the NAc. Optic fibers were implanted into the VTA (Fig. 7l), and the circuit was optically activated with blue light each day during the CES paradigm over 10 days of modeling (Fig. 7m). In the OFT, optically stimulated mice increased central exploration time and central distance without changes in total locomotion (Fig. 7n–q). In the EPM, optically stimulated mice increased the time spent in the open arms and the number of open-arm entries (Fig. 7r–t). These results indicate

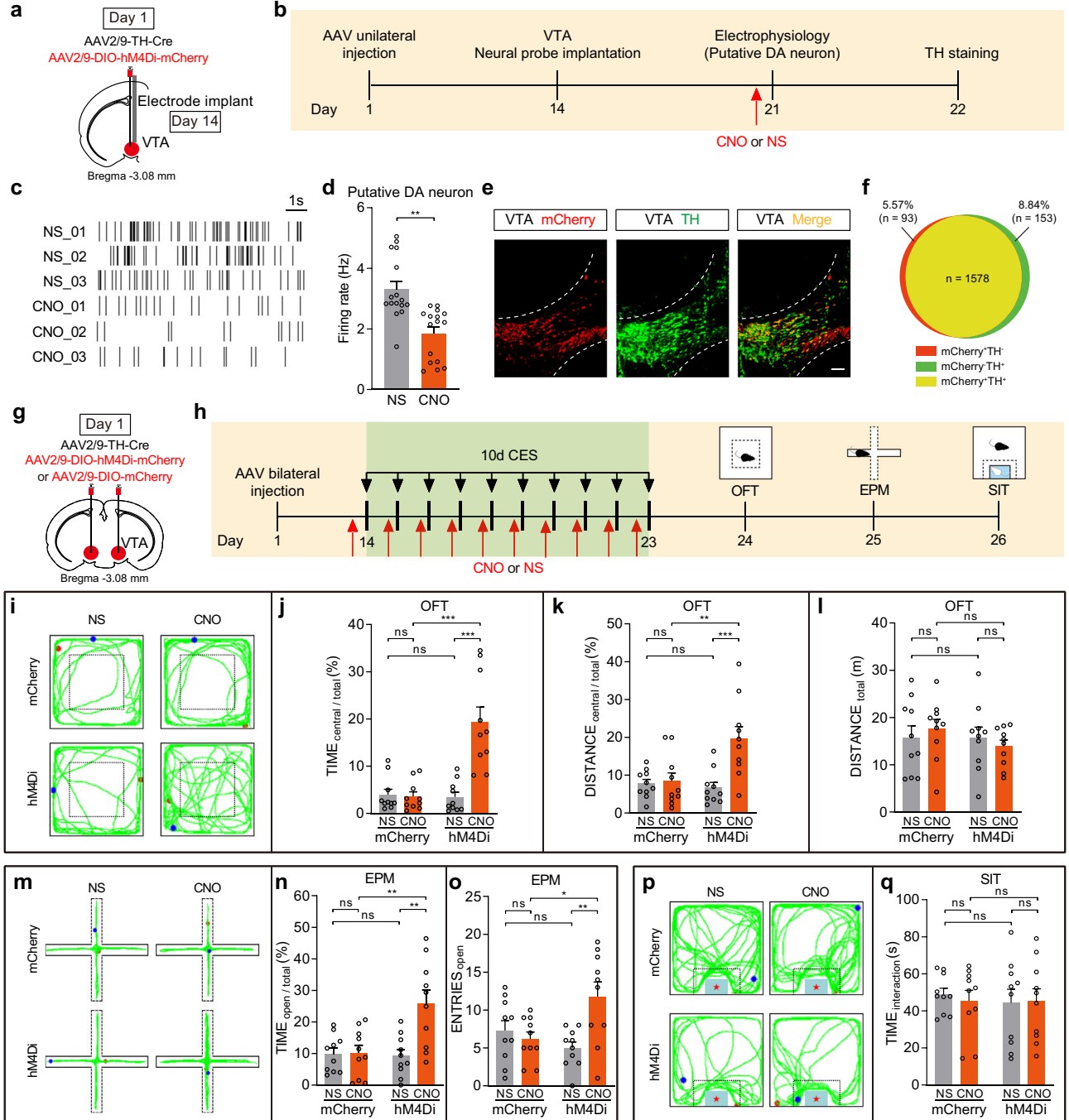

that activation of NAc–VTA circuit could efficiently alleviate the CES-induced anxiety-like behaviors.

Due to backpropagating action potentials to the somata of the NAc neurons when using the above approach of optogenetic stimulation, many other unwanted terminals of NAc neurons might be involved in anxiety. In order to eliminate this defect and make our conclusion more convinced, C57BL/6 mice were infused with AAV2/1-TH-Cre into the NAc, and AAV2/9-DIO-NpHR-mCherry into the VTA. Optic-fiber cannula was implanted over the VTA, and the circuit was optically inhibited with yellow light (Fig. 8a, b). The results showed that inhibiting the neural projection from the NAc to the VTA[DA] neurons triggered anxiety-like behavior in the OFT (Fig. 8c–f) and EPM (Fig. 8g–i). Going further, we use the same strategy of virus injection. C57BL/6 mice were infused with AAV2/1-TH-Cre into the NAc, and AAV2/9-DIO-ChR2-mCherry into the VTA. Optic

fibers were implanted into the VTA of mice, and the circuit was optically activated with blue light each day during the CES paradigm over 10 days of modeling (Fig. 8j, k). The results of OFT and EPM also showed that activation of neural projection from the NAc to the VTA[DA] neurons could reverse the CES-induced anxiety-like behavior (Fig. 8l–r).

## Discussion

Using the redesigned 3C-VSDS model in mice, we revealed that CES induced anxiety-like symptoms and transient social dysfunction but not depression-like phenotypes in mice. We targeted DA neurons of VTA as a potential mechanism of this behavioral deficit. VTA[DA] neuron hyperactivity induced by CES is involved in the anxiety-like behavior in the innate anxiogenic environment. Chemogenetic activation of VTA[DA] neurons directly

**Fig. 4 Chemogenetic inhibition of VTA$^{DA}$ neurons promotes resilience to the CES-induced anxiety-like behavior. a–f** Verify the experimental validity of chemogenetic inhibition and in vivo electrophysiology recordings of VTA$^{DA}$ neurons. **a** Schematic of virus infection and electrode implantation in VTA. **b** Experimental scheme of chemogenetic inhibition and in vivo electrophysiology recordings of VTA$^{DA}$ neurons. **c** Raster plot of an hM4Di-mCherry mouse after normal saline (NS) or CNO injection. **d** Firing rate of VTA$^{DA}$ neurons following NS or CNO administration (two-tailed paired $t$-test, $P = 0.0038$), $n = 16$ neurons from 3 mice. **e** Representative image of VTA injection sites. Scale bar, 100 μm. **f** The quantitative analysis of Venn diagram shows the co-expression level of mCherry with TH in VTA (3 sections per mouse from 5 mice). **g–q** Bilateral chemogenetic inhibition of VTA$^{DA}$ neurons and behavioral tests ($n = 10$ mice for each group). **g** Schematic of bilateral virus infection. **h** Experimental scheme of CES modeling and chemogenetic inhibition of VTA$^{DA}$ neurons and behavioral tests. **i–l** Results of OFT. **i** Representative exploration traces of OFT. **j** Percentage of time spent in the central area of OFT (two-way ANOVA, group: $P = 0.0002$, treatment: $P = 0.0001$, interaction: $P = 7.80 \times 10^{-5}$; post hoc Tukey's test, hM4Di(NS vs. CNO): $P = 2.46 \times 10^{-6}$, CNO(mCherry vs. hM4Di): $P = 2.90 \times 10^{-6}$). **k** Percentage of distance traveled in the central area of OFT (two-way ANOVA, group: $P = 0.0250$, treatment: $P = 0.0033$, interaction: $P = 0.0075$; post hoc Tukey's test, hM4Di(NS vs. CNO): $P = 0.0009$, CNO(mCherry vs. hM4Di): $P = 0.0043$). **l** Total distance traveled in the OFT (two-way ANOVA, group: $P = 0.3714$, treatment: $P = 0.9985$, interaction: $P = 0.3757$; post hoc Tukey's test, hM4Di(NS vs. CNO): $P = 0.9207$, CNO(mCherry vs. hM4Di): $P = 0.5848$). **m–o** Results of EPM. **m** Representative exploration traces in EPM. **n** Percentage of time spent in the open arms of EPM (two-way ANOVA, group: $P = 0.0127$, treatment: $P = 0.0052$, interaction: $P = 0.0070$; post hoc Tukey's test, hM4Di(NS vs. CNO): $P = 0.0011$, CNO(mCherry vs. hM4Di): $P = 0.0023$). **o** Number of open-arm entries in the EPM (two-way ANOVA, group: $P = 0.2143$, treatment: $P = 0.0356$, interaction: $P = 0.0046$; post hoc Tukey's test, hM4Di(NS vs. CNO): $P = 0.0040$, CNO(mCherry vs. hM4Di): $P = 0.0221$). **p, q** Results of SIT. **p** Representative traces in SIT. **q** Statistics of time spent in the social interaction zone of SIT (two-way ANOVA, group: $P = 0.6860$, treatment: $P = 0.8187$, interaction: $P = 0.7186$; post hoc Tukey's test, hM4Di(NS vs. CNO): $P = 0.9997$, CNO(mCherry vs. hM4Di): $P = 0.9999$). The red and blue dots in the OFT, EPM, and SIT locomotion traces reflect the start and end points of the mouse, respectively. All data are shown as mean ± SEM. ns not significant, $^{*}P \leq 0.05$, $^{**}P \leq 0.01$, $^{***}P \leq 0.001$.

triggers anxiety-like behavior, while chemogenetic inhibition of these neurons promotes resilience to the CES-induced anxiety-like behavior. Specifically, the NAc–VTA circuit mediates this behavior. Together, we propose that a NAc–VTA circuit critically establishes and regulates the CES-induced anxiety-like behavior.

In the previous VSDS model, a two-chamber cage was used. After emotional stress exposure, the CES mouse was moved to a new cage, and the CSDS mouse was moved across the divider to the left chamber and then remained overnight adjacent to the CD1 mouse[14,15]. During the 10-day VSDS modeling, CES mice were transferred many times, and the environment was frequently changed every day, which did not effectively isolate the physical stress and environmental changes on mice. Compared with the previous VSDS model, our 3C-VSDS model simplifies the procedure, and most importantly, effectively isolates the physical stress and focuses on the impact of emotional stress alone.

The 3C-VSDS model is based on observational learning, which is the capacity to acquire behaviors by witnessing a relevant experience in another animal[23]. Animals possess an affective sensitivity to another's state of distress via emotional contagion and show emotion-related behaviors[24,25]. The ability to learn by observing others' experiences and extracting predictive information about potential threats is critical to evolutionary fitness[26]. It has been proven that visual cues play a central role in the perception of vicarious social stress, and auditory and olfactory stresses are not enough to cause the change of emotion in this model[27–29]. Therefore, 3C-VSDS model is a useful behavioral paradigm to assess affective behaviors through direct observation. The observer mice in this behavior paradigm have apparently identified the stressful nature of what is happening next door. As a result, the observer mouse experienced emotional stress leading to anxiety. We believe that there are two main reasons for the anxiety. One important cause is that the observer mouse is experiencing stress through sharing the affective state of the CSDS mouse. This refers to emotional contagion—the transfer of emotions between individuals—is thought to occur automatically. The other vital cause is that the observer mouse is experiencing social stress observing the violence of the aggressive CD1 mouse attacking the conspecific mice. This observer mouse could be stressed by sounds and smells of CD1 mouse, or internal state changes related to self-preservation. These two causes work together in the observed mice and the extent of their respective roles remains to be studied.

Interestingly, we found that the CES mice initially decreased and increased social interaction before returning to baseline, unlike persistent decreased interaction in CSDS mice. This process may relate to habituation[30], which is defined as a progressive decrement in response to a repeated stimulus[31]. Animals encounter an array of diverse environmental stimuli. While novel stimuli usually increase exploratory behavior, animals may become less interested and responsive over chronic repeated exposure to the same event. Habituation and novelty recognition allow animals to focus attention on the unknown, promote exploratory behavior, and facilitate learning and adaptive responses to changes in the environment[32,33]. Disturbed habituation is a salient feature of many neuropsychiatric disorders[34,35]. Therefore, we believe that the emotional stress of social interaction is initially extreme in CES mice but gradually normalizes due to habituation.

Pioneering studies have implicated VTA$^{DA}$ neurons in social behavior, as well as the processing of positively and negatively salient emotional stimuli[36,37]. Therefore, we considered VTA$^{DA}$ neurons as a potential neural substrate of social interaction. However, the results showed that VTA$^{DA}$ neurons are not required for social interaction behavior. DA neurons have also been researched extensively for their role in motivation, reward, and aversive processing. Unrewarding novel stimuli increase VTA$^{DA}$ neuron activity, and this response habituates as the stimulus becomes familiar[38,39]. Therefore, it has been proposed that novelty, in itself, may be rewarding. It has been demonstrated that unfamiliar conspecifics or unfamiliar objects can enhance $Ca^{2+}$ activity in VTA$^{DA}$ neurons, which is necessary to promote affiliative social behaviors, but not object exploration[40]. This is consistent with the current findings that VTA neuron activation and inactivation did not alter behavior in the SIT. VTA$^{DA}$ neurons consist of several DA neuron subpopulations[41]. Importantly, those DA subpopulations display distinct anatomical, electrophysiological, and molecular properties that mediate different behavioral responses to specific stimuli[42]. It was reported that VTA DA terminals in ventral NAc medial shell are excited by aversive stimuli, while VTA DA terminals in all other NAc subregions are inhibited by aversive stimuli[43]. These findings may explain a number of previously conflicting observations, such as a role for DA in processing rewarding and aversive events, and that aversive stimuli can excite or inhibit VTA$^{DA}$ neurons[44,45]. Combined with the present SIT results, we propose that one

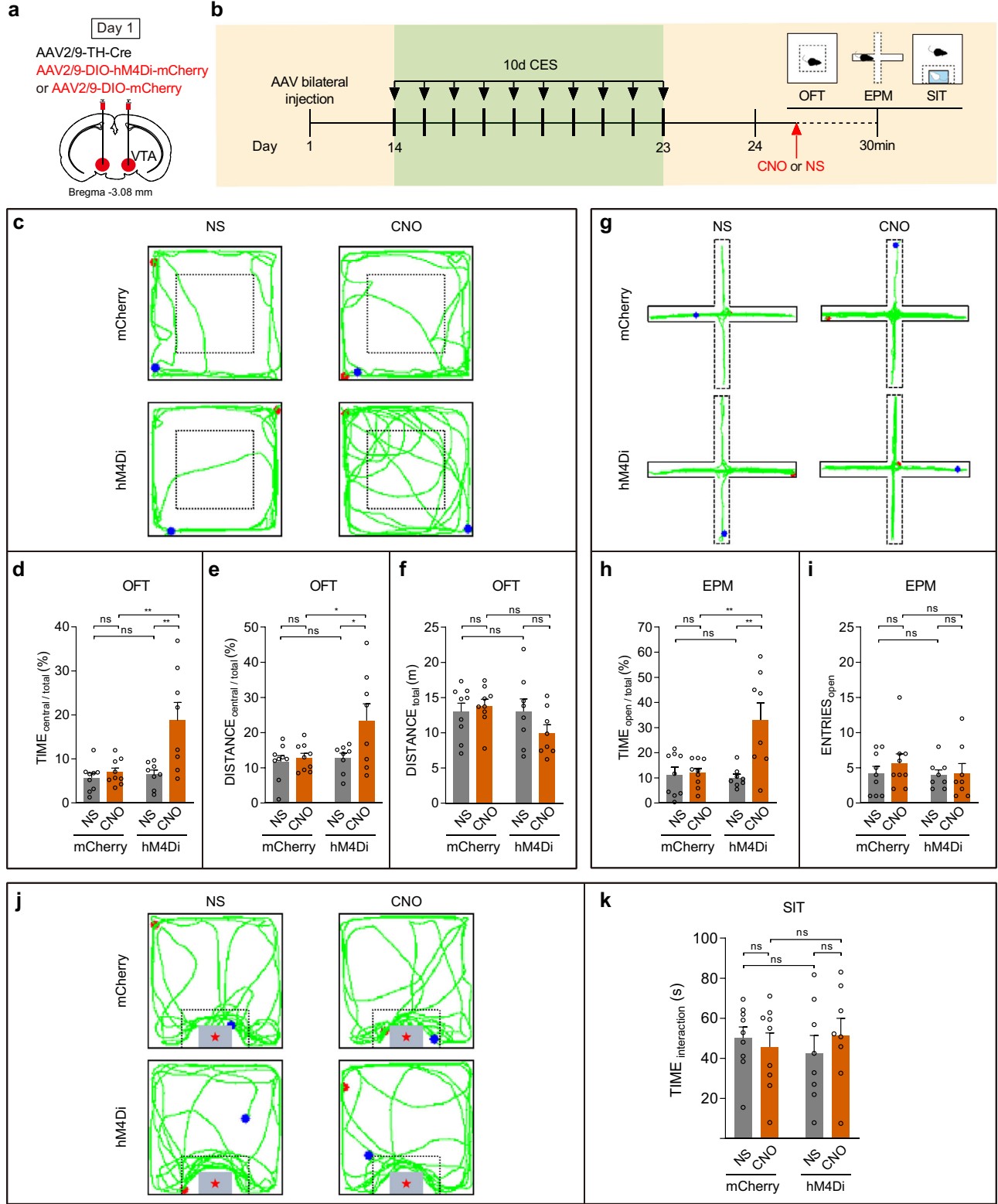

VTA$^{DA}$ neuron subpopulation may promote social interaction, while another may inhibit it. The balance of those two subpopulations would promote normal social interaction.

In addition to VTA, we also found elevated c-fos protein expression in the red nucleus following SES. The red nucleus might represent a locus of coordinating oromotor, respiratory, locomotor, and antinociceptive responses to hypoxia stimuli, and

might be involved in modulation of the respiratory output during the defense reaction[46]. Certainly, it would be interesting to assess the effects of red nucleus in emotional stress-related abnormal behavior.

Because of the specific expression of TH in dopaminergic neurons, it is widely used as molecular markers of dopaminergic neurons[47–50]. Using the TH antibody to do immunofluorescence

**Fig. 5 Acute chemogenetic inhibition of VTA[DA] neurons can block anxiogenic effects of CES. a** Schematic of bilateral virus infection. **b** Experimental scheme of virus injection, CES modeling, and behavioral tests. **c–f** Results of OFT. **c** Representative exploration traces of OFT. **d** Percentage of time spent in the central area of OFT (two-way ANOVA, group: $P = 0.0060$, treatment: $P = 0.0033$, interaction: $P = 0.0141$; post hoc Tukey's test, hM4Di(NS vs. CNO): $P = 0.0022$, CNO(mCherry vs. hM4Di): $P = 0.0025$). **e** Percentage of distance traveled in the central area of OFT (two-way ANOVA, group: $P = 0.0351$, treatment: $P = 0.0326$, interaction: $P = 0.0786$; post hoc Tukey's test, hM4Di(NS vs. CNO): $P = 0.0425$, CNO(mCherry vs. hM4Di): $P = 0.0373$). **f** Total distance traveled in the OFT (two-way ANOVA, group: $P = 0.1474$, treatment: $P = 0.3982$, interaction: $P = 0.1413$; post hoc Tukey's test, hM4Di(NS vs. CNO): $P = 0.3792$, CNO(mCherry vs. hM4Di): $P = 0.1699$). **g–i** Results of EPM. **g** Representative exploration traces in EPM. **h** Percentage of time spent in the open arms of EPM (two-way ANOVA, group: $P = 0.0126$, treatment: $P = 0.0033$, interaction: $P = 0.0053$; post hoc Tukey's test, hM4Di(NS vs. CNO): $P = 0.0010$, CNO(mCherry vs. hM4Di): $P = 0.0020$). **i** Number of open-arm entries in the EPM (two-way ANOVA, group: $P = 0.4757$, treatment: $P = 0.4610$, interaction: $P = 0.6025$; post hoc Tukey's test, hM4Di(NS vs. CNO): $P = 0.9987$, CNO(mCherry vs. hM4Di): $P = 0.8136$). **j, k** Results of SIT. **j** Representative traces in SIT. **k** Time spent in social interaction zone of SIT (two-way ANOVA, group: $P = 0.9095$, treatment: $P = 0.7632$, interaction: $P = 0.3787$; post hoc Tukey's test, hM4Di(NS vs. CNO): $P = 0.8431$, CNO(mCherry vs. hM4Di): $P = 0.9456$). $n = 9$ for mCherry-NS or mCherry-CNO group, n = 8 for hM4Di-NS or hM4Di-CNO group. The red and blue dots in the OFT and EPM locomotion traces reflect the start and end points of the mouse, respectively. All data are shown as mean ± SEM. ns not significant, $^*P \le 0.05$, $^{**}P \le 0.01$.

double staining, we showed that the infection efficiency and the specificity of TH-Cre virus used in our research both reached over 90%, suggesting that we chose the TH-Cre virus with both high-expression efficiency and high specificity to faithfully limit the expression to DA neurons. However, it should be pointed out that, due to the technical limitations of promoter-dependent strategies, such as promoter-independent gene expression, it is inevitable that there is partial expression in a small number of nondopamine neurons in TH-Cre or DAT-Cre strategies.

Our results indicate that the activation of VTA[DA] neurons is involved in the anxiety-like behavior induced by CES in the innate anxiogenic environment. A previous study using freely moving calcium imaging, it is shown that compared with the closed-arm compartment, lots of ventral CA1 (vCA1) neurons exhibited a significant increase in $Ca^{2+}$ activity and rate of $Ca^{2+}$ transients during exploration of the anxiogenic open-arm compartment[51]. The potential link between the VTA and CA1 during exploration of innately anxiogenic environments warrants further experiments in further research.

It was reported that disinhibition of VTA[DA] neurons, by inhibiting midbrain GABAergic neuronal activity, induced locomotor hyperactivity in mice[52]. Also, we displayed that chemogenetic activation of VTA[DA] neurons showed increment on locomotor activities, which was consistent with numerous published studies[53,54].

The VTA plays a key role in emotional processing. Research shows that the inhibitory projections from the dorsolateral bed nucleus of the stria terminalis (dlBNST) to the VTA are enhanced in animal models of depression, thereby suppressing the mesolimbic dopaminergic system[55]. Another major downstream target of the VTA is the NAc. The NAc is a vital component in the reward circuitry, which responds to stress signals and has a dominant effect on emotion regulation[56,57]. Dopamine 1 (D1)–medium spiny neurons (MSNs) in the NAc play roles in modulating reward-related responses[58], whereas D2–MSNs regulate anxiety-like aversion or avoidance behavior[59]. It is reported that GABAergic somatostatin projection from the BNST to NAc controls anxiety[60]. Here, we demonstrated that NAc neurons have long-range projections that densely innervate the VTA[DA] neurons, and the NAc–VTA circuit mediates the anxiety-like behavior of CES.

Using optogenetic methods, we found that bidirectional modulation of NAc terminals in the VTA could mimic or reverse the CES-induced anxiety-like behavior. Due to backpropagating action potentials to the somata of the NAc neurons, many other unwanted terminals of NAc neurons might be involved in anxiety. It was reported that the application of anterograde transsynaptic spread viruses is useful for tracing and manipulating neural circuits in a postsynaptic cell-type- and input-specific

manner[21]. In order to eliminate this defect and make our conclusion more convinced, we injected anterograde transsynaptic virus (AAV2/1-TH-Cre) into the NAc, and AAV2/9-DIO-NpHR(or ChR2)-mCherry into the VTA. Optic-fiber cannula was implanted over the VTA. In concert with the previous studies, the results indicate that inhibition or activation of neural projection from NAc to VTA[DA] neurons could trigger or reverse the CES-induced anxiety-like behavior. These findings further support the specific role of NAc–VTA circuit in the anxiety-like behavior of CES.

It is widely accepted that VTA[DA] neurons play an essential role in reward. Photoactivation of VTA[DA] neurons induces a strong conditioned place preference[61]. Conversely, using a genetic knockout model to impair phasic firing, while leaving tonic activity largely intact, results in selective impairment of specific forms of reward learning[62]. DA neurons are strongly excited when a reward is larger than predicted, whereas they are inhibited when an award is smaller than predicted[63]. It has been demonstrated that VTA–NAc DA projections play a central role in reward processing[64]. Phasic activation of VTA[DA] neurons leads to transient DA release, especially in NAc[61]. The "tracing the relationship between input and output" method has shown that, generally, neurons projecting to diverse output regions receive similar inputs[65]. Another, chronic emotional stress administration that induces structure plasticity of synapses in NAc neurons is largely unknown. Recently, extensive researches have been reported that rabies virus-based monosynaptic tracing facilitates screening of circuit elements that contribute to behavioral changes[66,67]. Here, we show that the projection neurons from NAc to VTA[DA] neurons are most GABAergic and the inhibitory effect of NAc on VTA[DA] neurons was weakened after 10-day CES. However, with a small sample size, caution must be applied, as the findings about the CES triggering structure plasticity of direct inputs to VTA[DA] neurons might need further verification. Given the diverse functions of projections between the VTA and the NAc, we propose a hypothetical model of the relationship between reward and anxiety. Reward stimuli elicit immediate VTA[DA] neuron activation, increasing DA release in the NAc. This rapidly activates GABAergic neurons in the NAc, which directly synapse onto DA neurons in the VTA. This preferentially activates $GABA_B$ receptors[20], which can inhibit VTA[DA] neurons' activation to relieve anxiety. Sequential activation of VTA[DA] neurons and NAc[GABA] neurons might underlie the reward's ability to relieve anxiety. However, detailed mechanisms of this phenomenon need to be studied further.

In summary, our study demonstrates that the NAc–VTA circuit mediates the anxiety-like behavior of CES. This study not only characterizes a preclinical model representing the nuanced aspect of CES, but also provides insight into the circuit-level neuronal processes underlying empathy-like behavior.

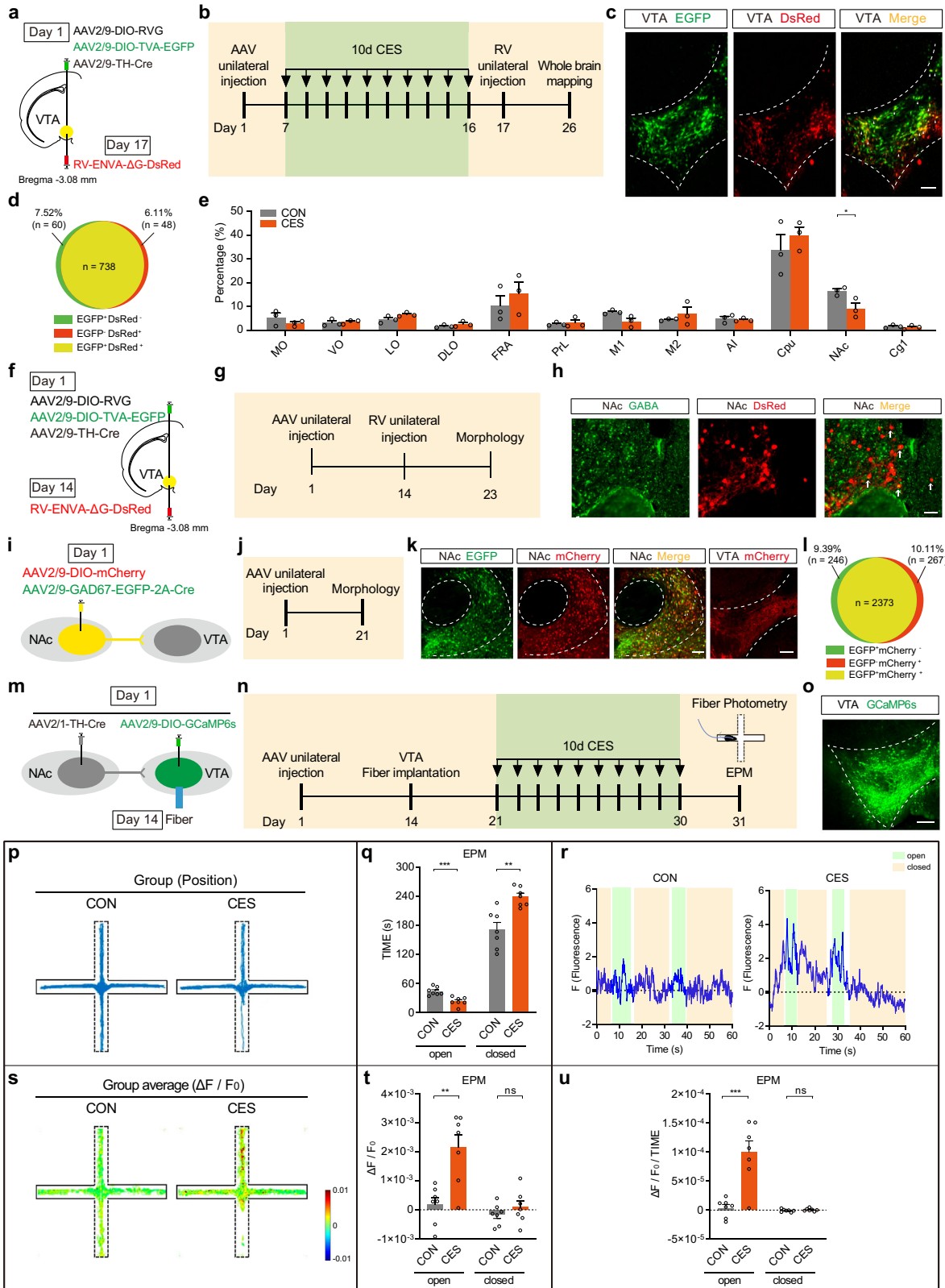

## Methods

**Animals**. C57BL/6 J mice (6–8 weeks, male) and CD1 mice (30–35 weeks, male) were purchased from Beijing Vital River Laboratory Animal Technology Co., Ltd. Adult male DAT-Cre mice (No. NM-KI-200092, 8–10 weeks) were purchased from Shanghai Model Organisms Center, Inc. All mice were housed under stable temperature (22–25°C) and consistent humidity (50 ± 5%) in a 12-hour light/dark cycle. Mice were group-housed for 1 week prior to the experiment and allowed to acclimate to the

behavioral testing room for at least 1 hour before testing. All procedures were approved by the Animal Care Committee at Huazhong University of Science and Technology and performed in accordance with the Institutional Animal Welfare Guidelines.

**Virus preparations**. We used the following virus vectors carrying the following specific genes purchased from BrainVTA (Wuhan): AAV2/9-hSyn-DIO-GCaMP6s-WPRE-pA, AAV2/9-TH-NLS-Cre-WPRE-pA, AAV2/9-Ef1α-DIO-

**Fig. 6 VTA$^{DA}$ neurons receive direct, monosynaptic inputs from the NAc GABAergic neurons. a–e** Whole-brain retrograde tracing of VTA$^{DA}$ neurons. **a** Schematic of the Cre-dependent retrograde trans-monosynaptic rabies virus tracing strategy. **b** Experimental scheme of retrograde tracing and CES modeling. **c** Starter cells (yellow) in VTA, which co-infected by AAV2/9-TH-Cre, AAV2/9-DIO-RVG, AAV2/9-DIO-TVA-EGFP (green), and RV-ENVA-ΔG-DsRed (red). Scale bar, 100 μm. **d** The quantitative analysis of Venn diagram shows the co-expression level of EGFP with DsRed in VTA (3 sections per mouse from 3 mice). **e** Whole-brain quantification of VTA$^{DA}$ neuron inputs; percent of total cells in a given brain area relative to the total number of brain-wide inputs (two-way ANOVA, group: $P = 0.7916$, region: $P = 1.00 \times 10^{-15}$, interaction: $P = 0.2266$; uncorrected Fisher's LSD, medial orbital cortex (MO): $P = 0.4351$, ventral orbital cortex (VO): $P = 0.9005$, lateral orbital cortex (LO): $P = 0.5135$, dorsolateral orbital cortex (DLO): $P = 0.7896$, frontal association cortex (FrA): $P = 0.1153$, prelimbic cortex (PrL): $P = 0.8681$, primary motor cortex (M1): $P = 0.2274$, secondary motor cortex (M2): $P = 0.4620$, agranular insular cortex (AI): $P = 0.8618$, caudate putamen (striatum) (CPu): $P = 0.0569$, nucleus accumbens (NAc): $P = 0.0256$, cingulate cortex area 1 (Cg1): $P = 0.9153$), $n = 3$ mice for each group. **f–h** NAc GABAergic neurons innervate VTA$^{DA}$ neurons. **f** Schematic of the Cre-dependent retrograde trans-monosynaptic rabies virus tracing strategy. **g** Experimental scheme of retrograde tracing. **h** DsRed-labeled neurons in the NAc traced from the VTA$^{DA}$ neurons and DsRed signals were colocalized with GABAergic neuronal marker GABA immunofluorescence in the NAc (green, GABA; red, DsRed). The experiment was repeated three times with similar results. Scale bars, 50 μm. **i–l** Reconfirm the NAc–VTA circuit by anterograde tracing. **i** Schematic of virus infection of anterograde tracing. **j** Experimental scheme of anterograde tracing. **k** Representative images of viral expression within the NAc and mCherry expression in the VTA. Scale bar, 100 μm. **l** The quantitative analysis of Venn diagram shows the co-expression level of EGFP with mCherry in NAc (3 sections per mouse from 5 mice). **m–u** VTA$^{DA}$ neurons receiving NAc projections are significantly activated in CES mice ($n = 7$ mice for each group). **m** Schematic of in vivo virus infection and optical fiber implantation. **n** Experimental scheme of CES modeling and position-synchronized in vivo calcium imaging in EPM. **o** Representative image of VTA-injection sites. Scale bar, 100 μm. **p** Group of exploration traces in EPM. **q** Statistics of time spent in the open arm and closed arm of EPM (open arms: two-tailed unpaired $t$-test, $P = 0.0008$; closed arms: two-tailed unpaired $t$-test, $P = 0.0014$). **r** The representative traces of Ca$^{2+}$ fluorescence in CON and CES mouse. The transparent cyan background and papaya whip color display the mice located in the open arms and closed arms, respectively. **s** Group average of Ca$^{2+}$ activities of VTA$^{DA}$ neurons receiving NAc projections in EPM. **t** Statistics of average Ca$^{2+}$ activities of VTA$^{DA}$ neurons receiving NAc projections in the open arm and closed arm of EPM (open arms: two-tailed unpaired $t$-test, $P = 0.0025$; closed arms: two-tailed unpaired $t$-test, $P = 0.3500$). **u** Statistics of variation rate of Ca$^{2+}$ fluorescence in the open arm and closed arm of EPM (open arms: two-tailed unpaired $t$-test, $P = 0.0006$; closed arms: two-tailed unpaired $t$-test, $P = 0.3182$). All data are shown as mean ± SEM. ns not significant, $^*P \leq 0.05$, $^{**}P \leq 0.01$, $^{***}P \leq 0.001$.

hM3D(Gq)-mCherry-WPRE-pA, AAV2/9-Ef1α-DIO-mCherry-WPRE-pA, AAV2/9-Ef1α-DIO-hM4D(Gi)-mCherry-WPREs-pA, AAV2/9-Ef1α-DIO-RVG-WPRE-pA, AAV2/9-Ef1α-DIO-TVA-EGFP-WPRE-pA, RV-ENVA-ΔG-DsRed, AAV2/1-TH-NLS-Cre-WPRE-pA, AAV2/9-Ef1α-DIO-eNpHR3.0-mCherry-WPRE-pA, and AAV2/9-Ef1α-DIO-hChR2(H134R)-mCherry-WPRE-pA. The following vector was from OBiO Technology (Shanghai): AAV2/9-GAD67-EGFP-2A-Cre. The viral titers were in the range of $3–8 \times 10^{12}$ genome copies/ml. Viral vectors were subdivided into aliquots stored at −80 °C until use.

**Three-chamber vicarious social defeat stress model.** The length, width, and height of the customized three-chamber cage are 545 mm, 395 mm, and 200 mm, respectively. In the three-chamber cage, the aggressive retired male CD1 mouse was placed in the middle chamber; two conspecific male C57BL/6 mice were placed in the remaining two chambers. Each mouse (intruder) in the right chamber was introduced into the middle chamber and subjected to physical contact and defeat by the aggressive CD1 mouse (resident). These physically defeated mice were defined as chronic social-defeated stress mice (CSDS mice). Without receiving direct physical stimuli, mouse (observer) in the left chamber was vicariously conditioned for scene-dependent emotion by observing CSDS mouse (demonstrator) that received repetitive social defeats in the middle chamber. These observed mice were defined as chronic emotional stress mice (CES mice). After 5 minutes of physical interaction, the CSDS mouse was separated from the CD1 and released back into the right chamber. The CD1 mice, CES mice, and CSDS mice were housed together, separated by a perforated plastic divider that allowed visual, olfactory, and auditory contacts, but prevented physical interaction, for the remainder of the 24-h period. The same CSDS mouse was subjected to social defeat from a different resident CD1 mouse each day for 10 consecutive days. Without receiving any physical and emotional stimuli, control mice were housed with new CD1 mice every day in the three chambers cage. After the end of CES, each mouse was raised in a single cage.

**Behavioral procedures.** On the test day, mice were transferred to the testing room and acclimated to the room conditions for at least 1 h. After each individual test session, the apparatus was thoroughly cleaned with 70% alcohol to eliminate the previously tested mouse's odor and trace.

*Open-field test.* Mice were always placed at the periphery of an open-field arena (50 cm × 50 cm). The mice were allowed to explore their surroundings freely. The open field was divided into the central area and peripheral area. A video-tracking system (SuperMaze$^+$, Xinruan, Shanghai) was used to measure mice's spontaneous activity. Movements were monitored, and their behavior was statistically analyzed for 5 min.

*Elevated plus maze.* The EPM consists of a central platform, two arms enclosed by light-gray walls, and two open arms (length 350 mm each, elevated 100 mm aboveground). Mice were placed in the central platform, and their behavior was monitored with a camera (SuperMaze$^+$, Xinruan, Shanghai) placed on the top of the maze. The behavior was statistically analyzed for 5 min.

*Social interaction test.* Except the last of SIT was conducted 24 hours after social defeat session (Fig. 1f), the rest of SIT was tested 1 h after social defeat session (Fig. 1d). Other than Fig. 1, SIT took place after OFT and EPM. Test mice (CES mice, CSDS mice, and nondefeated control mice) were subjected to the social interaction test, which was performed in a 50-cm × 50-cm arena equipped with a plastic mesh cage. The approach-avoidance behavior of a test mouse to a CD1 mouse was recorded with a video-tracking system. An unfamiliar CD1 mouse was placed into the plastic mesh cage, and the plastic mesh cage was put into the arena. The plastic mesh cage allowed visual, olfactory, and auditory interactions between the test mouse and the target CD1 mouse, but prevented direct physical contact. Then experimental mice were allowed to explore the open arena freely. A video-tracking software (SuperMaze$^+$, Xinruan, Shanghai) was used to measure the time spent by the experimental mouse in the "interaction zone" (28 cm × 16 cm) of the arena. The total time is 150 s.

*Tail-suspension test.* Mice were suspended by tails with adhesive tape 1 cm from the tip of the tail and roughly 30 cm above the desk, so that no contact could be made. Plastic tubes were placed over mouse tails to ensure mice could not climb or hang on to their tail. The behavior was videotaped from the side, and the animal's immobility time was statistically analyzed for 5 min.

*Sucrose-preference test.* At the start of the experiment, mice were singly housed and given access to two bottles (containing water or 1% sucrose) for 24 h. The water and sucrose bottles' position was switched every 8 h to ensure that the mice did not develop a side preference. After this period, mice were left undisturbed, and their overnight fluid consumption was measured the next morning (9:00 a.m.). Sucrose preference was calculated as a percentage (amount of sucrose consumed (bottle A) × 100/total volume consumed (bottles A + B)).

**Single emotional stress.** In the three-chamber cage, without receiving direct physical stimuli, mouse (observer) in the left chamber was vicariously conditioned for scene-dependent emotion by observing conspecific mouse (demonstrator) that received repetitive social defeats in the middle chamber. These observed mice were defined as single emotional stress mice (SES mice). After 5 minutes of emotional stress, the SES mouse was taken out of the left chamber of three-chamber cage.

**Immunofluorescence staining.** Mice were intracardially perfused with 4% paraformaldehyde in phosphate-buffered saline (PBS). Brains were postfixed at 4 °C overnight, dehydrated in 20%, and 30% sucrose. About 40-μm coronal sections were sliced on a Leica 1860 vibratome. The brain slices were permeabilized with 0.3% Triton X-100 in PBS for 10 min and blocked with 5% bovine serum albumin (BSA) in PBS for 0.5 h. The brain slices were then incubated with the primary antibodies at 4 °C overnight. Primary antibodies used were c-fos (1:1000, CST,

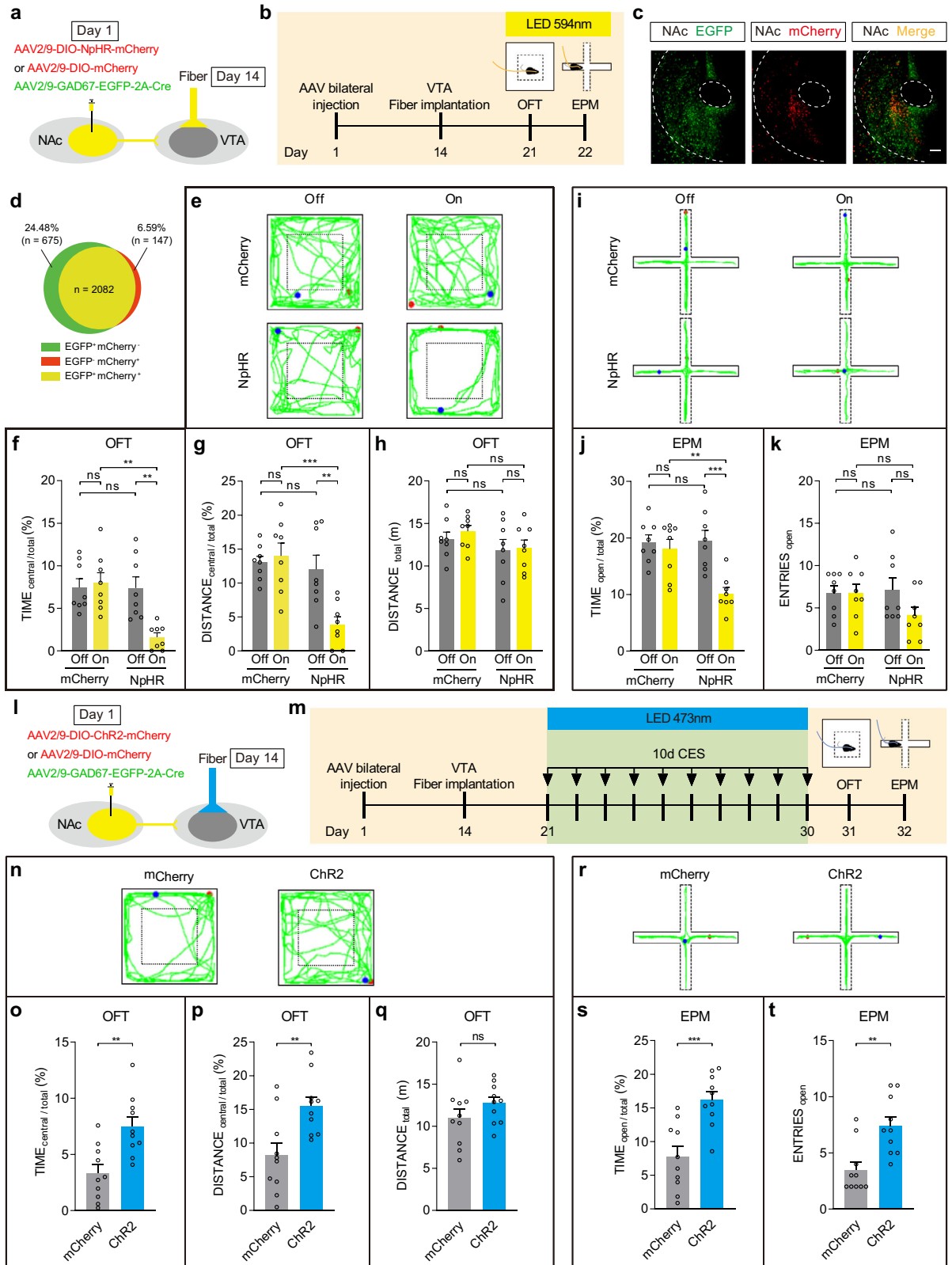

2250 S), TH (1:500, Proteintech, 25859-1-AP), and GABA (1:100, Sigma, A2052). After rinsing three times in PBS for 5 min each, brain slices were incubated with secondary antibodies for 1 h at room temperature in the dark. The secondary antibodies used were goat anti-rabbit antibodies conjugated to Alexa Fluor 488 (Jackson, 111-545-144, 1:500, green stain) or Cy™3 (Jackson, 111-165-144, 1:500, red stain). Nuclei were stained by DAPI. For the negative controls, rabbit IgG was used at the same concentration as the primary antibody. The sections were coverslipped with fluorescent mounting medium and stored at 4 °C before analysis. All images were acquired with an Olympus VS120 microscope.

For automated cell counting of fluorescent protein-positive neurons, the fluorescence images were analyzed and quantitated by ImageJ Fiji plugins and features. Briefly, the red channel and green channel of fluorescence images were separately segmented by Trainable Weka Segmentation plugin first. The Trainable Weka Segmentation is a Fiji plugin that combines a collection of machine-learning

**Fig. 7 Bidirectional modulation of NAc–VTA circuit could mimic or reverse the CES-induced anxiety-like behavior. a–k** Inhibition of NAc–VTA circuit directly triggers anxiety-like behavior. **a** Schematic representation of in vivo virus infection and optical fiber implantation. **b** Experimental scheme. **c** Starter cells (yellow) in NAc, which are co-infected by AAV2/9-GAD67-EGFP-2A-Cre (green) and AAV2/9-DIO-mCherry (red). Scale bar, 200 μm. **d** The quantitative analysis of Venn diagram shows the co-expression level of EGFP with mCherry in NAc (3 sections per mouse from 4 mice). **e–h** Results of OFT ($n = 8$ mice for each group). **e** Representative locomotion traces of NpHR or mCherry mice in OFT with laser on or off. **f** Percentage of time spent in the central area of OFT (two-way ANOVA, group: $P = 0.0049$, laser: $P = 0.0193$, interaction: $P = 0.0058$; post hoc Bonferroni's test, NpHR(off vs. on): $P = 0.0036$, On(mCherry vs. NpHR): $P = 0.0012$). **g** Percentage of distance traveled in the central area of OFT (two-way ANOVA, group: $P = 0.0016$, laser: $P = 0.0322$, interaction: $P = 0.0079$; post hoc Bonferroni's test, NpHR(off vs. on): $P = 0.0070$, On(mCherry vs. NpHR): $P = 0.0007$). **h** Total distance traveled in the OFT (two-way ANOVA, group: $P = 0.0938$, laser: $P = 0.5533$, interaction: $P = 0.7353$; post hoc Bonferroni's test, NpHR (off vs. on): $P = 0.9999$, On(mCherry vs. NpHR): $P = 0.9193$). **i–k** Results of EPM ($n = 8$ mice for each group). **i** Representative exploring traces of NpHR or mCherry mice in EPM with laser on or off. **j** Percentage of time spent in the open arms of EPM (two-way ANOVA, group: $P = 0.0177$, laser: $P = 0.0017$, interaction: $P = 0.0117$; post hoc Bonferroni's test, NpHR(off vs. on): $P = 0.0009$, On(mCherry vs. NpHR): $P = 0.0058$). **k** Number of open-arm entries in the EPM (two-way ANOVA, group: $P = 0.3078$, laser: $P = 0.1770$, interaction: $P = 0.1770$; post hoc Bonferroni's test, NpHR(off vs. on): $P = 0.3611$, On(mCherry vs. NpHR): $P = 0.5857$). **l–t** Activation of NAc–VTA circuit could reverse the CES-induced anxiety-like behavior. **l** Schematic representation of in vivo virus infection and optical fiber implantation. **m** Experimental scheme of bilateral activation of VTA neurons receiving NAc projections and CES modeling. **n–q** Results of OFT ($n = 10$ mice for each group). **n** Representative locomotion traces of ChR2 or mCherry mice in OFT. **o** Percentage of time spent in the central area of OFT (two-tailed unpaired t-test, $P = 0.0018$). **p** Percentage of distance traveled in the central area of OFT (two-tailed unpaired t-test, $P = 0.0052$). **q** Total distance traveled in the OFT (two-tailed unpaired t-test, $P = 0.2154$). **r–t** Results of EPM ($n = 10$ mice for each group). **r** Representative EPM traces of ChR2 or mCherry mice. **s** Percentage of time spent in the open arms of EPM (two-tailed unpaired t-test, $P = 0.0005$). **t** Number of open-arm entries in the EPM (two-tailed unpaired t-test, $P = 0.0018$). The red and blue dots in the OFT and EPM locomotion traces reflect the start and end points of the mouse, respectively. All data are shown as mean ± SEM. ns not significant, $^{**}P \le 0.01$, $^{***}P \le 0.001$.

algorithms with a set of selected image features to produce pixel-based segmentations[68]. Second, a useful feature of ImageJ is the ROI Manager that allows selection of specific areas (such as red only, green only, and double-labeled) for evaluation. Finally, the cells with red only, green only, and double-labeled were automatically counted by "Analyze Particles" tool (lower threshold level: 50, upper threshold level: 255). In addition, to compare multiple specimens, staining, image acquisition (exposure time and gain), and image analysis were performed in parallel for the entire set. Detailed quantification methods and procedures of histology data were shown in Supplementary Fig. 12.

**Stereotactic surgeries**. For virus injection, mice were anesthetized with a mixture of 4% chloral hydrate (0.1 ml/10 g) and ketorolac tromethamine (20 μl/10 g) and placed on a stereotaxic apparatus (RWD Life Technology Co. Ltd., China). A volume of 150–300-nl virus was injected using calibrated glass microelectrodes connected to an infusion pump (micro 4, WPI). After injection, the glass micro-electrodes were left in place for an additional 10 min.

For electrophysiological recordings, mice were unilaterally injected with AAV2/9-DIO-hM3Dq-mCherry and AAV2/9-TH-Cre (1:1, 200 nl) into the VTA (anterior–posterior (AP): −3.08 mm; medial–lateral (ML): +0.5 mm; dorsal–ventral (DV): −4.55 mm); or mice were unilaterally injected with AAV2/9-DIO-hM4Di-mCherry and AAV2/9-TH-Cre (1:1, 200 nl) into the VTA. After allowing for 14 days of virus expression, the electrodes were implanted into the VTA.

For fiber-photometry experiments, mice were unilaterally injected with AAV2/9-DIO-GCaMP6s and AAV2/9-TH-Cre (1:1, 200 nl) into the VTA; or the AAV2/9-DIO-GCaMP6s was unilaterally injected into the VTA of DAT-Cre transgenic mice; or mice were unilaterally injected with AAV2/1-TH-Cre into the NAC (AP: +1.42 mm; ML: +0.75 mm; DV: −4.7 mm) and AAV2/9-DIO-GCaMP6s into the VTA. After allowing for 7 or 14 days of virus expression, mice were anesthetized again as previously described and the ceramic fiber-optic cannula (200 μm in diameter, 0.37 NA, Inper Technology Co., Ltd) was implanted into the same site above the VTA.

Chemogenetic behavioral experiments. For hM3Dq-mediated stimulation experiments, Cre-dependent hM3Dq virus (AAV2/9-DIO-hM3Dq-mCherry) and AAV2/9-TH-Cre were bilaterally injected into VTA (AP: −3.08 mm; ML: ±0.5 mm; DV: −4.55 mm) of mice; or the AAV2/9-DIO-hM3Dq-mCherry was bilaterally injected into VTA of DAT-Cre transgenic mice. For hM4Di-mediated stimulation experiments, Cre-dependent hM4Di virus (AAV2/9-DIO-hM4Di-mCherry) and AAV2/9-TH-Cre were bilaterally injected into VTA of mice. Mice injected with AAV2/9-DIO-mCherry and AAV2/9-TH-Cre viruses, or DAT-Cre transgenic mice injected with AAV2/9-DIO-mCherry virus, were used as the controls.

Optogenetic behavioral experiments. For optogenetic inhibition of NAc–VTA circuit experiments, Cre-dependent virus (AAV2/9-DIO-NpHR3.0-mCherry or AAV2/9-DIO-mCherry) and AAV2/9-GAD67-EGFP-2A-Cre were bilaterally injected into NAC (AP: +1.42 mm; ML: ±0.75 mm; DV: −4.7 mm) of mice; or mice were bilaterally injected with AAV2/1-TH-Cre into the NAC, and AAV2/9-DIO-NpHR3.0-mCherry (or AAV2/9-DIO-mCherry) into the VTA. For optogenetic activation of NAc–VTA circuit experiments, Cre-dependent virus (AAV2/9-DIO-ChR2-mCherry or AAV2/9-DIO-mCherry) and AAV2/9-GAD67-EGFP-2A-Cre were bilaterally injected into NAC of mice; or mice were bilaterally injected with AAV2/1-TH-Cre into the NAC, and AAV2/9-DIO-ChR2-mCherry

(or AAV2/9-DIO-mCherry) into the VTA. After allowing for 14 days of virus expression, ceramic fiber-optic cannulas were implanted bilaterally above the VTA (AP: −3.08 mm; ML: ±1.2 mm; DV: −4.3 mm; 8° angle).

For retrograde monosynaptic tracing, a mixture of helper viruses (AAV2/9-DIO-RVG and AAV2/9-DIO-TVA-EGFP) and AAV2/9-TH-Cre were unilaterally co-injected into the VTA of mice, or the helper viruses were unilaterally injected into the VTA of DAT-Cre mice. The total volume is 300 nl. After 10 days of repeated emotional stress exposure or 14 days of virus expression, mice were again anesthetized as previously described, and the rabies virus RV-ENVA-ΔG-DsRed was infused into the same site in the VTA using scar on the skull as guide validation. Nine days post RV-ENVA-ΔG-DsRed infusion, mice were transcardially perfused, and brain slices were prepared (40 μm) for tracing DsRed for circuit-mapping analysis or immunofluorescence staining analysis.

For the anterograde tracing of the NAC–VTA circuit, the Cre-dependent virus AAV2/9-DIO-mCherry and AAV2/9-GAD67-EGFP-2A-Cre were unilaterally delivered into the NAC of mice. Three weeks later, the mice were perfused, and the brains were sliced for imaging.

**In vivo electrophysiological recordings**. Broadband (0.3 Hz–7.5 kHz) neural signals were simultaneously recorded (16 bits @ 30 kHz) from implanted 16-ch arrays using a multichannel data-acquisition system (Zeus, Bio-Signal Technologies: McKinney, TX, U.S.A.). Spikes were extracted with high-pass (300 Hz) filter. Real-time spike sorting was performed using principal component analysis (PCA). Offline Sorter (Plexon: Dallas, TX, U.S.A.) was used for spike-sorting refinement before analyzing data in NeuroExplorer 5 (Nex Technologies: Boston, MA, U.S.A.). The targeted brain regions with the following coordinates: VTA (AP: −3.08 mm; ML: +0.5 mm; DV: −4.55 mm). Electrodes consisted of 16 individually insulated nichrome wires (35-μm inner diameter, impedance 300–900 Kohm; Stablohm 675, California Fine Wire, U.S.A.). Arrays of 16 micro-wires were arranged in a $3 \times 5 \times 5 \times 3$ pattern (~200 μm spacing between wires). Wires were attached to an 18-pin connector (Mil-Max). The implanted electrodes were secured with dental cement. According to the typical firing pattern of midbrain DA neurons, we classified neurons whose baseline firing rate (measured in the home cage) was below 10 Hz and exhibited long-duration action potentials (peak-to-peak duration >450 ms) as putative DA neurons[69,70].

**In vivo fiber photometry**. The fiber-photometry system (Inper Technology Co., Ltd) consisted of a 473 nm excitation light from LEDs, reflected off a dichroic mirror with a 435–488 nm reflection band and a 502–730 nm transmission band, and coupled into a 200 μm 0.37 NA optical fiber by an objective lens. We plotted the fluorescence change of different behavioral individual trials with different behavioral events. To examine the alterations of neuronal activity in ES-related events, we obtained the synchronized recordings of calcium signals in mice observing CSDS mice that received social defeats in the middle chamber. To detect $VTA^{DA}$ neurons or the functional connectivity from the NAc to the $VTA^{DA}$ neurons is engaged during exploration of innately anxiogenic environments, we recorded the track of mice in the behavioral tests, and synchronously recorded the neuronal calcium activity of $VTA^{DA}$ as a change in GCaMP6s fluorescence by in vivo calcium imaging. The raw $Ca^{2+}$ fluorescence (F) data were normalized and converted to z-scored traces. We derived the values of fluorescence change by calculating $\Delta F/F_0$, where $\Delta F$ is the variation of fluorescence between each sampling

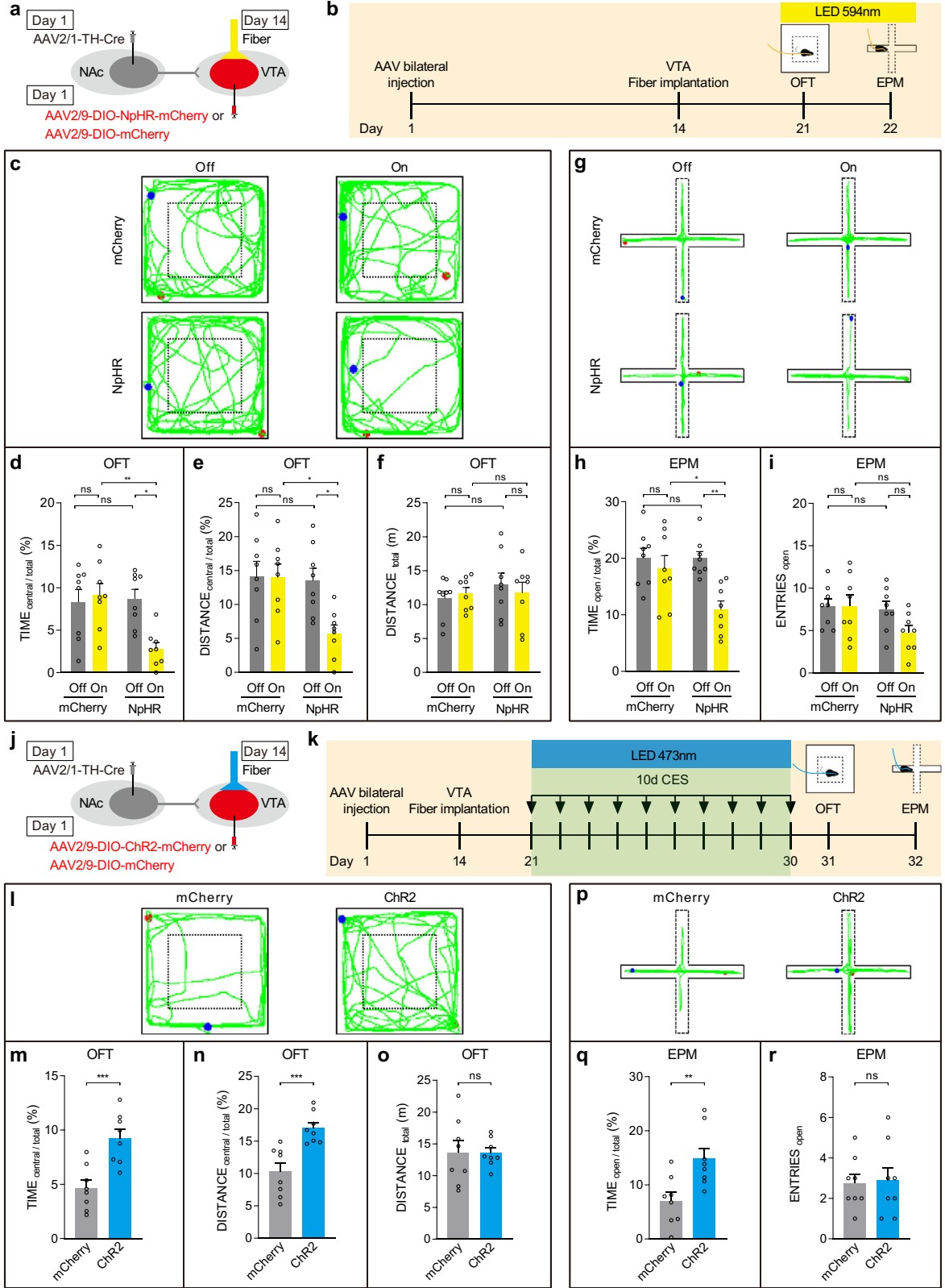

(sampling frequency 20 Hz), and $F_0$ is the averaged fluorescence baseline in the whole duration of the entire test (5 minutes). Photometry data were exported to MATLAB R2018a for further analysis.

**In vivo chemogenetic manipulation.** For hM3Dq-mediated activation experiments, a total of 32 or 40 C57BL/6 mice were divided into four groups: mCherry-NS group, mCherry-CNO group, hM3Dq-NS group, and hM3Dq-CNO group,

with 8 or 10 in each group. After allowing for 21 days of virus expression, mice were injected with CNO (3 mg per kg, i.p., BrainVTA, 190924) or normal saline (NS) and 30 min later were tested in the OFT, EPM, and SIT. For hM4Di-mediated inhibition experiments, C57BL/6 mice were divided into four groups: mCherry-NS group, mCherry-CNO group, hM4Di-NS group, and hM4Di-CNO group, with 8–10 in each group. After allowing for 14 days of virus expression, mice were injected with CNO or NS; 30 min later, mice were exposed to the emotional stress by observing the other mice that received repetitive social defeats (5 min, once a

**Fig. 8 Inhibition or activation of neural projection from NAc to VTA$^{DA}$ neurons could trigger or reverse the CES-induced anxiety-like behavior. a–i** Inhibition of neural projection from NAc to the VTA$^{DA}$ neurons triggered anxiety-like behavior. **a** Schematic representation of in vivo virus infection and optical fiber implantation. **b** Experimental scheme of virus injection and behavior tests. **c–f** Results of OFT ($n = 8$ mice for each group). **c** Representative locomotion traces of NpHR or mCherry mice in OFT with laser on or off. **d** Percentage of time spent in the central area of OFT (two-way ANOVA, group: $P = 0.0227$, laser: $P = 0.0477$, interaction: $P = 0.0104$; post hoc Tukey's test, NpHR(off vs. on): $P = 0.0102$, On(mCherry vs. NpHR): $P = 0.0056$). **e** Percentage of distance traveled in the central area of OFT (two-way ANOVA, group: $P = 0.0255$, laser: $P = 0.0432$, interaction: $P = 0.0505$; post hoc Tukey's test, NpHR(off vs. on): $P = 0.0310$, On(mCherry vs. NpHR): $P = 0.0208$). **f** Total distance traveled in the OFT (two-way ANOVA, group: $P = 0.4439$, laser: $P = 0.8520$, interaction: $P = 0.4565$; post hoc Tukey's test, NpHR(off vs. on): $P = 0.9086$, On(mCherry vs. NpHR): $P = 0.9999$). **g–i** Results of EPM ($n = 8$ mice for each group). **g** Representative exploring traces of NpHR or mCherry mice in EPM with laser on or off. **h** Percentage of time spent in the open arms of EPM (two-way ANOVA, group: $P = 0.0468$, laser: $P = 0.0043$, interaction: $P = 0.0488$; post hoc Tukey's test, NpHR(off vs. on): $P = 0.0055$, On(mCherry vs. NpHR): $P = 0.0321$). **i** Number of open-arm entries in the EPM (two-way ANOVA, group: $P = 0.0906$, laser: $P = 0.1793$, interaction: $P = 0.1793$; post hoc Tukey's test, NpHR(off vs. on): $P = 0.2318$, On(mCherry vs. NpHR): $P = 0.1443$). **j–r** Activation of neural projection from NAc to the VTA$^{DA}$ neurons could reverse the CES-induced anxiety-like behavior. **j** Schematic representation of in vivo virus infection and optical fiber implantation. **k** Experimental scheme of virus injection, CES modeling, and behavior tests. **l–o** Results of OFT ($n = 8$ mice for each group). **l** Representative locomotion traces of ChR2 or mCherry mice in OFT. **m** Percentage of time spent in the central area of OFT (two-tailed unpaired $t$-test, $P = 0.0009$). **n** Percentage of distance traveled in the central area of OFT (two-tailed unpaired $t$-test, $P = 0.0008$). **o** Total distance traveled in the OFT (two-tailed unpaired $t$-test, $P = 0.9683$). **p–r** Results of EPM ($n = 8$ mice for each group). **p** Representative EPM traces of ChR2 or mCherry mice. **q** Percentage of time spent in the open arms of EPM (two-tailed unpaired $t$-test, $P = 0.0075$). **r** Number of open-arm entries in the EPM (two-tailed unpaired $t$-test, $P = 0.8755$). The red and blue dots in the OFT and EPM locomotion traces reflect the start and end points of the mouse, respectively. All data are shown as mean ± SEM. ns, not significant, $^*P \leq 0.05$, $^{**}P \leq 0.01$, $^{***}P \leq 0.001$.

day) in three-chamber chronic emotional stress cage. After 10 consecutive days, mice were tested in the OFT, EPM, and SIT; or mice were exposed to 10-day CES, and OFT, EPM, and SIT were tested after 0.5 h of CNO injection on day 11.

**In vivo optogenetic manipulation.** For optogenetic inhibition of NAc–VTA circuit experiments, mice were tested in the OFT and EPM, while bilateral continuous yellow light (594 nm, 5–8 mW) illuminated on the VTA. For optogenetic activation of NAc–VTA circuit experiments, in the three-chamber cage, mice received blue-light (473 nm, 3–5 mW, 10 ms pulse, 20 Hz) stimulation on the VTA and light stimulation was delivered before the onset of the emotional stress stimulus and continued until the stimulus was finished. After 10 days of repeated emotional stress exposure, the OFT and EPM were tested. The same stimulus protocol was applied in the control group. Behavioral assays were performed immediately after light stimulation. The location of the fibers was examined after all the experiments.

**Statistical analysis.** We conducted the two-tailed Student's t-test for the experiments with only two groups, one-way ANOVA with post hoc Tukey's multiple-comparison test for the single-factor experiments with >two groups, two-way ANOVA followed by multiple comparisons with post hoc Tukey's test, or Bonferroni's test for the double factor experiments. Two-way repeated-measures (RM) ANOVA with post hoc Tukey's multiple-comparison test was used for data in the SIT conducted at multiple time points. Data distribution was analyzed using the Kolmogorov–Smirnov test. All data are expressed as mean ± SEM, and significance levels are indicated as $^*P \leq 0.05$, $^{**}P \leq 0.01$, and $^{***}P \leq 0.001$. P-values were calculated using GraphPad Prism 7 (Graph Pad Software, Inc.).

**Reporting summary.** Further information on research design is available in the Nature Research Reporting Summary linked to this article.

## Data availability

The data supporting the findings of this study are available within the article and Supplementary Information files. All the source data generated in this study are provided in the Source Data file. Source data are provided with this paper.

## Code availability

Associated code can be found here: https://github.com/BehavioralLAB/Fiberphotometry.

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

## Acknowledgements

This work was supported financially by grants from National Natural Science Foundation of China (Nos. 31871073 and 32171023 to B.T., and No. 31600821 to P.Z.), Program for New Century Excellent Talents in University (No. NCET-10-0415 to B.T.), and Program for Meridian-Viscera Correlationship Innovative Research Team of Shaanxi University of Chinese Medicine (No. 2019-YL09 to H.Q.).

## Author contributions

G.Q., P.Z. and T.L. performed most of the experiments. T.L. performed viral injections. G.Q., T.L. and M.L. performed the behavioral evaluations. L.Z., H.C. and X.L. performed the immunostaining. Q.Z. and F.H. performed the electrophysiology. G.Q. and M.L. performed the multichannel recordings. G.Q., P.Z., T.L., H.Q., X.C., J.M. and B.T. designed the research, analyzed the data, and wrote and edited the paper.

## Competing interests

The authors declare no competing interests.
