## [Peer Review File · Nature Communications]

NAc-VTA circuit underlies emotional stress-induced anxiety-like behavior in the three-chamber vicarious social defeat stress mouse modelREVIEWER COMMENTS

Reviewer #1 (Remarks to the Author):

An important implication of the work done in this manuscript is to elucidate the unique aspects of circuitry mediating anxiety following vicarious stress from circuitry mediating anxiety following directly experienced stress. The authors present strong evidence for the causal contributions of VTA, as well as NAc-VTA circuitry to anxiety-like phenotypes following vicarious stress. The premise of the paper is intriguing and there is great value in establishing the mechanism by which different forms of stress (in this case, vicarious rather than directly experienced) may contribute to subsequent psychopathologies. The studies in this regard were well planned – the authors included important control conditions, considered both structural and functional aspects of the circuitry, and evaluated phenotypes related to anxiety, depression, and sociability. However, limitations exist that decrease the accessibility and potential impact of the manuscript. In particular, there are inconsistencies with terminology and data analyses, as well as missing details in the methods that detract from the ability to follow the progression of the research. [Redacted]

Below, I have outlined areas that require clarification as well as suggestions for improvement.

Major comments:

- 1.) The authors state (line 86-88) that the current chronic emotional stress (CES) model represents an “improved” vicarious social defeat stress (VSDS) model, and also note that the presently used three-chamber (3C) setup is “re-designed”. The differences between the model used in this manuscript and previous models is not discussed further, making it difficult to evaluate how different it might be, or in what ways it might reflect an improvement. Furthermore, the paragraph beginning on line 96 (“This 3C-VSDS model is based on...”) seems to be describing the benefits of using the 3C-VSDS model. It is unclear whether this is referring to the protocol used in this manuscript (subsequently termed CES) or previously published versions of VSDS models. Finally, in the discussion, starting at line 389, the authors state that “the current study utilized 3C-VSDS” further confusing the differences between VSDS and CES.
- 2.) Standardizing the terminology related to CES would improve understanding of the manuscript. CES is used to describe the protocol through which mice are exposed to stress (10 days of observer stress). However, at other points CES is used to describe the experience of receiving stress over an extended period, regardless of the specific protocol details. The authors also reference “CES behavior”, which in context seems to be a term encompassing the long-term behavioral outcomes that appear in mice after exposure to the CES protocols. An example of appropriately precise CES language is on line 260: “CES-induced anxiety-like behavior”.
- 3.) While it is certainly true that animals are sensitive to the experiences and emotional states of other animals, the current phrasing in the paragraph beginning on line 96 implies that animals have empathy – which is a very difficult claim to test. Similarly, Lines 403-405 in the discussion also claim that the observer mouse is “sharing the affective state of the CSDS mouse” and that the resulting anxiety is “empathy-related”. Observational learning does not necessarily imply that processes related to empathy are at play. Please temper the language in this section to show that VSDS/CES have translational implications (but cannot directly replicate) empathy and empathy-related psychiatric disorders.

[Redacted]

6.) Along similar lines, an important implication of the work done in this manuscript is to elucidate the unique aspects of circuitry mediating anxiety following vicarious stress from circuitry mediating anxiety following directly experienced stress. Thus, the discussion would be greatly strengthened by discussing the anxiety findings with relation to other literature examining circuitry in CSDS mice.

General comments related to figures:

7.) It is unclear what the red and blue dots in the OFT, EPM and SIT locomotion traces indicate. It seems likely that they may reflect the start and end points of the mouse, but if this is the case, please explain why the start location is not consistent between groups.

8.) Please clarify the rationale for including different data in the OFT and EPM analyses, between figures. In particular, both fiber photometry experiments (Figures 2 and 5) analyses time in open and closed arms separately, whereas for all other figures data is presented as center (OFT) or open arm (EPM) time as a percentage of total time. Inconsistencies in data presentation make the figures more difficult to interpret. Similarly, please clarify why OFT was included in some but not other experiments.

9.) In multiple of the merged/co-labeled viral expression images, the merged image doesn't show as much co-labeling as would be expected given that expression of the active virus (e.g., hM4Dq in 3e, hM4Di in 4e, etc.) should be contingent upon the presence of TH-Cre in the same neurons. A quantification of the co-labeling may help reinforce the strength of the specificity. If expression of the active virus (DREADD, GCaMP, Optogenetic viruses) was more constitutive, this should be noted as a limitation of the conclusions that can be drawn about the specificity of the types of neurons comprising the VTA and NAC-VTA contributions to CES-induced anxiety-like behavior.

10.) In Figure 2 r,w, and Figure 5 m, o, the legend uses the term "position-adjusted" while the figure's axis label notes that the data is showing fluorescence/time. Please clarify.

11.) Check the axis for part 'n' of Figure 6 and Figure 7 – it should be "central", not "open".

Figure 1 and related text:

12.) It would be informative to know whether reduced interaction times in CES and CSDS mice reflect a reduction in social interaction, or instead a reduction in overall movement in the arena. Including distance travelled in the SIT arena may address this.

13.) Please clarify the presentation of SIT data. The results of the social interaction test are presented and discussed after the results from OFT and EPM, yet according to the Figure 1 schematic, SIT was the first behavioral test conducted. Along these same lines, in the Methods description of SIT, line 536 states that SIT was conducted 24 hours after the last social defeat session, however, other than Figure 1, SIT took place after OFT and EPM. Also, the methods include no description of the SIT sessions conducted throughout CES for the first experiment. At minimum, please note in the methods whether the first SIT was conducted before or after the first CES exposure – this is important to inform whether or not the data from SIT Day 1 is a true baseline.

14.) Line 921 of the Figure 1 legend should read "SPT" instead of "SFT"

Figure 2 and related text:

15.) Please clarify the rationale for using a single stress exposure to elicit VTA activity as it seems possible that the mechanisms underlying single versus repeated stress exposures would differ.

16.) Please clarify what stress protocol was used in the first fiber photometry experiment. The text (line 185) and description on the figure (2h) is "ES" (rather than CES, or SES). The legend for 2j notes that the data is "trial-by-trial", indicating that the stress exposure must be CES, but this is not clear from the related text describing the results of this experiment.

17.) Please indicate whether the staining shown in 2i is GCaMP or eGFP. If the same co-labeling with mCherry and TH viruses was done, as in 3e and subsequent figures, please include that here as well.

Figure 3 and related text:

18.) Locomotor hyperactivity in OFT is apparent only in this one experiment – and is especially notable given the parallel strong effect of center avoidance. This may warrant a brief discussion.

Figure 5 and related text:

19.) Lines 267-270 are hard to follow. It certainly seems true based on the tracing done in this experiment that NAc neurons innervate VTA-DA neurons. However, this conclusion does not follow from the finding that CES impacts the number of inputs. Please revise the sentence accordingly.

20.) The support for a specific connection between NAC-GABA neurons and VTA-DA neurons would be strengthened by showing co-labeling of GABA in NAC-projecting neurons and TH in VTA (post-synaptic terminals). The data showing that NAC-GABA neurons project to VTA is convincing, but the follow up studies that isolate VTA neurons receiving projections from NAC for GCaMP analysis do not necessarily exclude other non-GABAergic projections.

Discussion:

21.) The lack of depression-like phenotypes following CES should be acknowledged.

22.) Lines 412-421 posit that the transient nature of reduced interaction in the SIT was due to habituation. Please reconcile this explanation with the results of Figure 2, where interaction time did not differ between CES and control mice – there was only one SIT conducted, so habituation would not have occurred.

23.) The discussion of reward learning (lines 443-462) is not well integrated. At the end of this section it becomes somewhat clearer that this is an attempt to explain a possible mechanism by which NAc can attenuate anxiety through bidirectional modulation of VTA. However, the lengthy discussion of reward learning from 443-452 seems out of place, and the entire section would be improved by contextualizing this discussion in terms of the present findings.

24.) Discussing the finding that NAc-VTA projections are less dense in CES mice (Figure 5) would benefit the discussion, as it could potentially go a long way towards explaining why there are decrements in the NAc-mediated regulation of anxiety following CES.

Additional minor points of clarification:

25.) Line 76, 77 – Please clarify what is meant by the “stress system”. Is this referring to neural circuitry? Also clarify what is meant by “a coactivation state”. Is the idea that these three systems all interact with each other under stress conditions, or that they simply are all activated thus magnifying the effects of stress?

26.) Line 90 (also 132 and 508) – Please clarify what is meant by “context-dependent emotion”. There are not tests to show that the “emotion” related behaviors are specific to one context, and in fact the repercussions of CES exposure extends to multiple different contexts.

27.) Lines 162-164 – Please attenuate the phrasing to note that CSDS results in “evidence/symptoms of” anxiety etc, or, results in anxiety-like (etc) phenotypes.

28.) Some sentences have missing or unnecessary added words that make understanding the content of the sentence difficult. Please check the following: Lines 368-370; Line 1073; Lines 686-689;

29.) The sentence from 184-188 seems to have important information, but as written is hard to follow. Please consider revising.

30.) Line 317 – both ‘center’ and ‘central’ are used to describe the center zone of the OFT. Please be consistent.

31.) Line 326-327 – the sentence “It has been proven that..” should include citations of the relevant literature.

[Redacted]

33.) Line 394 and 398 – please revise the use of the word “mood” when referring to animals.

34.) Line 654 – what is “day 60” referring to? Is this the age of the mouse, or a time point in the experimental timeline?

35.) In general, the manuscript contained a lot of abbreviated terms that at times made it difficult to follow. Please consider only abbreviating terms that are used frequently throughout the manuscript (e.g., CES, OFT) and limit abbreviations of infrequent terms (e.g., SES, FOV).

Additional information needed in methods:

36.) In the "Animals" section: Please note how long mice were acclimated to the facility after arrival before surgery or testing began. Also note where the CD1 mice were obtained from

37.) In the "Virus Preparations" section: Please clarify what is meant by "AAV2/9" – were different serotypes used for different experiments? Also please include details for the AAV1-TH-Cre virus mentioned in Figure 5 and related text.

38.) In the "Three-chamber" section: Please provide the dimensions of the three chambers (this is noted in Figure 1 but should also appear in the methods text). Also please clarify whether the control mice also had a new CD1 housemate every day.

39.) In the "Behavioral Procedures" section:

a. Please describe the housing conditions for the mice after the end of CES. Were they returned to standard housing? If so, were they group housed? Did they share a cage with CSDS mice? Or, did they remain in the three-chamber housing but without a CD1? The housing conditions are important to consider when evaluating anxiety-like phenotypes.

40.) In the "Statistical Analysis" section: Was the data for SIT conducted at multiple time points subjected to a repeated-measures analysis?

Reviewer #2 (Remarks to the Author):

In this manuscript, the authors propose a GABAergic NAc to dopaminergic VTA circuit that mediates the anxiogenic effects of vicariously learned social defeat in male mice. The authors demonstrate that chronic emotional stress (CES)(induced by observing a conspecific mouse experience multiple social defeats), causes anxiety-like behaviors in the observer, but only transiently reduces social interaction. The authors demonstrate bidirectional control of these anxiety-like behaviors by VTA dopamine neurons, and further demonstrate bidirectional control by GABAergic NAc inputs to VTA on the same behaviors. [Redacted]

Overall this is an interesting paper that shows a novel role of the GABAergic NAc inputs to VTA dopamine neurons in mediating the anxiogenic effects of chronic emotional stress, with a strong emphasis on circuit dissection, and robust behavioral effects using a wide variety of chemogenetic and optogenetic strategies. The experiments as a whole are very thorough and clearly described. The behavioral measures, control groups, statistical analyses, and interpretation as a whole are appropriate.

Specific comments:

1) Some clarification is needed as to why VTA DREADD excitation was performed acutely right before open field/elevated plus/social interaction tests, whereas DREADD inhibition was performed chronically during each emotional stress exposure, and not before the open field/elevated plus/social interaction tests. This would perhaps speak to the effects of VTA dopamine firing on acquisition of anxiety vs. expression, but also raises the question of whether acute VTA dopamine inhibition prior to open field and EPM tests can block anxiogenic effects of CES. Along the same lines, the authors state in the discussion on page 17, lines 425-26 that 'VTA_DA neurons are not required for social interaction behavior'. However, since CNO injections were administered during chronic stress but not before social interaction tests, requirement of VTA neurons for this test cannot necessarily be ruled out.

[Redacted]

4) Regarding the finding that 'VTADA neurons receive proportionally fewer inputs from NAc area in CES mice compared to controls', further elaboration may be needed on the functional implications of these 'fewer inputs'. Specifically, would this imply reduced postsynaptic VTA inhibition by NAc (VTA disinhibition)? Furthermore, discussion with regard to experience-dependent NAc plasticity is needed.

Minor:

1) For VTA_DA DREADD excitations (Figure 3), it is unclear whether for open field, elevated plus, and social interaction tests, the CNO or NS injections were administered within or between subjects – specifically, did the same Gq mice undergo the tests twice (once with CNO and once with saline), or were they separate groups? This should be clarified in the methods, and if these CNO/NS treatments were within subjects, were order of treatment counterbalanced?

Reviewer #3 (Remarks to the Author):

Qi, Tian and co-workers investigate the circuit mechanisms of observational stress, using optogenetic, chemogenetic, and circuit tracing techniques in mice. The paper contains a large number of experiments, which in general seem to have been well-conducted. Nevertheless, in many cases data should be analyzed in more detail (see points below, and specific point 2), and conclusions should be stated more carefully (specific point 1).

Overall, the conceptual impact of the paper is difficult to assess, because unfortunately, the paper suffers from a complex mix of employed behaviors. The principal behavioral read-out is a "3-chamber vicarious social defeat stress" (3C-VSDS) model, in which an observer mouse experiences during a 10-days period a repeated observation of chronic social defeat of another conspecific mouse. This is complemented by standard open field tests (OFT) and elevated plus maze (EPM) tests to assay the anxiogenic nature of the previous "chronic emotional stress" (CES) in the observer mouse. The authors start by showing that cfos is elevated in VTA neurons (however, cfos was even more strongly elevated dorsally to the VTA [Fig. 2b] - this must be analyzed in an unbiased fashion), and that VTA neurons seem to be activated during ES (but the authors should explain how many mice went into the fiber photometry Ca trace trace shown in Fig. 2k). Chemogenetic activation of the VTA-dopamine neurons is then shown to increase "anxiety" in the OFT and EPM (Fig. 3). However, this was apparently simply a general measure of anxiety, without a previous 10-days CES protocol. Conversely, chemogenetic inactivation of the VTA-dopamine neurons reduces anxiety in the OFT and EPM tests which here, were studied specifically after the 10-day CES (Fig. 4). Nevertheless, the question remains whether the data in Fig. 4 might as well reflect a general role of VTA-dopamine neurons in anxiety, unrelated to the 10-day CES that the mice experienced before OFT and EPM.

The authors then go on to identify circuit elements connecting to the VTA that might be involved in these anxiety-related effects. They use rabies-mediated approaches to study the presynaptic neuron pools that provides input to VTA-dopamine neurons (Fig. 5). Here, reference and discussion to the previous detailed work of the group of Uchida-Watabe on exactly this question should be added (e.g. Watabe-Uchida et al. 2012 Neuron). The authors find that amongst other brain areas, the NAc connects to the VTA. They find that after CES, the number of back-labelled neurons in the NAc is "significantly" decreased (but N was only 3); however, this effect is not further discussed in the paper and remains a mystery - does it mean that the NAc structurally disconnects from the VTA after CES? They then exploit the anterograde jumping properties of

AAV1 serotype vectors, to measure Ca transients with fiber photometry specifically in those VTA-dopamine neurons that receive an input from the NAc (here, the previous work by Zingg et al. 2016 *Neuron*, and Hutson et al. 2016 *Gene Therapy*, who first showed the anterograde properties specifically of AAV1 must be cited, and the serotype number should always be given for all AAV constructs). However, the Ca transient is small (0.2 % Δ /FO; Fig. 5m), and raw Ca traces should be shown, and the description how this quantification was performed should be improved (same critique applies to Fig. 2q). Moreover, the proportion of VTA-neurons, amongst all VTA-dopamine neurons, that express GCamp6s with this anterograde jumping experiment should be quantified (see also specific point 2 below), and a suitable "control" experiment, expressing GCamp6s in another population of VTA-dopamine neurons, which might NOT be activated in the EPM, should be performed to demonstrate the specificity of this experiment. Thus overall, the experiments in Fig. 5 remain vague.

In Fig. 6, the authors show that optogenetic inhibition of NAc terminals in the VTA increases anxiety (Fig. 6, top part), but this is again simply investigates a role of the NAc - VTA terminals in anxiety alone, because these experiments were performed without a previous 10-day CES protocol. In Fig. 6k-s, the NAc terminals are then optogenetically activated in the VTA, and the effect on anxiety after 10-day CES is investigated. The results suggest that activating the NAc-VTA terminals "alleviates the CES-induced anxiety" (l. 321). However, there are two caveats to this conclusion: First, this experiment could simply indicate a general role of the NAc-VTA connection in anxiety which "overrides" the anxiety observed after 10-day CES (see also my above criticisms), and second, because the authors used optogenetic stimulation, it remains possible that due to backpropagating APs to the somata of the NAc neurons, that many other terminals of NAc neurons might be involved in this effect.

[Redacted]

Further specific points.

Many of my major, conceptual points were mentioned above in passing; I apologize for not having written them out in a point-point fashion for time reasons.

- 1) On many instances, the conclusion sentences in the Results section are not backed by the experiments (for example, the statements on l. 235-236 ("hallmark"); the concluding sentence in l. 292-293).
- 2) The presumed co-localization of two markers in the histological data should be analyzed on the single-cell level, and then quantified, to come up with a tight conclusion whether two markers are co-expressed or not (see data in Fig. 3e; Fig. 4e, Fig. 5c, Fig. 5g, Fig. 6c, Fig. 7f+i - in all cases, many "green" and "red" neurons seem to be visible).
- 3) This reviewer is not a specialist in NAc - VTA circuits. I nevertheless presume that there must have been previous studies on the role of the VTA - NAc (and back) projections in anxiety. A detailed literature search should be done, and these studies should then be discussed. In this context, the paper by Jong et al. 2019 (cited in another context in the current ms) should be discussed in more detail. de Jong et al. (2019) describe two separate dopamine projections from the VTA to the NAc; one involved in reward processing, and the other in aversive coding.
- 4) The discussion is not satisfactory, and largely mentions concepts which are not directly related to the experimental findings. The discussion should be completely re-worked. Best start by describing the main experimental results, also considering the possible limitations of the techniques you used, and discuss the mechanistic bases for your interpretations.

Point-by-point response to Reviewers' comments

Response to Reviewer # 1:

An important implication of the work done in this manuscript is to elucidate the unique aspects of circuitry mediating anxiety following vicarious stress from circuitry mediating anxiety following directly experienced stress. The authors present strong evidence for the causal contributions of VTA, as well as NAc-VTA circuitry to anxiety-like phenotypes following vicarious stress. The premise of the paper is intriguing and there is great value in establishing the mechanism by which different forms of stress (in this case, vicarious rather than directly experienced) may contribute to subsequent psychopathologies. The studies in this regard were well planned – the authors included important control conditions, considered both structural and functional aspects of the circuitry, and evaluated phenotypes related to anxiety, depression, and sociability. However, limitations exist that decrease the accessibility and potential impact of the manuscript. In particular, there are inconsistencies with terminology and data analyses, as well as missing details in the methods that detract from the ability to follow the progression of the research.

[Redacted]Below, I have

outlined areas that require clarification as well as suggestions for improvement.

Response: Thanks for carefully reviewing our work and propounding valuable comments, which greatly improves the quality of our paper. Below is our point-by-point response to the comments.

Major comments:

1.) The authors state (line 86-88) that the current chronic emotional stress (CES) model represents an “improved” vicarious social defeat stress (VSDS) model, and also note that the presently used three-chamber (3C) setup is “re-designed.” The differences between the model used in this manuscript and previous models is not discussed further, making it difficult to evaluate how different it might be, or in what ways it might reflect an improvement. Furthermore, the paragraph beginning on line 96 (“This 3C-VSDS model is based on...”) seems to be describing the benefits of using the 3C-VSDS model. It is unclear whether this is referring to the protocol used in this manuscript (subsequently termed CES) or previously published versions of VSDS models. Finally, in the discussion, starting at line 389, the authors state that “the current study utilized 3C-VSDS” further confusing the differences between VSDS and CES.

Response: Thanks for the valuable comments. We have added a corresponding discussion about the differences between the 3C-VSDS model and previous VSDS models in the revised manuscript [Line 420]. Animal models that can distinguish the neurobiological complexity between physiological and psychological stress are rare. Most models rely primarily on physical stressors

(e.g., social defeat, chronic unpredictable or mild stress, learned helplessness) and neglect the impact of emotional stress alone.

In the previous VSDS model, a two-chamber cage was used, and the CES mouse in the left chamber witnessed the CSDS mouse defeated by an aggressive CD1 mouse in the right chamber. After 5 minutes, the CES mouse was moved to a new cage, and the CSDS mouse was moved across the divider to the left chamber and then remained overnight adjacent to the CD-1. During the 10-day VSDS modeling, CES mice were transferred many times, and the environment was frequently changed every day, which did not effectively isolate the physical stress and environmental changes on mice.

In the current study, we improved the previous VSDS model. We established a three-chamber VSDS (3C-VSDS) model, providing an effective behavioral paradigm that can effectively induce chronic emotional stress (CES) by isolating mice's physical stress or confrontation. In our 3C-VSDS model, we employed a customized cage divided into three chambers. In the left chamber, the CES mouse observed the CSDS mouse received repetitive social defeats in the middle chamber. After 5 minutes of physical interaction, the CSDS mouse was separated from the CD1 and released into the right chamber. The CD1 mice, CES mice, and CSDS mice were housed together, separated by a perforated plastic divider that allowed visual, olfactory, and auditory contacts, but prevented physical interaction. Taken together, compared with the previous VSDS model, our 3C-VSDS model simplifies the procedure, and most importantly, effectively isolates the physical stress and focuses on the impact of emotional stress alone.

2.) Standardizing the terminology related to CES would improve understanding of the manuscript. CES is used to describe the protocol through which mice are exposed to stress (10 days of observer stress). However, at other points CES is used to describe the experience of receiving stress over an extended period, regardless of the specific protocol details. The authors also reference “CES behavior”, which in context seems to be a term encompassing the long-term behavioral outcomes that appear in mice after exposure to the CES protocols. An example of appropriately precise CES language is on line 260: “CES-induced anxiety-like behavior”.

Response: We much appreciate the Reviewer for this important suggestion, and we agree with the Reviewer that the terminology related to CES should be standardized. Accordingly, we have made corrections throughout the manuscript.

3.) While it is certainly true that animals are sensitive to the experiences and emotional states of other animals, the current phrasing in the paragraph beginning on line 96 implies that animals have empathy – which is a very difficult claim to test. Similarly, Lines 403-405 in the discussion also claim that the observer mouse is “sharing the affective state of the CSDS mouse” and that the resulting anxiety is “empathy-related”. Observational learning does not necessarily imply that processes

related to empathy are at play. Please temper the language in this section to show that VSDS/CES have translational implications (but cannot directly replicate) empathy and empathy-related psychiatric disorders.

Reply: We understand the Reviewer's concern here. We agree with the Reviewer that CES mice's observational learning in the 3C-VSDS model does not necessarily imply that processes are related to empathy. In the 3C-VSDS behavioral paradigm, the observer mouse experienced emotional stress leading to anxiety. We believe that there are two possible reasons to cause anxiety. The first one is that the observer mouse is experiencing emotional stress by sharing the CSDS mouse's affective state. This refers to empathy-related emotional contagion, the transfer of emotions between individuals is thought to occur automatically¹. The second one is that the observer mouse is experiencing social stress observing the aggressive CD1 mouse's violence attacking the conspecific mice. This observer mouse could be stressed by the sounds and smells of the CD1 mouse, or internal state changes related to self-preservation. These two causes work together in the observer CES mice in this 3C-VSDS model, and the extent of their respective roles remains to be studied. Accordingly, we have made some modifications to the revised manuscript.

[Redacted]

6.) Along similar lines, an important implication of the work done in this manuscript is to elucidate the unique aspects of circuitry mediating anxiety following vicarious

stress from circuitry mediating anxiety following directly experienced stress. Thus, the discussion would be greatly strengthened by discussing the anxiety findings with relation to other literature examining circuitry in CSDS mice.

Response: Thanks for the Reviewer's insightful comments. Accordingly, we have added the related discussion in the revised manuscript [Line 532-541].

General comments related to figures:

7.) It is unclear what the red and blue dots in the OFT, EPM and SIT locomotion traces indicate. It seems likely that they may reflect the start and end points of the mouse, but if this is the case, please explain why the start location is not consistent between groups.

Response: Thanks for the Reviewer's careful review. We apologize for not clearly stating this in the Figure Legend section. The red and blue dots in the OFT, EPM, and SIT locomotion traces reflect the start and end points of the mouse, respectively. We have supplemented the indication of the red and blue dots throughout figures to avoid the potentially confusing. There's a delay of 1-2 seconds between we put the mouse into the experimental apparatus and the time we started video recording the behavior, so the start location is not consistent between groups. Accordingly, we have added the related description in the revised manuscript in each figure legend.

8.) Please clarify the rationale for including different data in the OFT and EPM analyses, between figures. In particular, both fiber photometry experiments (Figures 2 and 5) analyses time in open and closed arms separately, whereas for all other figures data is presented as center (OFT) or open arm (EPM) time as a percentage of total time. Inconsistencies in data presentation make the figures more difficult to interpret. Similarly, please clarify why OFT was included in some but not other experiments.

Response: Thanks for the helpful comments. According to the previous study⁷, using the ratio of time spent on the open arms to the time spent on the closed arms is a significant indicator to assess the anxiety behavior of rodents; moreover, the behaviors that are typically recorded when rodents are in the elevated plus maze are the time spent and entries made on the open and closed arms.

9.) In multiple of the merged/co-labeled viral expression images, the merged image doesn't show as much co-labeling as would be expected given that expression of the active virus (e.g., hM4Dq in 3e, hM4Di in 4e, etc.) should be contingent upon the presence of TH-Cre in the same neurons. A quantification of the co-labeling may help reinforce the strength of the specificity. If expression of the active virus (DREADD, GCaMP, Optogenetic viruses) was more constitutive, this should be noted as a limitation of the conclusions that can be drawn about the specificity of the types of neurons comprising the VTA and NAC-VTA contributions to CES-induced anxiety-like behavior.

Response: Thanks for the Reviewer's significant comments. We agree with the Reviewer's opinion that it is essential to quantify the merged/co-labeled viral expression images. Meanwhile, we have performed quantification analysis of the co-localization of two markers in each histological data, such as revised Fig. 3e, f and 4e, f.

10.) In Figure 2 r,w, and Figure 5 m, o, the legend uses the term "position-adjusted" while the figure's axis label notes that the data shows fluorescence/time. Please clarify.

Response: We thank the Reviewer for pointing out this. The methods used in our manuscript was referenced to previously published study⁸. According to the reviewer suggested, we have amended the legend to avoid confusion [Line 1113, 1117, 1192].

11.) Check the axis for part 'n' of Figure 6 and Figure 7 – it should be "central," not "open."

Response: We thank the Reviewer for pointing out this unintended mistake, and this has been amended in the corresponding Figure as requested.

Figure 1 and related text:

12.) It would be informative to know whether reduced interaction times in CES and CSDS mice reflect a reduction in social interaction, or instead a reduction in overall movement in the arena. Including distance travelled in the SIT arena may address this.

Response: Thanks for the Reviewer's meaningful comments. In accordance with extensively published studies⁹⁻¹¹, we tested the approach-avoidance behavior of a C57BL/6 experimental mouse to an aggressive CD1 mouse during social interaction testing, in which the time spent in the social interaction zone is the major indicator of social interaction behavior. We agree with the Reviewer's opinion that it is crucial to analyze the distance traveled in the SIT arena. Accordingly, we have now performed an additional analysis that the total distance traveled in the SIT arena among all groups (Supplementary Fig. 1a, b). The result suggested that the decrease of total distance is compatible with the conclusion that the reduced interaction time in CES and CSDS mice.

13.) Please clarify the presentation of SIT data. The results of the social interaction test are presented and discussed after the results from OFT and EPM, yet according to the Figure 1 schematic, SIT was the first behavioral test conducted. Along these same lines, in the Methods description of SIT, line 536 states that SIT was conducted 24 hours after the last social defeat session, however, other than Figure 1, SIT took place after OFT and EPM. Also, the methods include no description of the SIT

sessions conducted throughout CES for the first experiment. At minimum, please note in the methods whether the first SIT was conducted before or after the first CES exposure – this is important to inform whether or not the data from SIT Day 1 is a true baseline.

Response: We apologize for not making this clear. In Fig. 1d, the social interaction tests (SIT) were conducted 1 hour after 5-minute social defeat session at day 1, 3, 5, 7 and 9. But on day 11, the SIT was performed after 24h. In order to avoid confusion, we revised the manuscript to describe the results of the SIT from Day1-9 and SIT in Day11 separately.

Exposure to a novel environment, such as an open field or elevated plus maze, has been used in studies assessing rodents' anxiety behavior. To this end, we conducted OFT/EPM prior SIT.

14.) Line 921 of the Figure 1 legend should read “SPT” instead of “SFT”.

Response: We thank the Reviewer for pointing out this unintended mistake. This has been amended as requested [Line 1073].

Figure 2 and related text:

15.) Please clarify the rationale for using a single stress exposure to elicit VTA activity as it seems possible that the mechanisms underlying single versus repeated stress exposures would differ.

Response: According to numerous published studies^{12,13}, short-term single stress exposure maybe leads functional plasticity and long-term repeated stress exposure lead to structural plasticity.

16.) Please clarify what stress protocol was used in the first fiber photometry experiment. The text (line 185) and description on the figure (2h) is “ES” (rather than CES, or SES). The legend for 2j notes that the data is “trial-by-trial”, indicating that the stress exposure must be CES, but this is not clear from the related text describing the results of this experiment.

Response: We apologize for not making this clear. In this experiment, we detected the neuronal activity of GCaMP6s-infected VTA^{DA} neurons in anxiety-inducing events by in vivo calcium recording. The target mice were given a 5-minute single emotional stress session, and each trail represents the target mice observed their conspecific receive attacks by CD1 aggressor. We revised the manuscript, and changed “ES” in Figure 2h to “SES”, and “ES” in Figure 2j and 2k to “Attack”, and in the corresponding Figure legend as requested [Line 1102-1105].

17.) Please indicate whether the staining shown in 2i is GCaMP or eGFP. If the same co-labeling with mCherry and TH viruses was done, as in 3e and subsequent figures, please include that here as well.

Response: Thanks for the helpful comments. GCaMP is created from a fusion of green fluorescent protein (GFP), calmodulin, and M13, a peptide sequence from myosin light chain kinase¹⁴. As suggested, the image of co-labeling with GCaMP and TH has been shown in Supplementary Fig. 3a, b and Supplementary Fig. 6a, b.

Figure 3 and related text:

18.) Locomotor hyperactivity in OFT is apparent only in this one experiment – and is especially notable given the parallel strong effect of center avoidance. This may warrant a brief discussion.

Response: The behavior results of Fig. 3, chemogenetic activation of VTA^{DA} neurons showed increment on locomotor activities, which was consistent with numerous published studies¹⁵⁻¹⁷. Meanwhile, we have added a brief discussion in the revised manuscript [Line 492-495].

Figure 5 and related text:

19.) Lines 267-270 are hard to follow. It certainly seems true based on the tracing done in this experiment that NAc neurons innervate VTA-DA neurons. However, this conclusion does not follow from the finding that CES impacts the number of inputs. Please revise the sentence accordingly.

Response: We thank the Reviewer for pointing out this inappropriate sentence, and this has been revised as requested [Line 274].

20.) The support for a specific connection between NAC-GABA neurons and VTA-DA neurons would be strengthened by showing co-labeling of GABA in NAC-projecting neurons and TH in VTA (postsynaptic terminals). The data showing that NAC-GABA neurons project to VTA is convincing, but the follow up studies that isolate VTA neurons receiving projections from NAC for GCaMP analysis do not necessarily exclude other non-GABAergic projections.

Response: We thank the Reviewer for raising this important point. Numerous studies have reported that the principal projection neurons of NAc region are GABAergic^{2,3}. We have performed an additional experiment using cell-specific retrograde monosynaptic tracing with EnvA-pseudotyped rabies virus (Supplementary Fig. 5a, b). The result also showed that the projection neurons from NAc to VTA^{DA} neurons are mostly GABAergic (Supplementary Fig. 5c).

Discussion:

21.) The lack of depression-like phenotypes following CES should be acknowledged.

Response: Thanks for the Reviewer's helpful comments. The lack of depression-like phenotypes following CES is a very important outcome. We have added this result in the revised manuscript [Line 411].

22.) Lines 412-421 posit that the transient nature of reduced interaction in the SIT was due to habituation. Please reconcile this explanation with the results of Figure 2,

where interaction time did not differ between CES and control mice – there was only one SIT conducted, so habituation would not have occurred.

Response: We thank the Reviewer's meaningful suggestion. As shown in the results in Fig. 2s and 2u, CES mice showed similar interaction levels with control mice after a 10-day exposure period. This result is consistent with the findings of fig. 1d to 1f. Habituation is an important adaptive property of the nervous system improving selective response to salient environmental aspects by ignoring familiar, repeated stimuli¹⁸. Here, habituation is generally referred to the CES mouse resilient to observational stress by repetitive exposures.

23.) The discussion of reward learning (lines 443-462) is not well integrated. At the end of this section it becomes somewhat clearer that this is an attempt to explain a possible mechanism by which NAc can attenuate anxiety through bidirectional modulation of VTA. However, the lengthy discussion of reward learning from 443-452 seems out of place, and the entire section would be improved by contextualizing this discussion in terms of the present findings.

Response: As suggested by the Reviewer, we have carefully revised and polished the Discussion section.

24.) Discussing the finding that NAc-VTA projections are less dense in CES mice (Figure 5) would benefit the discussion, as it could potentially go a long way towards explaining why there are decrements in the NAc-mediated regulation of anxiety following CES.

Response: We have added the discussion about the CES-induced structure plasticity of NAc-VTA projections in the revised manuscript [Line 506].

Additional minor points of clarification:

25.) Line 76, 77 – Please clarify what is meant by the “stress system”. Is this referring to neural circuitry? Also clarify what is meant by “a coactivation state”. Is the idea that these three systems all interact with each other under stress conditions, or that they simply are all activated thus magnifying the effects of stress?

Response: We apologize for not making this clear. The stress system involves the autonomic nervous system and the hypothalamic-pituitary-adrenal (HPA) axis. Stress conditions result in the stimulation of the HPA axis mediated by an increase in the production of corticotrophin-releasing hormone (CRF) and adrenocorticotrophic hormone (ACTH). Stressors can also activate the sympathetic innervation and promote a rise in the release of prolactin and growth hormones¹⁹. Stress-related pathways may alter the state of the immune system and cytokine release by the immune cells, which have receptors for stress hormones²⁰. These interactions are bidirectional as cytokines can alter the release ACTH and CRF. The dysfunction of the stress system and immune system can directly activate oxidative stress^{21, 22}. In turn, oxidative stress can affect the release of CRF²³ and the secretion of cytokines by immune cells²⁴, and thus, intensify the response of the stress and immune systems to a stressor.

We believe that these three systems all interact with each other under stressful conditions.

26.) Line 90 (also 132 and 508) – Please clarify what is meant by “context-dependent emotion”. There are not tests to show that the “emotion” related behaviors are specific to one context, and in fact the repercussions of CES exposure extends to multiple different contexts.

Response: Thanks for the Reviewer’s meaningful comments. To avoid the potentially confusing, we have changed “context” to “scene” [Line 87, 118, 579].

27.) Lines 162-164 – Please attenuate the phrasing to note that CSDS results in “evidence/symptoms of” anxiety etc, or, results in anxiety-like (etc) phenotypes.

Response: Thanks for the Reviewer’s meaningful suggestions. We have made a correction and replace “anxiety, depression” with “anxiety-like and depression-like phenotypes” [Line 150].

28.) Some sentences have missing or unnecessary added words that make understanding the content of the sentence difficult. Please check the following: Lines 368-370; Line 1073; Lines 686-689;

Response: Thanks for the Reviewer’s careful review. We have made corrections in revised manuscript [Line 400, 765-768, 1240].

29.) The sentence from 184-188 seems to have important information, but as written is hard to follow. Please consider revising.

Response: Thanks for the reviewer’s helpful comments. We rewrite this part in the revised manuscript [Line 176-181].

30.) Line 317 – both ‘center’ and ‘central’ are used to describe the center zone of the OFT. Please be consistent.

Response: We thank the Reviewer for pointing out this, and all the “center” have been corrected to “central” [Line 325].

31.) Line 326-327 – the sentence “It has been proven that..” should include citations of the relevant literature.

Response: Thanks for the Reviewer’s careful review. We have included the relevant literature as [Ref 23, 24].

[Redacted]

33.) Line 394 and 398 – please revise the use of the word “mood” when referring to animals.

Response: We sincerely appreciate the valuable comments. We have revised the word “mood” when referring animals as your kindly suggestion [Line 432, 436].

34.) Line 654 – what is “day 60” referring to? Is this the age of the mouse, or a time point in the experimental timeline?

Response: We apologize for not making this clear. The “day 60” represents a time point in the experimental timeline. We have corrected as following. “Mice were exposed to the emotional stress by observing the other mice received single social defeats at the timepoint of day 60 [Line 735].”

35.) In general, the manuscript contained a lot of abbreviated terms that at times made it difficult to follow. Please consider only abbreviating terms that are used frequently throughout the manuscript (e.g., CES, OFT) and limit abbreviations of infrequent terms (e.g., SES, FOV).

Response: We thank the Reviewer for pointing out this. The infrequent terms have been corrected as full name throughout the manuscript.

Additional information needed in methods:

36.) In the “Animals” section: Please note how long mice were acclimated to the facility after arrival before surgery or testing began. Also note where the CD1 mice were obtained from.

Response: We thank the Reviewer’s significant comments. Mice were group-housed for a week prior to the experiment and allowed to acclimate to the behavioral testing room for at least one hour before testing. The CD1 mice were also purchased from Beijing Vital River Laboratory Animal Technology Co., Ltd. And the above information has been added in the “Animals” section of Methods [Line 548, 550].

37.) In the “Virus Preparations” section: Please clarify what is meant by “AAV2/9” – were different serotypes used for different experiments? Also please include details for the AAV1-TH-Cre virus mentioned in Figure 5 and related text.

Response: We apologize for not making this clear. AAV2/9 refers to recombinant adeno-associated virus that single-stranded AAV2 vectors pseudotyped with viral capsids from serotype 9.

AAV2/1 refers to recombinant adeno-associated virus that single-stranded AAV2 vectors pseudotyped with viral capsids from serotype 1. AAV2/1 exhibit anterograde trans-synaptic spread properties²⁵. AAV2/1-Cre from transduced presynaptic neurons effectively and specifically drives Cre-dependent transgene expression in selected postsynaptic neuronal targets, allowing axonal tracing and functional manipulations of the latter input-defined neuronal population²⁵. Therefore, mice were infused with AAV2/9-DIO-GCaMP6s into the VTA, and AAV2/1-TH-Cre into the NAc. AAV2/1-TH-Cre virus could cross the NAc-VTA synapse and help the expression of GCaMP6s in VTA^{DA} neurons.

38.) In the “Three-chamber” section: Please provide the dimensions of the three chambers (this is noted in Figure 1 but should also appear in the methods text). Also please clarify whether the control mice also had a new CD1 housemate every day.

Response: Thanks for the Reviewer’s important comments. We have supplemented the dimensions of the three chambers in the “Three-chamber vicarious social defeat stress model” section of Methods. Without receiving any physical and emotional stimuli, control mice were housed with new CD1 mice every day in the three chambers cage. This has also been described in the Methods section of the revised manuscript.

39.) In the “Behavioral Procedures” section:

a. Please describe the housing conditions for the mice after the end of CES. Were they returned to standard housing? If so, were they group housed? Did they share a cage with CSDS mice? Or, did they remain in the three-chamber housing but without a CD1? The housing conditions are important to consider when evaluating anxiety-like phenotypes.

Response: After the end of CES, each mouse was housed in a single cage. And we have supplemented this in the “Three-chamber vicarious social defeat stress model” section of Methods.

[Redacted]

40.) In the “Statistical Analysis” section: Was the data for SIT conducted ad multiple time points subjected to a repeated-measures analysis?

Response: Thanks for the Reviewer’s significant comments. In the SIT conducted ad multiple time points, we performed two-way repeated-measures (RM) ANOVA with post hoc Tukey's multiple comparisons test to analyze the data. We have supplemented the two-way RM ANOVA in Statistical analysis of Methods [Line 805]. Moreover, the exact P values of the ANOVA parameters (row, column factor and interaction) and multiple comparisons have been reported in the Statistical analysis of Supplementary Information.

References

1. Kelly JR, Iannone NE, McCarty MK. Emotional contagion of anger is automatic: An evolutionary explanation. *Br J Soc Psychol* **55**, 182-191 (2016).

[Redacted]

7. Walf AA, Frye CA. The use of the elevated plus maze as an assay of anxiety-related behavior in rodents. *Nat Protoc* **2**, 322-328 (2007).
8. Carta I, Chen CH, Schott AL, Dorizan S, Khodakhah K. Cerebellar modulation of the reward circuitry and social behavior. *Science* **363**, (2019).
9. Golden SA, Covington HE, 3rd, Berton O, Russo SJ. A standardized protocol for repeated social defeat stress in mice. *Nat Protoc* **6**, 1183-1191 (2011).
10. Lorsch ZS, *et al.* Stress resilience is promoted by a Zfp189-driven transcriptional network in prefrontal cortex. *Nat Neurosci* **22**, 1413-1423 (2019).
11. Shen CJ, *et al.* Cannabinoid CB1 receptors in the amygdalar cholecystokinin glutamatergic afferents to nucleus accumbens modulate depressive-like behavior. *Nat Med* **25**, 337-349 (2019).
12. Josselyn SA, Tonegawa S. Memory engrams: Recalling the past and imagining the future. *Science* **367**, (2020).
13. He HY, Shen W, Zheng L, Guo X, Cline HT. Excitatory synaptic dysfunction cell-autonomously decreases inhibitory inputs and disrupts structural and functional plasticity. *Nat Commun* **9**, 2893 (2018).
14. Nakai J, Ohkura M, Imoto K. A high signal-to-noise Ca(2+) probe composed of a single green fluorescent protein. *Nat Biotechnol* **19**, 137-141 (2001).
15. Wang S, Tan Y, Zhang JE, Luo M. Pharmacogenetic activation of midbrain dopaminergic neurons induces hyperactivity. *Neurosci Bull* **29**, 517-524 (2013).
16. Vardy E, *et al.* A New DREADD Facilitates the Multiplexed Chemogenetic Interrogation of Behavior. *Neuron* **86**, 936-946 (2015).

17. Boekhoudt L, *et al.* Chemogenetic activation of dopamine neurons in the ventral tegmental area, but not substantia nigra, induces hyperactivity in rats. *Eur Neuropsychopharmacol* **26**, 1784-1793 (2016).
18. Ramaswami M. Network plasticity in adaptive filtering and behavioral habituation. *Neuron* **82**, 1216-1229 (2014).
19. Glaser R, Kiecolt-Glaser JK. Stress-induced immune dysfunction: implications for health. *Nat Rev Immunol* **5**, 243-251 (2005).
20. Dhabhar FS, *et al.* High-anxious individuals show increased chronic stress burden, decreased protective immunity, and increased cancer progression in a mouse model of squamous cell carcinoma. *PLoS One* **7**, e33069 (2012).
21. Rivero-Segura NA, *et al.* Prolactin prevents mitochondrial dysfunction induced by glutamate excitotoxicity in hippocampal neurons. *Neurosci Lett* **701**, 58-64 (2019).
22. Zhang C, Rissman RA. Corticotropin-releasing factor receptor-1 modulates biomarkers of DNA oxidation in Alzheimer's disease mice. *PLoS One* **12**, e0181367 (2017).
23. Raff H, Jacobson L, Cullinan WE. Augmented hypothalamic corticotrophin-releasing hormone mRNA and corticosterone responses to stress in adult rats exposed to perinatal hypoxia. *J Neuroendocrinol* **19**, 907-912 (2007).
24. Naik E, Dixit VM. Mitochondrial reactive oxygen species drive proinflammatory cytokine production. *J Exp Med* **208**, 417-420 (2011).
25. Zingg B, *et al.* AAV-Mediated Anterograde Transsynaptic Tagging: Mapping Corticocollicular Input-Defined Neural Pathways for Defense Behaviors. *Neuron* **93**, 33-47 (2017).

Response to Reviewer # 2:

In this manuscript, the authors propose a GABAergic NAc to dopaminergic VTA circuit that mediates the anxiogenic effects of vicariously learned social defeat in male mice. The authors demonstrate that chronic emotional stress (CES)(induced by observing a conspecific mouse experience multiple social defeats), causes anxiety-like behaviors in the observer, but only transiently reduces social interaction. The authors demonstrate bidirectional control of these anxiety-like behaviors by VTA dopamine neurons, and further demonstrate bidirectional control by GABAergic NAc inputs to VTA on the same behaviors.

[Redacted]

Overall this is an interesting paper that shows a novel role of the GABAergic NAc inputs to VTA dopamine neurons in mediating the anxiogenic effects of chronic emotional stress, with a strong emphasis on circuit dissection, and robust behavioral effects using a wide variety of chemogenetic and optogenetic strategies. The experiments as a whole are very thorough and clearly described. The behavioral measures, control groups, statistical analyses, and interpretation as a whole are appropriate.

Response: Thanks for your professional review work and propounding significant comments, which are helpful to improve the quality of this paper significantly. Below, we set out our responses to these suggestions.

Specific comments:

1) Some clarification is needed as to why VTA DREADD excitation was performed acutely right before open field/elevated plus/social interaction tests, whereas DREADD inhibition was performed chronically during each emotional stress exposure, and not before the open field/elevated plus/social interaction tests. This would perhaps speak to the effects of VTA dopamine firing on acquisition of anxiety vs. expression, but also raises the question of whether acute VTA dopamine inhibition prior to open field and EPM tests can block anxiogenic effects of CES. Along the same lines, the authors state in the discussion on page 17, lines 425-26 that 'VTA DA neurons are not required for social interaction behavior'. However, since CNO injections were administered during chronic stress but not before social interaction tests, requirement of VTA neurons for this test cannot necessarily be ruled out.

Response: Thanks for the Reviewer's valuable comments. Using *in vivo* fiber photometry, we found that the robust activation of VTA^{DA} neurons is involved in the anxiety-like behavior in the innate anxiogenic environment induced by CES (Fig. 2). This result linked the VTA^{DA} neuron hyperactivity to CES's anxiety-like behavior and suggested that VTA^{DA} neuron hyperactivity is a crucial feature of anxious states. Therefore, we first tested whether the direct activation of VTA^{DA} neurons, without a previous 10-days CES protocol, is sufficient to induce

anxiety-like behavior (Fig. 3). Moreover, we hypothesized that artificially inhibiting VTA^{DA} neuron activity would promote resilience to CES's anxiety-like behavior; in this session, we used chemogenetic methods to inhibit the VTA^{DA} neuron hyperactivity induced by CES, so the 10-days CES protocol is necessary (Fig. 4).

As suggested, we conducted a new experiment to test whether one-time acute VTA dopamine inhibition prior to behavioral tests can also block CES-induced anxiogenic effects. AAV2/9-DIO-hM4Di-mCherry and AAV2/9-TH-Cre were bilaterally injected into the VTA of C57BL/6 mice (Supplementary Fig. 4a, b). All mice received 10-day CES treatment and a battery of behavior tests following the one-time administration of CNO or saline (intraperitoneally, 30 min before testing). At the behavioral level, we found that acute VTA dopamine inhibition prior to open field test (OFT) and evaluated plus maze test (EPM) can also significantly block the anxiogenic effects of CES (Supplementary Fig. 4c–i).

Furthermore, we also tested whether acute VTA dopamine inhibition prior to social interaction test (SIT) can influence social interaction behavior. The results showed that acute CNO injection, comparing with other groups, did not alter the time spent in the social interaction zone after 10-day CES (Supplementary Fig. 4j, k). These data further suggested that VTA DA neurons are not required for social interaction behavior.

[Redacted]

4) Regarding the finding that 'VTA DA neurons receive proportionally fewer inputs from NAc area in CES mice compared to controls', further elaboration may be needed on the functional implications of these 'fewer inputs'. Specifically, would this imply reduced postsynaptic VTA inhibition by NAc (VTA disinhibition)? Furthermore, discussion with regard to experience-dependent NAc plasticity is needed.

Response: We thank the Reviewer for raising this important point. Using retrograde viral tracing, we revealed that VTA^{DA} neurons receive proportionally fewer inputs from the NAc area in CES mice than controls (Fig. 5a–e). The result suggested that the structural plasticity of axons of direct inputs from NAc to VTA^{DA} neurons was changed after 10-day CES exposure. Numerous studies have reported that the principal projection neurons of the NAc region are all GABAergic^{1,2}. We have performed an additional experiment using cell-specific retrograde monosynaptic tracing with EnvA-pseudotyped rabies virus (Supplementary Fig. 5a, b). The result also showed that the projection neurons from NAc to VTA^{DA} neurons are most GABAergic (Supplementary Fig. 5c). The discussion about the experience-dependent NAc plasticity has been added in the revised Discussion section [Line 506] accordingly.

Minor:

1) For VTA DA DREADD excitations (Figure 3), it is unclear whether for open field, elevated plus, and social interaction tests, the CNO or NS injections were administered within or between subjects – specifically, did the same Gq mice undergo the tests twice (once with CNO and once with saline), or were they separate groups?

This should be clarified in the methods, and if these CNO/NS treatments were within subjects, were order of treatment counterbalanced?

Response: We thank the Reviewer for pointing out this ambiguous language. And this has been amended in the corresponding part of Methods as following. For hM3Dq-mediated activation experiments (Figure 3), according to the method of random number table, a total of 40 male C57BL/6 mice were divided into four groups: mCherry-NS, mCherry-CNO, hM3Dq-NS, and hM3Dq-CNO group, with 10 in each group. After allowing for 21 days of virus expression, mice in mCherry-NS and hM3Dq-NS group were administered normal saline; mice in mCherry-CNO and hM3Dq-CNO group were administered Clozapine-N-oxide (CNO, 3 mg per kg); and 30 min later, mice were tested in the OFT, EPM and SIT. This has been clarified in the methods of the revised manuscript.

References for Reviewer # 2

1. Kohnomi S, Konishi S. Multiple actions of a D(3) dopamine receptor agonist, PD128907, on GABAergic inhibitory transmission between medium spiny neurons in mouse nucleus accumbens shell. *Neurosci Lett* **600**, 17-21 (2015).
2. Koo JW, *et al.* Loss of BDNF signaling in D1R-expressing NAc neurons enhances morphine reward by reducing GABA inhibition. *Neuropsychopharmacology* **39**, 2646-2653 (2014).

Response to Reviewer # 3:

Qi, Tian and co-workers investigate the circuit mechanisms of observational stress, using optogenetic, chemogenetic, and circuit tracing techniques in mice. The paper contains a large number of experiments, which in general seem to have been well-conducted. Nevertheless, in many cases data should be analyzed in more detail (see points below, and specific point 2), and conclusions should be stated more carefully (specific point 1).

Response: Thanks for your professional review work and positive comments on our manuscript. We have carefully revised the manuscript according to your valuable comments.

Overall, the conceptual impact of the paper is difficult to assess, because unfortunately, the paper suffers from a complex mix of employed behaviors. The principal behavioral read-out is a "3-chamber vicarious social defeat stress" (3C-VSDS) model, in which an observer mouse experiences during a 10-days period a repeated observation of chronic social defeat of another conspecific mouse. This is complemented by standard open field tests (OFT) and elevated plus maze (EPM) tests to assay the anxiogenic nature of the previous "chronic emotional stress" (CES) in the observer mouse. The authors start by showing that cfos is elevated in VTA neurons (however, cfos was even more strongly elevated dorsally to the VTA [Fig. 2b] - this must be analyzed in an unbiased fashion), and that VTA neurons seem to be activated during ES (but the authors should explain how many mice went into the fiber photometry Ca trace trace shown in Fig. 2k).

Response: Thanks for the Reviewer's meaningful comments. We agree with the reviewer's opinion and conducted the statistical analysis of the region dorsally to the VTA as suggested (Supplementary Fig. 2a). This region is the red nucleus, a key node of brain networks in motor control¹. In the 3C-VSDS model, the target mouse in the left chamber was vicariously conditioned for scene dependent emotion by observing CSDS mouse received repetitive social defeats in the middle chamber. This threatening event elicits a strong defense reaction in the SES mouse. According to the recent study², the red nucleus might represent a locus of coordinating oromotor, respiratory, locomotor and anti-nociceptive responses to hypoxia stimuli, and might be involved in modulation of the respiratory output during the defense reaction. Certainly, it would be interesting to assess the effects of red nucleus in emotional stress-related abnormal behaviour. Meanwhile, we have pointed out this in [Line 486] of Discussion section accordingly.

Additionally, we have added the number of mice (n = 4, Fig. 2k) in corresponding part of Figure legends in revised manuscript as suggested.

Chemogenetic activation of the VTA-dopamine neurons is then shown to increase "anxiety" in the OFT and EPM (Fig. 3). However, this was apparently simply a general measure of anxiety, without a previous 10-days CES protocol. Conversely, chemogenetic inactivation of the VTA-dopamine neurons reduces anxiety in the OFT and EPM tests which here, were studied specifically after the 10-day CES (Fig. 4). Nevertheless, the question remains whether the data in Fig. 4 might as well reflect a general role of VTA-dopamine neurons in anxiety, unrelated to the 10-day CES that the mice experienced before OFT and EPM.

Response: We thank the Review for pointing out this unclear content. We have carefully checked and revised the corresponding Results section to avoid confusion as suggested. Briefly, we found that the robust activation of VTA^{DA} neurons is involved in CES-induced anxiety-like behavior using *in vivo* fiber photometry (Fig. 2). This result linked the VTA^{DA} neuron hyperactivity to the CES-induced anxiety-like behavior, and suggested that VTA^{DA} neuron hyperactivity maybe as a key feature of anxious states. Therefore, we firstly tested whether the direct activation of VTA^{DA} neurons, without a previous 10-days CES protocol, is sufficient to induce anxiety-like behavior (Fig. 3). Secondly, we hypothesized that artificially inhibiting VTA^{DA} neuron activity would promote resilience to the anxiety-like behavior of CES; in this session, we used chemogenetic methods to inhibit the CES-induced VTA^{DA} neuron hyperactivity, so the 10-day CES protocol is necessary (Fig. 4).

Furthermore, we conducted a new experiment to test whether one-time acute VTA dopamine inhibition prior to behavioral tests can also block CES-induced anxiogenic effects. AAV2/9-DIO-hM4Di-mCherry and AAV2/9-TH-Cre were bilaterally injected into the VTA of C57BL/6 mice (Supplementary Fig. 4a, b). All mice received 10-day CES treatment and a battery of behavior tests following the one-time CNO administration or saline (intraperitoneally, 30 min before testing). At the behavioral level, we found that acute VTA dopamine inhibition prior to open field test (OFT) and evaluated plus maze test (EPM) can also significantly block the anxiogenic effects of CES (Supplementary Fig. 4c–i). We also tested whether acute VTA dopamine inhibition prior to social interaction test (SIT) can influence social interaction behavior. The results showed that acute CNO injection, comparing with other groups, did not alter the time spent in the social interaction zone after 10-day CES (Supplementary Fig. 4j–k). These data further suggested that VTA DA neurons are not required for social interaction behavior.

The authors then go on to identify circuit elements connecting to the VTA that might be involved in these anxiety-related effects. They use rabies-mediated approaches to study the presynaptic neuron pools that provides input to VTA-dopamine neurons (Fig. 5). Here, reference and discussion to the previous detailed work of the group of Uchida-Watabe on exactly this question should be added (e.g. Watabe-Uchida et al. 2012 Neuron).

Response: Thanks for the Reviewer's helpful comments. As suggested by the Reviewer, we have added the corresponding reference in the revised manuscript [Ref. 19].

The authors find that amongst other brain areas, the NAc connects to the VTA. They find that after CES, the number of back-labelled neurons in the NAc is "significantly" decreased (but N was only 3); however, this effect is not further discussed in the paper and remains a mystery - does it mean that the NAc structurally disconnects from the VTA after CES?

Response: Thanks for the Reviewer's meaningful comments. We agree with the Reviewer's opinion that the alteration of back-labelled neurons represents the CES triggering significant decrease in structure plasticity of direct inputs from NAc to VTA^{DA} neurons, which is consistent with previous study³. To make the manuscript more convincing, we have revised the corresponding sentences [Line 274] in Results section and added the limitations of the current study in Discussion section [Line 506] accordingly.

They then exploit the anterograde jumping properties of AAV1 serotype vectors, to measure Ca transients with fiber photometry specifically in those VTA-dopamine neurons that receive an input from the NAc (here, the previous work by Zingg et al. 2016 Neuron, and Hutson et al. 2016 Gene Therapy, who first showed the anterograde properties specifically of AAV1 must be cited, and the serotype number should always be given for all AAV constructs).

Response: Thanks for the Reviewer's valuable comments. As suggested by the Reviewer, we have added the corresponding references in the revised manuscript [Ref. 21, 22]. Meanwhile, the serotype number for all AAV constructs were given in the revised manuscript and figures.

However, the Ca transient is small (0.2 % delta/F0; Fig. 5n), and raw Ca traces should be shown, and the description how this quantification was performed should be improved (same critique applies to Fig. 2q).

Response: Thanks for Reviewer's significant comments. The quantitative method and statistical analysis of our study were according to the previously published study⁴. In the test of *in vivo* fiber photometry, we derived the values of fluorescence change ($\Delta F/F_0$) by calculating $(F-F_0)/F_0$, where F_0 is the averaged fluorescence baseline in the whole duration of the entire test (5 minutes). On this basis, the values of fluorescence change ($\Delta F/F_0$) is relatively small, which is also consistent with previous research findings⁴. As the Reviewer suggested, we have added the raw Ca traces (Supplementary Fig. 3c and Supplementary Fig. 6c) and the detailed description of raw data quantification analysis [Line 759 of Methods section].

Moreover, the proportion of VTA-neurons, amongst all VTA-dopamine neurons, that express GCamp6s with this anterograde jumping experiment should be quantified (see

also specific point 2 below), and a suitable "control" experiment, expressing GCaMP6s in another population of VTA-dopamine neurons, which might NOT be activated in the EPM, should be performed to demonstrate the specificity of this experiment. Thus overall, the experiments in Fig. 5 remain vague.

Response: Thanks for Reviewer's significant comments. As requested, we performed quantification analysis of percentage of co-labeling of GCaMP6s-positive cells in VTA TH-expression cells (Supplementary Fig. 3a, b and Supplementary Fig. 6a, b).

The VTA is a heterogeneous brain structure and VTA^{DA} neurons consist of several DA neuron sub-populations with different properties^{5,6}. At present, there are few studies focused on the role of VTA dopaminergic neurons projecting from different nuclei in anxiety. In this study, we just researched the function of VTA^{DA} neurons receiving the NAc projections, and proved that this group of VTA^{DA} neurons participate in anxiety. It is very interesting to find out the functions and roles of other population of VTA-dopamine neurons without NAc projections in the anxious stress. We will focus on those group neurons in our further research.

In Fig. 6, the authors show that optogenetic inhibition of NAc terminals in the VTA increases anxiety (Fig. 6, top part), but this is again simply investigates a role of the NAc - VTA terminals in anxiety alone, because these experiments were performed without a previous 10-day CES protocol. In Fig. 6k-s, the NAc terminals are then optogenetically activated in the VTA, and the effect on anxiety after 10-day CES is investigated. The results suggest that activating the NAc-VTA terminals "alleviates the CES-induced anxiety" (l. 321). However, there are two caveats to this conclusion: First, this experiment could simply indicate a general role of the NAc-VTA connection in anxiety which "overrides" the anxiety observed after 10-day CES (see also my above criticisms), and second, because the authors used optogenetic stimulation, it remains possible that due to backpropagating APs to the somata of the NAc neurons, that many other terminals of NAc neurons might be involved in this effect.

Response: Thanks for the Reviewer's important and constructive comments. This is a very important point the Reviewer makes here and we understand their concern. We totally agree with the Reviewer that due to backpropagating APs to the somata of the NAc neurons, many other terminals of NAc neurons might be involved in anxiety. The previous experimental strategy may result in unwanted activation of collateral targets via antidromic stimulation. To eliminate this defect, Zingg et al. revealed that the application of anterograde transsynaptic spread viruses is useful for tracing and manipulating neural circuits in a postsynaptic cell-type- and input-specific manner⁷.

Accordingly, in order to make our conclusion more convinced, we added a new experiment to eliminate the impact of unwanted activation of collateral targets. C57BL/6 mice were infused with AAV2/1-TH-Cre into the NAc, and

AAV2/9-DIO-NpHR-mCherry or AAV2/9-DIO-mCherry into the VTA. Optic fiber cannula was implanted over the VTA, and the circuit was optically inhibited with yellow light (Supplementary Fig. 7a, b). The results showed that inhibiting the neural projection from the NAc to the VTA^{DA} neurons triggered anxiety-like behavior in the OFT (Supplementary Fig. 7c–f) and EPM (Supplementary Fig. 7g–i). Going further, we use the same strategy for virus injection. Optic fibers were implanted into the VTA of mice, and the circuit was optically activated with blue light each day during the CES paradigm over 10 days of modeling (Supplementary Fig. 8a, b). The results of OFT and EPM also showed that activation of neural projection from the NAc to the VTA^{DA} neurons could reverse the CES-induced anxiety-like behavior (Supplementary Fig. 8c–i). These results also indicate that bidirectional modulation of NAc-VTA circuit could mimic or reverse the CES induced anxiety-like behavior.

[Redacted]

Further specific points.

Many of my major, conceptual points were mentioned above in passing; I apologize for not having written them out in a point-point fashion for time reasons.

1) On many instances, the conclusion sentences in the Results section are not backed by the experiments (for example, the statements on l. 235-236 ("hallmark"); the concluding sentence in l. 292-293).

Response: Thanks for the Reviewer's valuable comments, and the related sentences have been carefully checked and revised [Line 229, 299-301].

2) The presumed co-localization of two markers in the histological data should be analyzed on the single-cell level, and then quantified, to come up with a tight conclusion whether two markers are co-expressed or not (see data in Fig. 3e; Fig. 4e, Fig. 5c, Fig. 5g, Fig. 6c, Fig. 7f+i - in all cases, many "green" and "red" neurons seem to be visible).

Response: Thanks for the Reviewer's meaningful comments. As the Reviewer suggested, we have quantified the proportion of co-localization of two markers in each histological data (Fig. 3f; Fig. 4f; Fig. 5d; Fig. 5i; Fig. 6d; Fig. 7g and Fig. 7k; Supplementary Fig. 3a, b and Supplementary Fig. 6a, b in revised manuscript).

3) This reviewer is not a specialist in NAc - VTA circuits. I nevertheless presume that there must have been previous studies on the role of the VTA - NAc (and back) projections in anxiety. A detailed literature search should be done, and these studies should then be discussed. In this context, the paper by Jong et al. 2019 (cited in another context in the current ms) should be discussed in more detail. de Jong et al. (2019) describe two separate dopamine projections from the VTA to the NAc; one involved in reward processing, and the other in aversive coding.

Response: Thanks for the Reviewer's helpful comments. Accordingly, we have added the references [Ref 46] and a detailed discussion in the revised manuscript.

4) The discussion is not satisfactory, and largely mentions concepts which are not directly related to the experimental findings. The discussion should be completely re-worked. Best start by describing the main experimental results, also considering the possible limitations of the techniques you used, and discuss the mechanistic bases for your interpretations.

Response: Thanks for the Reviewer's significant comments. We have carefully checked and revised the content of Discussion section, as the Reviewer's and constructive suggestions.

References for Reviewer # 3

1. Houk JC. Red nucleus: role in motor control. *Curr Opin Neurobiol* **1**, 610-615 (1991).
2. Ghali MGZ. Rubral modulation of breathing. *Exp Physiol* **104**, 1595-1604 (2019).
3. Beier KT, et al. Rabies screen reveals GPe control of cocaine-triggered plasticity. *Nature* **549**, 345-350 (2017).
4. Carta I, Chen CH, Schott AL, Dorizan S, Khodakhah K. Cerebellar modulation of the reward circuitry and social behavior. *Science* **363**, (2019).
5. Lammel S, et al. Input-specific control of reward and aversion in the ventral tegmental area. *Nature* **491**, 212-217 (2012).
6. Brischoux F, Chakraborty S, Brierley DI, Ungless MA. Phasic excitation of dopamine neurons in ventral VTA by noxious stimuli. *Proc Natl Acad Sci U S A* **106**, 4894-4899 (2009).
7. Zingg B, et al. AAV-Mediated Anterograde Transsynaptic Tagging: Mapping Corticocollicular Input-Defined Neural Pathways for Defense Behaviors. *Neuron* **93**, 33-47 (2017).

REVIEWER COMMENTS

Reviewer #1 (Remarks to the Author):

Many aspects of the revised manuscript show significant improvement. Thank you to the authors for their careful consideration of each of my suggestions. Issues with terminology and clarity have been addressed, the necessary methodological details have been added, quantification of overlapping neuronal populations has been included, and much of the discussion has been revised.

[Redacted]

In addition, there are still sections of the discussion that are difficult to navigate. For example, the section from 496-503 details the role of VTA-DA neurons in reward processing, which seems out of place until line 515 when the authors propose a relationship between reward and anxiety relief. Moreover, it's unclear what in the CES environment would activate the VTA-NAC "reward" projection that the authors predict leads to the subsequent NAC-VTA inhibition that could reduce anxiety. Adding to the confusion in this section, the revised sentence added to line 506, serving to transition between the reward literature and the present findings, appears to be missing a word so the meaning is difficult to interpret.

[Redacted]

Reviewer #2 (Remarks to the Author):

The authors here have thoroughly addressed my comments and concerns. They have provided sufficient clarification on methodology, as well as new experiments and analyses. Their new results have provided convincing evidence that both supports their initial findings, and quells any alternative explanations. Overall, this manuscript demonstrates a clear and convincing role for the GABAergic NAc to dopaminergic VTA circuit in mediating CES-induced anxiety, [Redacted] These findings will be impactful in providing new mechanistic understanding of social/emotional stress circuits.

Reviewer #3 (Remarks to the Author):

- The full review is added as a pdf file so that editors and authors can see my text highlights -

Point-by-point response to Reviewers' comments

Response to Reviewer # 1:

Many aspects of the revised manuscript show significant improvement. Thank you to the authors for their careful consideration of each of my suggestions. Issues with terminology and clarity have been addressed, the necessary methodological details have been added, quantification of overlapping neuronal populations has been included, and much of the discussion has been revised.

Response: We thank the Reviewer for this comment. Through this suggestions, we greatly improved the quality of the manuscript.

[Redacted]

In addition, there are still sections of the discussion that are difficult to navigate. For example, the section from 496-503 details the role of VTA-DA neurons in reward processing, which seems out of place until line 515 when the authors propose a relationship between reward and anxiety relief. Moreover, it's unclear what in the CES environment would activate the VTA-NAC "reward" projection that the authors predict leads to the subsequent NAC-VTA inhibition that could reduce anxiety. Adding to the confusion in this section, the revised sentence added to line 506, serving to transition between the reward literature and the present findings, appears to be missing a word so the meaning is difficult to interpret.

Response: We thank the Reviewer for this comment. The projections from VTA^{DA} neurons to the NAc play a key role in reward, and dysfunction in this circuit has

been implicated in depression¹. It is reported that reward-induced dopamine release in the NAc was suppressed in depression². These findings suggest that suppression of the mesolimbic reward circuit may be a common neuroplastic change underlying depression. In view of this, we propose a relationship between reward and anxiety. Reward and anxiety, at once linked to each other and antagonistic to each other. We can only think about it separately.

[Redacted]

References for Reviewer # 1:

1. Nestler EJ, Carlezon WA, Jr. The mesolimbic dopamine reward circuit in depression. *Biol Psychiatry* **59**, 1151-1159 (2006).
2. Minami S, Satoyoshi H, Ide S, Inoue T, Yoshioka M, Minami M. Suppression of reward-induced dopamine release in the nucleus accumbens in animal models of depression: Differential responses to drug treatment. *Neurosci Lett* **650**, 72-76 (2017).

Response to Reviewer # 2:

The authors here have thoroughly addressed my comments and concerns. They have provided sufficient clarification on methodology, as well as new experiments and analyses. Their new results have provided convincing evidence that both supports their initial findings, and quells any alternative explanations. Overall, this manuscript demonstrates a clear and convincing role for the GABAergic NAc to dopaminergic VTA circuit in mediating CES-induced anxiety, [Redacted]
These findings will be impactful in providing new mechanistic understanding of social/emotional stress circuits.

Response: We thank the Reviewer for the useful comments. Through this suggestions, we greatly improved the quality of the manuscript.

Response to Reviewer # 3:

The authors have tried to address some of my previous concerns. In many of my previous points I asked that more raw data be provided. Unfortunately, the authors have put only little raw data, and what they provided went to the Suppl. Figures, such that it is very difficult to link the raw data to the bar graphs in the main Figures.

Response: We truly appreciate your consideration on our work and important comment. We apologize for the oversight of not blending the Statistical Report with the Supplementary Information, and putting more data in Suppl. Figures. Following the reviewer's advice, we have made some modifications in the revised manuscript.

In my re-review, I will go Figure-by-Figure to specifically raise each point, and to think about the conclusions which can, or cannot be drawn in my view, from each Figure:

Figure 1

- Statistics. The methods states how statistics was performed in general. However, the reader needs to know for each specific dataset, which statistical test was used. For example, in Fig.1d authors conclude " but with the extension of time, the social interaction time returned to normal (Fig. 1d)" (l. 128). Which statistical test was used to make sure that this conclusion is valid? I see some stars at some of the individual data points but no further explanation.

- In general, for each data set in the study, it has to be stated which statistical test was used. Moreover, have the authors tested for normal distribution of the experimental data before selecting the tests?

Response: We thank the Reviewer for this comment. In Fig. 1d, we performed two-way repeated-measures (RM) ANOVA with post hoc Tukey's multiple comparisons test to analyze the data. Before selecting statistic methods, data distribution was analyzed using the Kolmogorov-Smirnov test. As the Reviewer suggested, we have added the statistical analysis in the figure legends of the revised manuscript. Moreover, detailed statistical methods (the number of mice, normality test, equal variance test, statistic method, *P* value, *F*/*t* value, and Post hoc multiple comparisons test) were listed in the Supplementary Information.

- Suppl. Fig. 1 analyzes the "total distance travelled" in the SIT of Fig. 1d. Significant decreases are found. Does this mean that instead of a social interaction deficit, there was a locomotion deficit?

Response: We thank the Reviewer for this meaningful comment. Our result suggested that the variance tendency of total distance travelled between the three groups of mice was very similar to that of interaction time. And this result may be explained by two reasons as following. Firstly, in accordance with extensively published studies¹⁻³, the time spent in the social interaction zone as the major indicator of social interaction behavior. Secondly, we tested the approach-avoidance behavior of a C57BL/6 experimental mouse to an aggressive CD1

mouse during social interaction testing. The distance moved of experimental mouse, continued exposure the visual and olfactory cue transmission of the CD1 aggressor, will inevitably be affected during the SIT. Thus, the decrease of total distance is compatible with the conclusion that the reduced interaction time in CES and CSDS mice. In Fig 1j, we also measured total distance traveled in OFT, in which the only difference with SIT is that there is no CD1 aggressor. The results showed that there is no deference in CES and control mice in total distance traveled, which proved the locomotion activity of CES mice are normal.

Figure 2: *In this Figure, the authors would like to link the "CES" state of a mouse, to VTA neuron activity. However, I think that for oversights and limitations in the quantification of raw data (especially Ca traces), this statement can at present not firmly be made:*

- In response to my criticism, the authors have now added an analysis of the change of cfos+ neurons outside the VTA. They show that the "red nucleus" has a similarly increased number of cfos+ neurons (Suppl. Fig. 2). What is the conclusion from this analysis? Why was the VTA chosen for study, and not the "red nucleus"? How many other brain regions might show increased activity after CES?

Response: Thanks for the Reviewer's meaningful comments. In addition to VTA, we also found elevated c-fos protein expression in the red nucleus, indicating that the red nucleus participate in emotional stress processing. In the 3C-VSDS model, the experimental mouse in the left chamber was vicariously conditioned for scene dependent emotion by observing CSDS mouse received repetitive social defeats in the middle chamber. This threatening event elicits a strong defense reaction in the experimental mouse. According to the recent study⁴, the red nucleus might represent a locus of coordinating oro-motor, respiratory, locomotor and anti-nociceptive responses to hypoxia stimuli, and might be involved in modulation of the respiratory output during the defense reaction. By comparison, the VTA is involved in emotion-related behaviors, particularly in processing stressful events⁵. Therefore, we chose the VTA for study. Besides VTA, we think lots of other brain regions also might show increased activity after CES. The role of those brain regions in emotional stress processing needs further study.

- Results text states: "Because of the difficulties and defects of in vivo electrophysiology" (L.167). What do you mean by "defects"? - this statement must be amended.

Response: We thank the Reviewer for this comment. The term "defects" is a misnomer, and the related sentences have been carefully checked and amended [Line 165].

- Suppl. Fig. 3b. The author now show a "pie chart" to analyze the overlap between TH and GCamp6s expression. I guess this quantifies the percentage of GCamp6s+ cells which are ALSO positive for TH. Alternatively, a Venn diagram could be shown to show the 3 populations more completely and un-biased (- cells only pos. for TH, -

cells only pos. for GCamp6s, and double-pos. cells). However, at least the following n-numbers must be given: Number of cells analyzed - number of sections analyzed - (in addition to the number of mice which is now reported).

Response: Thank you very much for this comment. As the Reviewer suggested, we have changed the pie chart to the Venn diagram in Suppl. Fig. 3b in the the revised manuscript, and added additional information to show the number of cells analyzed, number of sections analyzed in addition to the number of mice.

- Figure 2k is entitled in Figure legends as " k) Representative peri-event plot of average Ca^{2+} transients (n = 4 mice)." (l. 1103). In response to my criticism, the n-number is now given. But how can it be a "representative" average Ca transient, if n = 4 mice were averaged? The authors should also let us know, first, how many "trials" went into this average (is the trace an "average" of the "average"?), and second, how many mice in total were measured (N = 4, or more?).

Response: Thank you very much for this comment. The term "representative" is a misnomer, and the sentences have been amended [Line 1037]. The average was obtained by 14 trials in total 4 mice.

- It is unclear why the Ca transient actually starts rising ~ 500 ms BEFORE the alignment of the attack (see Fig. 2k, dashed line). Is it possible that the VTA neurons are stimulated by some other aspect of the CD1 mice? Is it possible that olfactory / vomeronasal cues, or also sounds / ultrasounds get transmitted between the chambers, which might activate the VTA of the observed mouse even before an attack occurs?

Response: Thank you very much for this comment. In the three-chamber cage, the observer mouse in the left chamber was vicariously conditioned for scene-dependent emotion by observing conspecific demonstrator mouse received repetitive social defeats in the middle chamber. In order to accurately define the beginning of attack behavior, we defined the moment when CD1 aggressor contacted the body of demonstrator mice as the onset of attack. In this case, before the contact, the conspecific demonstrator mouse awoke to the danger by the approaching of CD1 aggressor, and tried to escape with a startled squawk. The olfactory or auditory cues get transmitted between the chambers, which might activate the VTA of the observed mouse even before the onset of attack occur.

- The Ca^{2+} traces now shown in Suppl. Fig. 3c (added in response to my criticism) unfortunately DO NOT have the same units as the quantification of the "average" data in Fig. 2q. Data in Fig. 2q (same as before) is given in $\Delta F/F_0$, whereas the newly added Ca traces are shown in "z-score". Thus, as a reader, I cannot make the comparison of how "typical" the raw data (Suppl. Fig. 3c) is with respect to the data shown in Fig. 2q. I think the raw data MUST BE INCLUDED in the main Figure 2 for easy viewing by the reader, and it MUST HAVE THE SAME UNITS as the analysis in Fig. 2q.

Response: Thank you very much for this comment. As the Reviewer suggested, we have unified the units and moved the raw Ca traces to the main Figure 2. In this group of experiments, we recorded the track of mice in the behavioral tests, and synchronously recorded the neuronal calcium activity of VTA^{DA} as a change in GCaMP6s fluorescence by in vivo calcium imaging. The purpose is to compare the activity changes of VTA^{DA} in CES mice in safe or innately anxiogenic environment. The raw Ca²⁺ fluorescence data were normalized and converted to z-scored traces. We derived the values of fluorescence change by calculating $\Delta F/F_0$, where ΔF is the variation of fluorescence between each sampling (sampling frequency 20Hz), and F_0 is the averaged fluorescence baseline in the whole duration of the entire test (5 minutes).

- Still regarding the Ca traces in Suppl. Fig. 3c, please state whether this is a single trace from a single mouse, or in some way averaged. Some discussion as to when the VTA neurons show Ca activity seems to be needed. E.g. are there Ca events before the "cyan" area i.e. before the animals enter the open arms?

Response: Thank you very much for this comment. The Ca traces in Suppl. Fig. 3c (Fig. 2p in the revised manuscript) is a representative single trace from a single mouse. As the Reviewer suggested, we have added the related discussion in the revised manuscript [Line 428-435].

- L. 193 states: "Compared to control mice, CES mice showed no significant differences in time in the interaction zone, average Ca²⁺ activity, and variation rate of Ca²⁺ transients (fluorescence/time) (Fig. 2s-w).

=> was only the 11th day analyzed, as in Fig. 1f? If not, were the results of the SIT different from the ones in Fig. 1, when control and CES mice were compared? (see Fig. 1d)

Response: We thank the Reviewer for this comment. Yes, the results of the SIT in Fig. 2s-w (Fig. 2t-x in the revised manuscript) were analyzed at the 11th day. CES mice showed no significant differences in time in the interaction zones, as in Fig. 1f.

=> what is "variation rate of Ca²⁺ transients"? and what is concluded from this measure?

Response: Thank you very much for this comment. The variation rate of Ca²⁺ transients is calculated by $\Delta F/F_0$, where ΔF is the variation of fluorescence between each sampling (sampling frequency 20Hz), and F_0 is the averaged fluorescence baseline in the whole duration of the entire test (5 minutes). The variation rate of Ca²⁺ transients reflects the changes in the active state of VTA^{DA} neurons. A positive value indicates that VTA^{DA} neurons are activated and a negative value indicates that VTA^{DA} neurons are inhibited. To better clarify this point, we have changed "variation rate of Ca²⁺ transients" into "variation rate of Ca²⁺ fluorescence" in the revised manuscript.

-L. 196 states: "Together, these results indicate that the robust activation of VTA^{DA} neurons is involved in the anxiety-like behavior in the innate anxiogenic environment induced by CES"

=> The only evidence for the "robust activation", as far as I can see, stems from the quantifications of Ca transients in Fig. 2q and 2r. Since it is unclear what a $\Delta F/F_0$ of actually one per thousand (!) means (this is likely a quite small Ca elevation), the statement of L. 196 cannot be maintained.

Response: We thank the Reviewer for this comment. Yes, the term "robust" is a misnomer, and the related sentences have been carefully checked and amended [Line 193]. In the test of *in vivo* fiber photometry, we derived the values of fluorescence change by calculating $\Delta F/F_0$, where ΔF is the variation of fluorescence between each sampling (sampling frequency 20Hz), and F_0 is the averaged fluorescence baseline in the whole duration of the entire test (5 minutes). On this basis, the values of fluorescence change (ΔF) between each sampling (50ms) is relatively small, which is also consistent with previous research findings⁶.

Figure 3

- In Fig. 3f, quantification should be made in form of a Venn diagram, which also takes into account the TH+ cells NOT infected by hM3Dq-mCherry (see above point regarding Suppl. Fig. 3b). At least, the "n" numbers of cells and sections should also be stated.

Response: We thank the Reviewer for this comment. As the Reviewer suggested, we have changed the pie chart to the Venn diagram in Fig. 3f, and added additional information to show the number of cells analyzed, number of sections analyzed in addition to the number of mice in the the revised manuscript.

- The design of this experiment is questionable. The authors chemogenetically stimulate many VTA-dopamine neurons non-selectively, and observe signs of anxiety in the open field test (Fig. 3j, k), and EPM (Fig. 3n, o). However, these results don't make a link to the CES state, but rather only analyze "general" anxiety (see also my previous general criticism). Also, given that VTA dopamine neurons are often regarded to process sensory inputs with a positive reinforcing values (rewards; rewards prediction errors ect.), the connection to these previous VTA DA studies needs to be discussed.

Response: We thank the Reviewer for this comment. In Fig. 3, we proved that chemogenetic activation of VTA^{DA} neurons directly triggers general anxiety-like behavior, while chemogenetic inhibition of VTA^{DA} neurons blocks the CES-induced anxiety. Through the above mimic and reverse experiments, we concluded that activation of VTA^{DA} neurons was both sufficient and indispensable conditions in CES-induced anxiety. These results linked the VTA^{DA} neuron hyperactivity to CES's anxiety-like behavior and suggested that VTA^{DA} neuron hyperactivity is a crucial feature of anxious states.

Accordingly, the related studies were discussed in the revised manuscript [Line 465-480].

Figure 4

- As in the above quantifications of cellular expression I would prefer to see a Venn-like diagram (see above). I am especially concerned about a large group of neurons located in the "bottom-right" part of the image of Fig. 4e (SAME in Fig. 3e, and VERY SIMILAR in Fig. 5c) that seem to be "red" alone. In addition, there are many "green" alone cells, but this might simply indicate that the transfection efficiency is not 100%. Coming back to the more important "red" cells, it might be that these cells have a weak expression of TH (otherwise the statistics shown in Fig. 4f cannot be correct). However, if the authors think there is a weak TH signal, this MUST BE DOCUMENTED in detail, by showing enhanced images zooming-into the group of "red" cells, and marking every one of the enhanced cells as TH (and mCherry) "positive", or "negative". Also, the threshold fluorescence for regarding a cell as "positive" or "negative" should be stated in the Methods.

Response: Thanks for the Reviewer's significant comments. As the Reviewer suggested, we have changed the pie chart to the Venn diagram and showed enlarged images of the bottom-right part of Fig. 3e, 4e and 6c in Supplementary Fig. 4a, 5a and 6a in the the revised manuscript.

The analysis and quantitation of fluorescence images were performed by ImageJ^{7,8}. As the previously published study⁹, the "positive" and "negative" cells were assessed by setting a "threshold" using the thresholding tool of ImageJ. According to the Reviewer's suggestion, the thresholding window (Lower threshold level: 50; Upper threshold level: 255) were stated in the Methods [Line 605-617].

For automated cell counting of fluorescent protein-positive neurons, the fluorescence images were analyzed and quantitated by ImageJ plugins and features. Briefly, the red channel and green channel of fluorescence images were separately segmented by Trainable Weka Segmentation plugin firstly. The Trainable Weka Segmentation is a Fiji plugin that combines a collection of machine learning algorithms with a set of selected image features to produce pixel-based segmentations¹⁰. Secondly, a useful feature of ImageJ is the ROI Manager that allows selection of specific areas (such as red only, green only, and double-labelled) for evaluation. Finally, the cells with red only, green only, and double-labelled were automatically counted by "Analyze Particles" tool. In addition, to compare multiple specimens, staining, image acquisition (exposure time and gain), and image analysis were performed in parallel for the entire set.

These points are important, because it might be doubted whether the use of an AAV2/9 vector with a TH promoter can indeed limit expression faithfully to VTA

dopamine neurons. Alternatively and probably better, a DATCre mouse would have been used? (see also Lammel et al. 2015 Neuron "Matters arising").

Response: We thank the Reviewer for this comment. Because of the specific expression of tyrosine hydroxylase (TH) and dopamine transporter (DAT) in dopaminergic neurons, they are widely used as molecular markers of dopaminergic neurons. Th-cre or DAT-cre recombinant viruses and transgenic mice have been widely used to specifically and efficiently express Cre recombinase in dopaminergic neurons, and then use cre-loxP system to specifically knock-in exogenous proteins into DA neurons. Th-cre recombinant virus or transgenic mice is the most commonly used strategies. In many researches (A Cell, 2019, 178, 653–671. Fig 7o, p. B Nature Neuroscience, 2018, 21, 952–962. Fig 3a, b. C Nature Neuroscience, 2018, 21, 1072–1083. Fig 1a. D Nature Communications, 2020, 11 6218. Fig 1a, b. E Nature Communications, 2020, 11: 3764. Fig 2c, d. F Biological Psychiatry, 2020, 88: 597-610. Fig 3E. As shown below)¹¹⁻¹⁶, Th-cre virus or transgenic mice have been used to specifically label DA neurons.

The researcher should choose TH-cre virus or transgenic mice with both high expression efficiency and high specificity to faithfully limit the expression to DA neurons. In the current study, we used TH antibody to do immunofluorescence double-staining to verify the infection efficiency and specificity of TH-cre virus. The results showed that the infection efficiency and the specificity of Th-cre virus used in the current research both reached over 90%, which is basically consistent with the literature cited above. However, it should be pointed out that, due to the technical limitations of promoter-dependent strategies, such as promoter independent gene expression, it is inevitable that there is partial

expression in a small number of non-dopamine neurons in TH-Cre or DAT-cre strategies. The reviewer mentioned that the expression specificity of TH-cre transgenic mice (Lammel et al. 2005 Neuron "Matters arising") is relatively lower than DAT-Cre mice. For this specific case, the reasons are not clearly explained in the article. To our knowledge, one should carefully screen the TH-promoter sequence when constructing virus or transgenic mice, and do experiment using the virus or mice strain with the highest infection efficiency and specificity to ensure the specific expression in bona fide VTA DA neurons.

- L. 249 states: " These findings ~~further~~ support the ~~specific~~ role of VTA^{DA} neurons in the anxiety-like behavior of CES."

=> please remove "further" and "specific" from this sentence.

Response: Thank you very much for this comment. As the Reviewer suggested, we have updated this sentence by removing "further" and "specific" in the revised manuscript [Line 248].

Suppl. Fig. 4

- This data should be shown as a main Figure - the finding that acute chemogenetic inhibition also dampens CES-induced anxiety seems important.

Response: Thank you very much for this important comment. As the Reviewer suggested, those data have been shown as a main Figure (Fig. 5 in the revised manuscript).

- Nevertheless, I don't understand the sentence:

L. 258: However, comparing with other groups, acute CNO injection did not alter the time spent in the social interaction zone after 10-day CES (Supplementary Fig. 4j, k)

=> In Fig. 4p, q, there was similarly no change, so I don't quite understand the above statement (?). Please explain ...

Response: Thank you very much for this comment. We apologize for not making this clear. The related sentences have been carefully checked and amended [Line 257]. In Fig. 4p, q, CNO was injected 30 min before each day of CES session, and SIT were tested after 10 days. The results demonstrated that VTA^{DA} neuron inhibition during CES did not affect social interaction behavior in the SIT, suggesting that VTA^{DA} neurons are not necessary to social interaction behavior. In comparison, in Suppl. Fig. 4j, k (Fig. 5j, k in the revised manuscript), mice were exposed to 10-day CES, and SIT were tested 30 min after injecting CNO on day 11. We found that comparing with other groups, one-time acute VTA dopamine inhibition prior to SIT did not alter the time spent in the social interaction zone after 10-day CES, also suggesting that VTA^{DA} neurons are not necessary to social interaction behavior. Collectively, the results of the two experiments were similar.

Figure 5

... shows with rabies virus that the VTA back-labels neurons in NAc (Fig. 5a-e)(this is not new; see Watabe-Uchida 2012 Neuron), and that "VTA-DA" neurons, that get contacted from NAc (analyzed with an AAV1 anterograde jumping approach") seem to be active during the Elevated plus maze (EPM) test in CES mice.

Response: Thank you very much for this comment. The reference has been incorporated into the revised manuscript [Line 265].

BUT:

- the histological example of Suppl. Fig. 6a (esp. right panel) shows MANY neurons clearly OUTSIDE OF THE NORMAL TH EXPRESSION AREA, which are labeled by GCamp6s (see green cells dorsally). Is it possible that the TH promoter in the AAV2/1:TH:Cre virus is "leaky"? Again, as above, a detailed analysis, with zoom-ins of apparently "green" cells (but which might in reality be slightly "yellow", and of "red" cells ect. needs to be done, to convince your self and the reviewers that expression of GCamp6s in indeed limited to "TH⁺", and thus dopamine neurons. It would be even better, if these and similar experiments could be re-done with a DAT-Cre mouse, which should restrict expression much better to bona fide VTA dopamine neurons.

Response: Thank you very much for this comment. We understand the Reviewer's concern here. In order to express GCamp6s in the VTA^{DA} neurons receiving NAc projections, we used a cre-dependent, anterograde transsynaptic AAV2/1:TH-Cre virus, which were injected into NAc and express TH-cre in the VTA^{DA} neurons receiving NAc projections (Supplementary Fig. 7 in the revised manuscript). To ensure both the high expression efficiency and high specificity of AAV2/1:TH-Cre virus, we tested several virus strains and chose the strain with the highest specificity. We did immunofluorescence double-staining using TH antibody to verify the infection efficiency and specificity of AAV2/1:TH-Cre virus. The results showed that the infection efficiency and the specificity of AAV2/1:TH-Cre virus used in the current research is 94.6% and 91.9% respectively, which is basically consistent with previous study (Xin-Yu Su et al, 2019, Neuron, fig.1A-C, as shown below)¹⁷.

Due to the technical limitations of promoter-dependent strategies, it is inevitable that there is partial expression in a small number of non-dopamine neurons in Th-cre strategies. In this experiment, we need transsynaptic expression of GCamp6s in the VTA^{DA} neurons receiving NAc projections, to this end,

anterograde transsynaptic AAV2/1:TH-Cre virus, but not DAT-Cre mice, were used.

- Again, as before, the analysis of Ca transients remains enigmatic (Fig. 5p, $\Delta F/F_0$ about 0.17 %, see also above comment regarding Fig. 2r).

Response: We thank the Reviewer for this comment. In the test of *in vivo* fiber photometry, we derived the values of fluorescence change by calculating $\Delta F/F_0$, where ΔF is the variation of fluorescence between each sampling (sampling frequency 20Hz), and F_0 is the averaged fluorescence baseline in the whole duration of the entire test (5 minutes). On this basis, the values of fluorescence change (ΔF) between each sampling (50ms) is relatively small, which is also consistent with previous research findings⁶.

Figure 6

... shows that optogenetic inhibition of (non-specific) GABA neurons in the NAc causes anxiety in the open field test (Fig. 6a-k). However, although this reviewer is not a specialist in the NAc brain area, similar findings on general anxiety must have been made before.

Response: Thank you very much for this comment. Accordingly, we have added the related discussion in the revised manuscript [Line 444-451]. The NAc is a vital component in the reward circuitry, which responds to stress signals and has a dominant effect on anxiety regulation¹⁸. Dopamine 1 (D1)-Medium spiny neurons (MSNs) in the NAc play roles in modulating reward-related responses¹⁹ whereas D2-MSNs regulate anxiety-like aversion or avoidance behavior²⁰. It is reported that GABAergic somatostatin projection from the BNST to NAc controls anxiety²¹. Here, we demonstrated that NAc neurons have long-range projections that densely innervate the VTA^{DA} neurons, and the NAc-VTA circuit mediates the anxiety-like behavior of CES.

Furthermore, there is again a concern in the quantification of expression overlap. The example images in Fig. 6c clearly show MANY neurons that are "green" alone, more than the ~ 25% visible in the (newly added) quantification of Fig. 6d. here again, I would like to see quantifications in the form of Venn diagrams, to show in an unbiased fashion, "green" only cells (GAD67+ cells - cells which express both markers - and "mCherry" only cells.

Response: Thank you very much for this comment. In Fig. 6c (Fig. 7c in the revised manuscript), starter cells (yellow) in NAc which are co-infected by AAV2/9-GAD67-EGFP-2A-Cre (green) and AAV2/9-DIO-mCherry (red). The green cells just represent infected NAc GABAergic neurons. As the Reviewer suggested, we have changed the pie chart to the Venn diagram in Fig. 7d in the revised manuscript.

Given the uncertainties in the example images, the statement on l. 312 ("Post hoc imaging showed that mCherry and EGFP expression was restricted to NAc GABAergic neurons (Fig. 6c, d)." is overly optimistic.

Response: Thank you very much for this comment. As the Reviewer suggested, we have made some modifications in the revised manuscript [Line 311].

... the bottom part of Fig. 6 then uses optogenetic stimulation of output fibers from the NAc in the VTA, to report that activation of NAc fibers in the VTA (which should in principle inhibit VTA neurons) during the 10 day CES procedure indeed decreases CES-induced anxiety in the OFT and EPM. The problem with this experiment could be, however, that NAc axons activate not only TH⁺ dopamine neurons, but also a substantial number of non-dopaminergic neurons, as the histology images with the anterograde- jumping virus suggest (see above, comment to Suppl. Fig. 6a).

Response: We thank the Reviewer for this comment. Our results of viral tracing (Fig. 6 in the revised manuscript) showed that VTA^{DA} neurons receive direct, monosynaptic GABA inputs from the NAc. Previously published studies have indicated that the principal neurons of NAc are the medium spiny GABAergic neurons, which take approximately 95% of the projection neurons in this region^{22, 23}. In the bottom part of Fig. 6 (Fig. 7 in the revised manuscript), to test the effects of activation of NAc-VTA circuit, mice were injected with AAV2/9-GAD67-EGFP-2A-Cre and AAV2/9-DIO-ChR2-mCherry (ChR2) using AAV2/9-DIO-mCherry as control, into the NAc. Optic fibers were implanted into the VTA. In this way, just VTA-projecting NAc GABAergic neurons could be activated.

In summary:

I think the various issues with presenting raw data in an open manner, to make sure correct conclusions can be drawn, should now be addressed with high priority for Figures 1 -6, along my detailed review (most of those points had been raised in my first review). Especially worrisome is the possibility that the use of AAV virus vectors with a "TH" promoter, might not impart sufficiently high selectivity for VTA dopamine neurons (see Lammel et al. 2005 Neuron "Matters arising"). [Redacted]

Response: We thank the Reviewer for this comment. [Redacted]

As for the efficiency and specificity of TH-Cre viruses used in the current study, we did immunofluorescence double-staining using TH antibody to verify the infection efficiency and specificity of TH-cre virus. The results showed that the infection efficiency and the specificity of Th-cre virus used in the current

research both reached over 90%, which basically ensure the specific expression in bona fide VTA DA neurons, and is consistent with the other literature cited above.

References to Reviewer #3:

1. Golden SA, Covington HE, 3rd, Berton O, Russo SJ. A standardized protocol for repeated social defeat stress in mice. *Nat Protoc* **6**, 1183-1191 (2011).
2. Lorsch ZS, *et al.* Stress resilience is promoted by a Zfp189-driven transcriptional network in prefrontal cortex. *Nat Neurosci* **22**, 1413-1423 (2019).
3. Shen CJ, *et al.* Cannabinoid CB1 receptors in the amygdalar cholecystokinin glutamatergic afferents to nucleus accumbens modulate depressive-like behavior. *Nat Med* **25**, 337-349 (2019).
4. Ghali MGZ. Rubral modulation of breathing. *Exp Physiol* **104**, 1595-1604 (2019).
5. Tye, K. M. *et al.* Dopamine neurons modulate neural encoding and expression of depression-related behaviour. *Nature* **493**, 537-541 (2013).
6. Carta I, Chen CH, Schott AL, Dorizan S, Khodakhah K. Cerebellar modulation of the reward circuitry and social behavior. *Science* **363**, (2019).
7. Schindelin, J. *et al.* Fiji: an open-source platform for biological-image analysis. *Nat Methods* **9**, 676-682 (2012).
8. Guirado, R., Carceller, H., Castillo-Gómez, E., Castrén, E., Nacher, J. Automated analysis of images for molecular quantification in immunohistochemistry. *Heliyon* **4**, e00669 (2018).
9. Jensen, E. C. Quantitative analysis of histological staining and fluorescence using ImageJ. *Anat Rec (Hoboken)* **296**, 378-381 (2013).
10. Arganda-Carreras, I. *et al.* Trainable Weka Segmentation: a machine learning tool for microscopy pixel classification. *Bioinformatics* **33**, 2424-2426 (2017).
11. Parker, K. E. *et al.* A Paranigral VTA Nociceptin Circuit that Constrains Motivation for Reward. *Cell* **178**, 653-671.e619 (2019).
12. Groessl, F. *et al.* Dorsal tegmental dopamine neurons gate associative learning of fear. *Nat Neurosci* **21**, 952-962 (2018).
13. Saunders, B. T., Richard, J. M., Margolis, E. B., Janak, P. H. Dopamine neurons create Pavlovian conditioned stimuli with circuit-defined motivational properties. *Nat Neurosci* **21**, 1072-1083 (2018).
14. Sofia Beas, B. *et al.* A ventrolateral medulla-midline thalamic circuit for hypoglycemic feeding. *Nat Commun* **11**, 6218 (2020).
15. Valyear, M. D. *et al.* Dissociable mesolimbic dopamine circuits control responding triggered by alcohol-predictive discrete cues and contexts. *Nat Commun* **11**, 3764 (2020).

16. Xia, S. H. et al. Chronic Pain Impairs Memory Formation via Disruption of Neurogenesis Mediated by Mesohippocampal Brain-Derived Neurotrophic Factor Signaling. *Biol Psychiatry* **88**, 597-610 (2020).
17. Su, X. Y. et al. Central Processing of Itch in the Midbrain Reward Center. *Neuron* **102**, 858-872.e855 (2019).
18. Bewernick BH, et al. Nucleus accumbens deep brain stimulation decreases ratings of depression and anxiety in treatment-resistant depression. *Biol Psychiatry* **67**, 110-116 (2010).
19. Yang H, de Jong JW, Tak Y, Peck J, Bateup HS, Lammel S. Nucleus Accumbens Subnuclei Regulate Motivated Behavior via Direct Inhibition and Disinhibition of VTA Dopamine Subpopulations. *Neuron* **97**, 434-449.e434 (2018).
20. Blomeley C, Garau C, Burdakov D. Accumbal D2 cells orchestrate innate risk-avoidance according to orexin signals. *Nat Neurosci* **21**, 29-32 (2018).
21. Xiao Q, et al. A new GABAergic somatostatin projection from the BNST onto accumbal parvalbumin neurons controls anxiety. *Mol Psychiatry*, (2020).
22. Kohnomi S, Konishi S. Multiple actions of a D(3) dopamine receptor agonist, PD128907, on GABAergic inhibitory transmission between medium spiny neurons in mouse nucleus accumbens shell. *Neurosci Lett* **600**, 17-21 (2015).
23. Koo JW, et al. Loss of BDNF signaling in D1R-expressing NAc neurons enhances morphine reward by reducing GABA inhibition. *Neuropsychopharmacology* **39**, 2646-2653 (2014).

Reviewers' comments:

Reviewer #3 (Remarks to the Author):

There are several unsolved concerns with the manuscript, foremost the apparent low specificity of the AAV:TH promoter approach, which likely leads to the infection of non-dopaminergic neurons. Although Venn diagrams were added to illustrate the degree of co-expression, the example images suggest that the co-expression must be of a significantly smaller degree. Thus, it remains possible that VTA GABA neurons contributed to the effects described here. The most important experiments should therefore be conducted again with a DATCre mouse, to limit the expression of Ca sensors / optogenetic actuators to bona-fide dopamine neurons. This is necessary to validate the main conclusions of the paper.

Point-by-point response to Reviewer # 3's comments

Response to Reviewer # 3:

There are several unsolved concerns with the manuscript, foremost the apparent low specificity of the AAV:TH promoter approach, which likely leads to the infection of non-dopaminergic neurons.

Response: We greatly appreciate the reviewer's meaningful comments. To make our results more convincing, we have performed additional experiments according to the suggestion (Supplementary Fig. 4, 5, 7, 10). And the detailed description of the results about the additional experiments was shown in the last point.

Additionally, a challenge with Cre-driven transgenics is the validation that expression of the Cre recombinase faithfully replicates the endogenous gene expression regulated by the selected promoter. In the ventral midbrain, TH, the rate-limiting enzyme in the synthesis of DA, is considered to be the gold standard for identifying DA neurons and is now the most common genetic "handle" used to drive exogenous gene expression. Accordingly, TH-Cre virus or knock-in lines have been used in ~80% of the studies that involved Cre-dependent targeting of midbrain DA neurons. The use of TH-Cre strategies to selectively target VTA DA neurons had come under fire due to a Matter Arising report in Neuron journal ¹. Lammel et al. reported that following Cre-inducible virus injection into the VTA of TH-Cre or DAT-Cre mice, there is appreciable viral-mediated expression in both TH-positive and TH negative neurons in and around the VTA in the TH-Cre line, while viral expression is restricted to VTA TH-positive neurons in the DAT-Cre line. While, we've also noticed that there is a direct Matter Arising Response ² to the Lammel's paper in the same issue of Neuron journal focus on the TH-Cre specificity. Stuber et al. proved that the lower TH-positive fraction in Lammel's paper may occur because immunofluorescent staining may not reliably label neurons that produce low levels of TH protein. It takes only a small amount of Cre to allow viral insertion and subsequent expression, whereas it requires substantial amounts of TH protein for detection by conventional immunostaining. The "TH-negative neurons" in TH-Cre lines of Lammel's paper is mainly due to sufficient Cre expression in neurons only weakly expressing TH mRNA or

protein. This may be less of an issue with the DAT-Cre mouse line since DAT is enriched in cells that also express high levels of TH³. Therefore, the discrepancy between the percentage of overlap between eYFP+ and TH+ neurons reported in Lammel et al. and other studies^{3, 4, 5, 6, 7, 8, 9, 10, 11, 12} could be due to differences in immunohistochemical techniques and confocal imaging parameters in addition to the quantification of particular VTA subregions as they discussed. In the current study, the specificity of TH-Cre is quite high (>90%) compared with other studies published recently, and we mainly tested the animal behavior and calcium imaging of population cells, which is enough to ensure that a very small number of TH-negative neurons with ectopic expression do not affect the preciseness of our experimental conclusion.

Although Venn diagrams were added to illustrate the degree of co-expression, the example images suggest that the co-expression must be of a significantly smaller degree. Thus, it remains possible that VTA GABA neurons contributed to the effects described here.

Response: Thanks for the reviewer's valuable comments. As suggested, we have changed the pie chart to the Venn diagram in the whole revised manuscript, and added additional information to show the number of cells analyzed (cells only positive for TH, cells only positive for GCamp6s or mCherry, and double-positive cells), number of sections analyzed in addition to the number of mice.

Take Fig 3f as an example, after the behavioral experiment was completed, mice were transcardially perfused, and brain slices were prepared for TH staining analysis. Three sections per mouse from 6 mice were chose to analyse the number of cells (as shown below picture). For automated cell counting of fluorescent protein-positive neurons, the fluorescence images were analyzed and quantitated by ImageJ Fiji plugins and features. Briefly, the red channel and green channel of fluorescence images were separately segmented by Trainable Weka Segmentation plugin firstly. Secondly, a useful feature of ImageJ is the ROI Manager that allows selection of specific areas (such as red only, green only, and double-labelled) for evaluation. Finally, the cells with red only, green only, and double-labelled were automatically counted by "Analyze Particles" tool (Lower threshold level: 50; Upper threshold level: 255) (as shown below table).

In addition, to compare multiple specimens, staining, image acquisition (exposure time and gain), and image analysis were performed in parallel for the entire set.

mouse #	slice #	mCherry	TH	mCherry ⁺ TH ⁺
1	1	115	119	114
	2	126	130	123
	3	134	140	132
2	1	109	115	102
	2	90	94	87
	3	86	89	78
3	1	119	113	110
	2	126	127	118
	3	112	108	99
4	1	116	118	115
	2	145	149	138
	3	135	136	125
5	1	118	121	116
	2	130	132	127
	3	115	121	105
6	1	120	122	114
	2	107	112	103
	3	100	102	95

	mCherry	TH	mCherry ⁺ TH ⁺
Total	2103	2148	2001
NO.	102	147	
%	4.85	6.84	

f

In our study, we used TH antibody to do immunofluorescence double-staining to verify the infection efficiency and specificity of TH-Cre virus. The results showed that the infection efficiency and the specificity of TH-Cre virus used in our research both reached over 90%, which basically ensure the specific expression in bona fide VTA DA neurons, and is consistent with the other literatures^{7, 8, 9, 10, 11, 12, 13}. However, it should be pointed out that, due to the technical limitations of promoter-dependent strategies, such as promoter

independent gene expression, it is inevitable that there is partial expression in a small number of non-dopamine neurons in TH-Cre or DAT-Cre strategies.

The most important experiments should therefore be conducted again with a DATCre mouse, to limit the expression of Ca sensors / optogenetic actuators to bona-fide dopamine neurons. This is necessary to validate the main conclusions of the paper.

Response: Thanks for the reviewer's valuable comments and significant suggestions. Accordingly, we have performed the additional experiments in DAT-Cre mice.

Firstly, we conducted a new experiment to detect the neuronal activity of VTA^{DA} neurons of DAT-Cre mice in anxiety-inducing events by *in vivo* calcium recording. The AAV2/9-DIO-GCaMP6s was injected into the VTA of DAT-Cre mice. We found that the Ca²⁺ signal of VTA^{DA} neurons of DAT-Cre mice increases in anxiety-inducing events (Supplementary Fig. 4a–f). This result is in line with our previous research, which showed that VTA^{DA} neurons were strongly activated by emotional stress (Fig. 2g–k).

Next, we investigated how VTA^{DA} neurons of DAT-Cre mice are engaged during the exploration of innately anxiogenic environments. DAT-Cre mice were infused with AAV2/9-DIO-GCaMP6s and optic fiber cannula was implanted over the same site, followed by 10-day of CES modeling was conducted. In agreement with previous studies (Fig. 2l–x), we also found that the VTA^{DA} neurons of DAT-Cre mice showed significant activation of GCaMP6s activity in anxiety-inducing contexts (Supplementary Fig. 5a–m).

Furthermore, the effects of chemogenetic activation of VTA^{DA} neurons of DAT-Cre mice *in vivo* on behavioral changes were tested (Supplementary Fig. 7a–m). We bilaterally infected AAV2/9-DIO-hM3Dq-mCherry in VTA of DAT-Cre mice, and measured the anxiety-like and social interaction behavior induced by VTA^{DA} neuron activation by CNO administration (Supplementary Fig. 7a, b). Post hoc immunofluorescent examination verified that hM3Dq-mCherry expression was primarily restricted to DA neurons in the VTA region (Supplementary Fig. 7c, d). Compared to mCherry and saline controls, DAT-Cre mice with CNO injections significantly reduced the exploration time and travel distance in the central area of OFT but showed increment on locomotor activities (Supplementary Fig. 7e–h). Similarly, CNO-mediated VTA^{DA} neuron activation

decreased the time spent in the open arms (Supplementary Fig. 7i, j). No significant differences were observed in the number of open arm entries (Supplementary Fig. 7k). Moreover, we assessed the effects of VTA^{DA} neuron activation in the SIT. CNO injection did not alter the time spent in the social interaction zone compared to the mCherry and saline controls (Supplementary Fig. 7l, m). These results demonstrate that activating DA neurons in the VTA of DAT-Cre mice contributed to the expression of anxiety but displayed little response to social interaction behavior, which was in line with previous research (Fig. 3).

Finally, we further clarified the upstream circuit of VTA^{DA} neurons. Cre-dependent helper viruses (AAV2/9-DIO-TVA-EGFP and AAV2/9-DIO-RVG) were injected into the VTA of DAT-Cre mice. After 14 days of virus expression, DAT-Cre mice were again anesthetized as previously described, and the rabies virus RV-ENVA-ΔG-DsRed was infused into the same site in the VTA (Supplementary Fig. 10a, b). Nine days post RV-ENVA-ΔG-DsRed infusion, DAT-Cre mice were transcardially perfused, and brain slices were prepared for tracing DsRed for immunofluorescence staining analysis. We found that the VTA-projecting NAc neurons are GABAergic (Supplementary Fig. 10c), which was consistent with our previous studies (Fig. 6f–h).

More pleasing, employing the same strategy in DAT-Cre mice, we found that the results are in agreement with our previous studies that TH-Cre viruses was used, which demonstrates that VTA^{DA} neurons are essential for emotional stress-induced anxiety-like behavior.

References for Reviewer # 3:

1. Lammel S, *et al.* Diversity of transgenic mouse models for selective targeting of midbrain dopamine neurons. *Neuron* **85**, 429-438 (2015).
2. Stuber GD, Stamatakis AM, Katak PA. Considerations when using cre-driver rodent lines for studying ventral tegmental area circuitry. *Neuron* **85**, 439-445 (2015).
3. Stamatakis AM, *et al.* A unique population of ventral tegmental area neurons inhibits the lateral habenula to promote reward. *Neuron* **80**, 1039-1053 (2013).
4. Tsai HC, *et al.* Phasic firing in dopaminergic neurons is sufficient for behavioral conditioning. *Science* **324**, 1080-1084 (2009).

5. Chaudhury D, *et al.* Rapid regulation of depression-related behaviours by control of midbrain dopamine neurons. *Nature* **493**, 532-536 (2013).
6. Tye KM, *et al.* Dopamine neurons modulate neural encoding and expression of depression-related behaviour. *Nature* **493**, 537-541 (2013).
7. Parker KE, *et al.* A Paranigral VTA Nociceptin Circuit that Constrains Motivation for Reward. *Cell* **178**, 653-671 e619 (2019).
8. Groessl F, *et al.* Dorsal tegmental dopamine neurons gate associative learning of fear. *Nat Neurosci* **21**, 952-962 (2018).
9. Saunders BT, Richard JM, Margolis EB, Janak PH. Dopamine neurons create Pavlovian conditioned stimuli with circuit-defined motivational properties. *Nat Neurosci* **21**, 1072-1083 (2018).
10. Sofia Beas B, *et al.* A ventrolateral medulla-midline thalamic circuit for hypoglycemic feeding. *Nat Commun* **11**, 6218 (2020).
11. Valyear MD, *et al.* Dissociable mesolimbic dopamine circuits control responding triggered by alcohol-predictive discrete cues and contexts. *Nat Commun* **11**, 3764 (2020).
12. Xia SH, *et al.* Chronic Pain Impairs Memory Formation via Disruption of Neurogenesis Mediated by Mesohippocampal Brain-Derived Neurotrophic Factor Signaling. *Biol Psychiatry* **88**, 597-610 (2020).
13. Yu W, *et al.* Periaqueductal gray/dorsal raphe dopamine neurons contribute to sex differences in pain-related behaviors. *Neuron* **109**, 1365-1380 e1365 (2021).